# Drivers and impacts of sediment deposition in Amazonian floodplains

**Dongyu Feng** [1] ✉, **Zeli Tan** [1] ✉, **Sebastien Pinel** [2], **Donghui Xu** [1], **João Henrique Fernandes Amaral**[3], **Alice César Fassoni-Andrade**[4], **Marie-Paule Bonnet** [5] & **Gautam Bisht** [1]

The Amazon River carries enormous amounts of sediment from the Andes mountains, much of which is deposited in its floodplains. However, accurate quantification of the sediment sink at fine spatiotemporal scales is still challenging. Here, we present a high-resolution hydrodynamic-sediment model to simulate sediment deposition in a representative Amazon/Solimões floodplain. The process is found to be jointly driven by inundation, suspended sediment concentration in the Amazon River, and floodplain hydrodynamics and only weakly correlated with inundation level. By upscaling the sediment deposition rate ($1.33 \pm 0.24$ kg m$^{-2}$ yr$^{-1}$), we estimate the trapping of $77.3 \pm 13.9$ Mt (or $6.1 \pm 1\%$) of the Amazon River sediment by the Amazon/Solimões floodplains every year. Widespread deforestation would reduce the trapping efficiency of the floodplains over time, exacerbating downstream river aggradation. Additionally, we show that the deposition of sediment-associated organic carbon plays a minor role in fueling carbon dioxide and methane emissions in the Amazon.

The Amazon River is the world's largest hydrological system with a drainage area of 6.1 million km$^2$ [1]. It delivers 17.8% of global freshwater (with an annual discharge rate of 6642 km$^3$ yr$^{-1}$)[2] and roughly 1200 million tons (Mt) of sediment to the Atlantic Ocean each year[3,4]. Sediments transported from their origins in Guyana/Brazilian Shield and Andes to the Atlantic Ocean undergo a complex journey with repeated cycles of deposition and resuspension[5]. A substantial portion of the sediment does not make its way to the ocean but is instead deposited over Amazonian floodplains[6]. This deposition helps shape the diverse riparian and lacustrine landscapes comprising 14% of the lowland area (<500 m altitude) in the Amazon Basin[7,8].

Sediment deposition affects river geomorphology[9] and ecosystems[10], and enriches the soil of the Amazon Basin with organic carbon and nutrients. This sustains high biodiversity and productivity in the region[8,11] while fueling carbon dioxide ($CO_2$) and methane ($CH_4$)[12,13] emissions. Despite its importance, our understanding of

sediment deposition drivers in Amazonian floodplains remains limited. Existing studies suggest that the Amazon River inundation controls sediment deposition in the floodplains near the Andes[14], where coarser sands are rapidly deposited during high water periods. However, this observation may not be applicable to the downstream floodplains, such as the Amazon/Solimões, which comprise the majority of Amazonian floodplain area. As the river progresses downstream, sediment composition transitions to finer sediments due to preferential deposition of coarse sediments, sediment attrition, and dilution by finer sediments from black-water tributaries[15]. Thus, sediment deposition in the Amazon/Solimões floodplains is less likely to be influenced by river inundation magnitude and moreso by local factors, such as inundation phases, river sediment concentration, and floodplain hydrodynamics.

Human impacts also remain a poorly understood influence on sediment deposition in Amazonian floodplains. Despite low

[1]Atmospheric, Climate, & Earth Sciences Division, Pacific Northwest National Laboratory, Richland, WA, USA. [2]Centre of Education and Research on Mediterranean environments (CEFREM), University of Perpignan Via Domitia (UPVD), Perpignan, Perpignan, France. [3]Earth System Science Program, Faculty of Natural Sciences, Universidad del Rosario, Bogota, Colombia. [4]Instituto de Geociências, Universidade de Brasília, Brasília, Brazil. [5]UMR Espace-DEV, Univ Montpellier, IRD, Montpellier, Montpellier, France. ✉e-mail: dongyu.feng@pnnl.gov; zeli.tan@pnnl.gov

population density, sediment dynamics in the Amazon are not immune to anthropogenic disturbance[16–18], especially from dam construction and land use change. Numerous dams on the mainstream and tributaries of the Amazon River have already trapped substantial amounts of sediment[19–21]. It is estimated that many proposed hydropower plants in the Amazon, without prioritizing hydrological connectivity, can potentially reduce sediment transport to downstream river reaches by up to 100%[22]. Additionally, the Amazon Basin is witnessing widespread deforestation activities[23–25], particularly in the south and southeast regions[26]. Soil erosion increased 300% in the sub-basins of the Madeira, Solimões, Xingu, and Tapajós due to the rapid expansion of cropland and rangeland[26]. Climate change is another critical driver of changes in Amazon sediment dynamics[14,27,28]. Increasing trends of suspended sediment concentrations (SSC) in the Amazon River could be the result of the increase of annual peak flow over the past two decades[29,30]. However, future projections show that the Amazon will become drier with time[28]. These aforementioned factors culminate in complex and changing trends in sediment dynamics for the Amazon River and its surrounding floodplain. To better understand these dynamics, there is a critical need for a modeling framework which accurately represents the interactions among different drivers.

The sediment budget of the Amazon River is well measured via a network of gauge stations on the mainstream and major tributaries[4] and remote sensing observations[31], but accurate representation of sediment dynamics within Amazonian floodplains remains challenging. Both in-situ data and estimates of SSC derived from remote sensing imagery have limited ability to resolve the processes in sufficient spatial and temporal resolution. For example, though the underlying drivers of sediment dynamics, such as inundation, fluctuate usually on a daily basis and vary significantly in space across the complex landscapes at a 10-m scale[32], in-situ sediment sampling was only occasionally conducted in specific floodplain locations for short time periods[33,34]. Meanwhile, despite the broad coverage from remote sensing technologies over the Amazon Basin floodplains[31,35,36], the data resolution remains coarse and its uncertainty is often high due to the empirical nature of the methodology, signal degradation by cloud cover, macrophytes, and shallow sediment beds[37–40]. Current measures of sediment-associated particulate organic carbon (POC) deposition also lack sufficient spatial and temporal details. In particular, they are too coarse to determine which landscape and oxic condition POC is dominantly deposited in Amazonian floodplains[41], and this is a determinant factor for the transformation pathways of POC in floodplain soils[42,43].

Numerical sediment modeling that couples a high-fidelity two-dimensional (2-D) hydrodynamic model with a detailed sediment model offers a way to overcome these limitations[44]. Previous applications of 2-D hydrodynamic models in the central Amazon Basin[45,46] as well as its floodplains[47,48] have shown promising performance in resolving inundation extent, water depth, and the river-floodplain flow exchange. But these hydrodynamic models were used either solely for flooding simulations[32,49] or together with sparse SSC data for quantifying the overall sediment budget in a floodplain[40]. Here, we used a recently developed hydrodynamic model, now coupled with sediment dynamics, to simulate high-resolution sediment deposition.

We selected a representative site of the Solimões floodplains, Janauacá, to investigate sediment dynamics and sediment/carbon deposition in Amazonian floodplains. Janauacá (3.2–3.25°, 60.23–60.13°) is a medium-size Amazonian floodplain along the right bank of Amazon/Solimões River, 40 km upstream from the confluence of the Amazon River and Rio Negro at Manaus, Brazil (Fig. 1a). This area is a typical subsystem of the Amazon River, consisting of floodplain forests, wetlands, river channels, and a lake. While its water stage follows a monomodal cycle, the water circulation pattern within the floodplain demonstrates significant complexities[32]. Janauacá has a small drainage area (~786 km²)[50], receiving annual precipitation

accumulations of ~2000 mm and little freshwater inflow from upstream rivers[51]. This floodplain connects to the Amazon River via a perennial channel (12 km in length; 100–200 m in width) with levees located in its northeastern section, where exchanging fluxes occur during rising and falling water periods. The flooded area varies from 23 to 390 km² during low water and high water, respectively. The water stage in Janauacá changes accordingly between below 11–24 m. The floodplain is progressively filled with sediment[52], which mainly sources from the "white-water" Amazon/Solimões River loaded with suspended sediments from the Andes. In contrast, the upland catchment around the southern lake drains with black and clear water streams of low suspended solids concentration[33]. The sediment-rich white water has critical impacts on the downstream soil fertility[53]. In-situ sampling of water level, velocity, sediment, other suspended matters, and carbon fluxes was established in Janauacá over the last few years[33] which helps improve the understanding of Amazonian floodplains. The data have been used to quantify $CO_2$ and $CH_4$ dynamics[54] and their response to vegetation and inundation[55]. The hydrodynamic model has been rigorously validated in representing floodplain hydrodynamics, including both water stage and flow current[32]. Our advanced model enables the estimation of sediment deposition in Amazonian floodplains at unprecedented spatial and temporal details. It also facilitates disentangling the drivers of sediment deposition and its impact on carbon cycling in the Amazon.

## Results

The hydrodynamic-sediment coupled model performs well in simulating water level and SSC in the Janauacá floodplain (Fig. 1b–e). As measured, the simulated water level in our computational domain follows four hydrological periods (LW: low water; RW: rising water; HW: high water; FW: falling water, see Method section for definitions) of the main course of the Amazon River tightly (Supplementary Fig. S2). The observed and modeled SSC are consistent in different locations of the floodplain, including the upstream river channel, the downstream river channel, and the open lake area (Fig. 1d). The SSC dynamics exhibit an annual monomodal cycle but asynchronous relationships with the hydrological cycle, with the water level peak lagged by over 4 months from the annual maximum SSC. Specifically, high and low SSC concentrations occur during RW and HW periods, respectively. The impacts of the water and sediment influx from upstream rivers on SSC dynamics are minor, whereas the largest contribution is from the Amazon River (Supplementary Fig. S5).

Sediment deposition in Janauacá is driven jointly by the hydrological cycle, SSC variations in the Amazon River, and floodplain hydrodynamics (Fig. 2). The monthly variations of the total sediment deposition in the floodplain and the sediment trapping ratio closely follow that of sediment influx from the Amazon River rather than water level(Fig. 2a, b). In Janauacá, sediment influx is shaped by the asynchronous variations of water level and SSC in the Amazon River (Supplementary Fig. S2). On average, 0.17 Mt $yr^{-1}$ of sediment was deposited on the Janauacá floodplain from 2007 to 2016, or equivalently 0.033 Mt $km^{-1}$ $yr^{-1}$ of sediment was deposited per unit river length as the adjacent boundary of Janauacá with the Amazon River is about 5 km long.

The interannual variability of sediment deposition in the floodplain, however, shows a weak negative correlation (−0.36) against water level, the indicator of river flooding magnitude. Conversely, it shows a strong positive correlation (0.94) with the SSC in the Amazon River (Fig. 2c). Consequently, sediment deposition in the floodplain remained unexpectedly intensive during the years of extremely low water levels in 2010 and 2011 when the river SSC was high, and fell to the lowest in 2015 when the water level was normal but SSC was the lowest.

Due to the effect of floodplain hydrodynamics, the simulated sediment deposition shows large and distinct spatial variability during

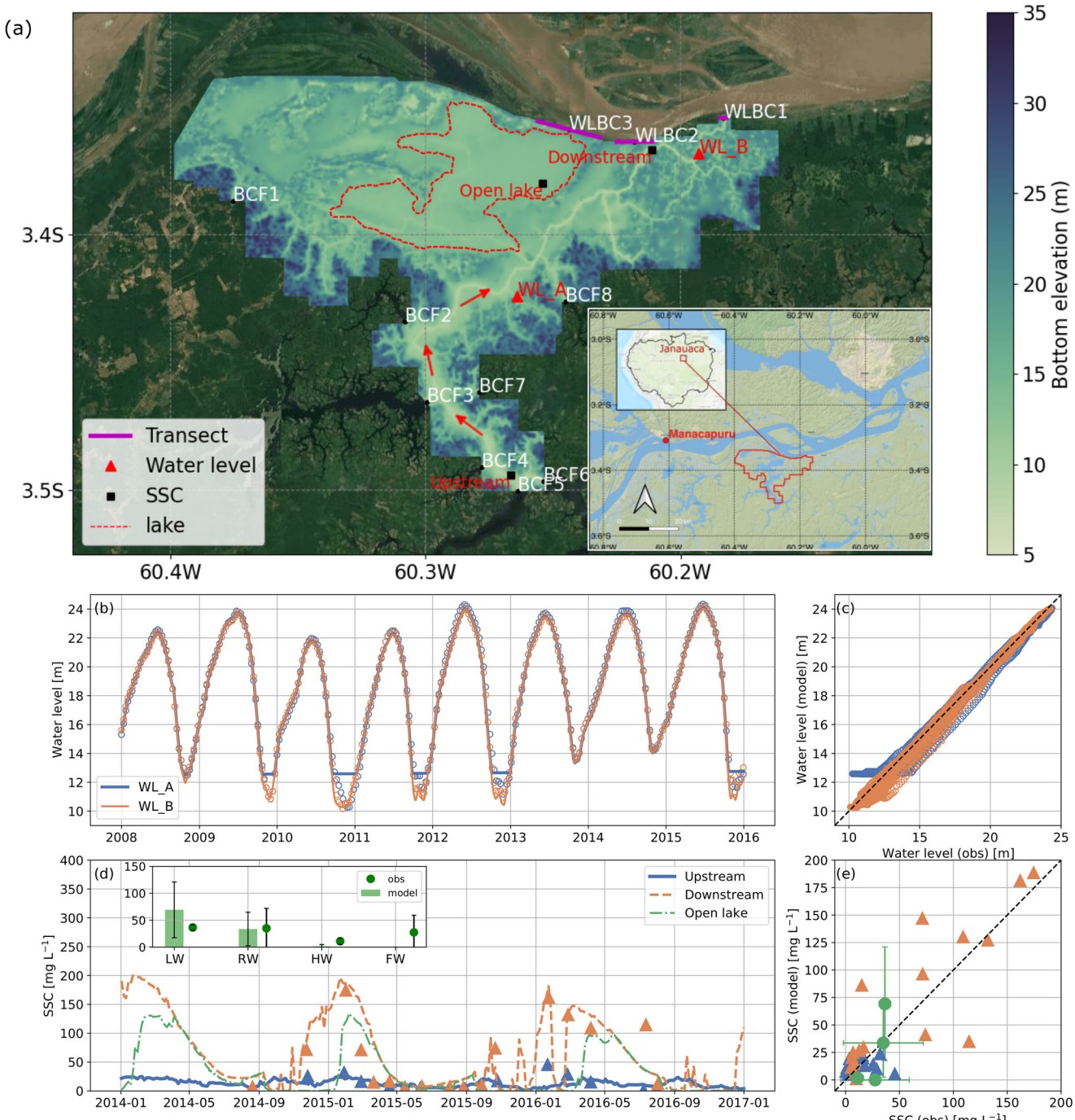

**Fig. 1 | The setup and validation of the hydrodynamic-sediment coupled model. a** The model's computational domain (limited to the low-lying floodable regions with bottom elevation < 29 m). White labels show 8 inflow boundaries (BCF) and 3 water level open boundaries (WLBC) connecting the main channel of the Amazon River. Magenta transects at WLBCs are used to calculate the flow and sediment exchange flux between the floodplain and the Amazon River main channel. Red labels show two sampling sites of water level at WL_A and WL_B (red triangles) and three sampling sites for SSC at Upstream, Open Lake, and Downstream (black squares). The red dashed line outlines the lake area, which is used to estimate the lake-specific POC deposition and the corresponding oxygen exposure time (OET). The arrows in the upstream region indicate the river flow direction. The inset shows the minimized view of the domain boundary and the location of the Manacapuru gauge at the mainstream of the Amazon River. Comparison of the modeled water level (**b**, **c**) and SSC (**d**, **e**) (solid and dashed lines) against the measurements (circles and triangles) in the Janauacá floodplain. The inset in (**d**) shows the model-data comparison at Open Lake, where model validation is based on the comparison with published mean and standard deviation values (error bars in (**d**) and (**e**))[55]. The black dashed line in both (**c**) and (**e**) represents the 1:1 line. The normalized root-mean-square error (NRMSE) is 0.04, 0.03, 0.29, and 0.21 for water level at WL_A and WL_B and for SSC at Upstream and Downstream sites, respectively. In (**b**), the bias in the simulated water level at WL_A during LW is due to the unresolved floodplain bathymetry at this specific location. The background maps in (**a**) are from Google Satellite Map (Map data © 2024 Google) and the Environmental Systems Research Institute (ESRI) National Geographic basemap. Source data are provided as a Source Data file.

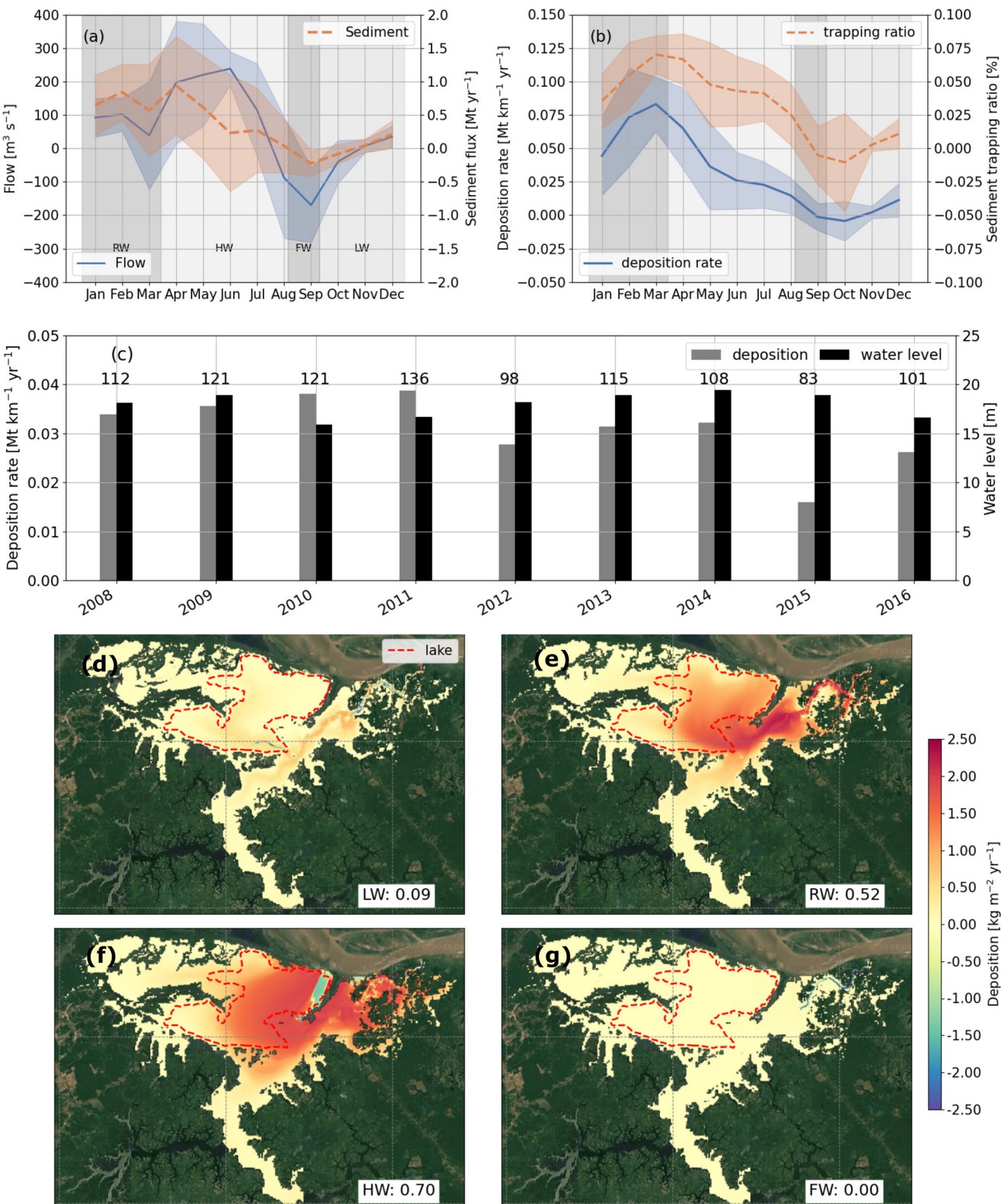

**Fig. 2 | Sediment dynamics in Janauacá. a** Daily averaged exchanging rate of monthly flow and suspended sediment through the open boundaries (transects in Fig. 1a). The positive and negative values represent the flux into and out of the Janauacá floodplain, respectively. **b** Daily averages of sediment deposition rate and sediment trapping ratio. The sediment trapping ratio is the ratio of the sediment deposition in the floodplain versus the sediment load of the Amazon River at the Manacapuru gauge. The negative value of deposition rate indicates bed erosion during FW and LW. **c** Annual mean sediment deposition rate of Janauacá, annual mean water level, and annual SSC of the Amazon River from 2008 to 2016. The annual SSC each year is the number over the bar. Corresponding scatter plots of annual mean deposition rate, water level, and SSC are provided in Supplementary Fig. S7. Maps of the modeled deposition rate averaged over four hydrological periods: **d** LW, **e** RW, **f** HW, and **g** FW. The number behind each hydrological period is the deposition rate. The background maps in (**d**–**g**) are from Google Satellite Map (Map data © 2024 Google). Source data are provided as a Source Data file.

the four hydrological periods, with rates ranging from less than $1 \, kg \, m^{-2} \, yr^{-1}$ in the areas with persistently fast flows to over $5 \, kg \, m^{-2} \, yr^{-1}$ in the deeper lake areas (Fig. 2d, g). These results fall within the reported range of 0.5 to $8.8 \, kg \, m^{-2} \, yr^{-1}$ for different Amazonian floodplains[6,34,56,57], as detailed in the Discussion section. Unlike the modeled SSC that reached a maximum during RW, the deposition rate reached elevated values of $>2 \, kg \, m^{-2} \, yr^{-1}$ in both the RW and HW periods (Fig. 2e, f). Between these two periods, the sediment deposition during HW is more widespread to the lake and wetland units in the floodplain (Fig. 2e, f), implying its higher relevance to the sediment and carbon budgets of these landscape components. Particularly, the model shows that the rising water supplied a dominant amount of sediments to the floodplain (Supplementary Fig. S4b). These intruded sediments were only deposited to the deep water area of Janauacá Lake during the subsequent HW, several months after RW (Fig. 2f), when the slowing of water movement favored sediment deposition (Supplementary Fig. S6c). In regions adjacent to WLBC3 where flows are persistently fast during RW and HW (Supplementary Figs. S6b and S6c), the deposition rates are reduced due to continuous sediment resuspension and transport (Fig. 2e, f). Overall, the annual sediment deposition can exceed $5 \, kg \, m^{-2} \, yr^{-1}$ in the deep water lake.

Despite its short length (about 5 km) along the Amazon River, the Janauacá floodplain alone can trap up to 0.09% of the Amazon River sediment (Fig. 2b). This high trapping ratio highlights the efficient retention of the sediment influx by the floodplain. Of the 0.34 Mt mean annual sediment influx through the river-floodplain boundary (Fig. 2a), ~11.7% was deposited in the floodplain (Supplementary Fig. S10) and the rest was returned to the Amazon River during FW (Fig. 2b). Like sediment deposition, the sediment trapping ratio of the floodplain also reached the peak values in the RW period (Fig. 2b).

Anthropogenic disturbances from river damming and land use change exert critical and nonlinear impacts on sediment deposition in Amazonian floodplains (Fig. 3, and see "Method" section for experiment design of the counterfactual scenarios). By reducing and increasing the SSC level in the Amazon River to respectively mimic the impacts from damming and deforestation, our model predicts that a reduction of SSC by reservoir trapping or a small increase of SSC by moderate deforestation in the Amazon would cause the same proportional changes (roughly linear increase) of sediment deposition in the Janauacá floodplain (Fig. 3b). In other words, under these perturbation levels, the sediment retention ratio in the floodplain does not change. However, if the SSC level in the Amazon River increases substantially due to deforestation, the gain of sediment deposition in the floodplain would gradually lose to the surge of river sediment influx over time (Fig. 3c and Supplementary Fig. S16), causing higher fractions of river sediment to transport downstream instead of being deposited in the floodplain. Our analysis shows that this reduction of sediment retention in the floodplain is a result of floodplain bed evolution (Supplementary Figs. S12–S14). Under the extreme scenario of agricultural expansion (60% increase in SSC), the trapping efficiency of the floodplain is predicted to decline by 7% by the tenth year of the simulation, as the floodplain elevation near its open boundary rises by more than 0.5 m (Supplementary Fig. S12). It is worth noting that the human-induced alteration of SSC levels will make little changes to the seasonal and inter-annual variabilities of sediment deposition and sediment exchange flux (Fig. 3a, b and Supplementary Fig. S11b), although their seasonal variations are intensified. The seasonal change of SSC implies a more dynamic response of the deposition rate, with shifts in SSC peaks causing notable increases in sediment influx during HW (Supplementary Fig. S15a, c, and e).

With sediment deposition, considerable biologically active POC from the Amazon River was also deposited in the floodplain. However, quantitatively, the deposited POC per year is markedly lower than the annual soil organic carbon input from ecosystem primary production (Fig. 4a, b). For the lake area where POC deposition is the strongest, the modeled annual mean POC deposition rate is only $0.064 \pm 0.013 \, g \, C \, m^{-2} \, day^{-1}$, which is one order of magnitude lower than the measured in-lake net primary production (NPP) in Janauacá and other Amazonian floodplain lakes, for instance $-5.7 – 29.2 \, g \, C \, m^{-2} \, day^{-1}$ in Janauacá Lake[33], $0.01 – 0.72 \, g \, C \, m^{-2} \, day^{-1}$ in the floodplain lakes of central Amazon[13], and $0.55 \, g \, C \, m^{-2} \, day^{-1}$ in the nearby Lake Calado[58] (Fig. 4a). Sediment deposition plays an even less important role in the carbon budget of the Janauacá wetland area (Fig. 4b). The modeled annual mean POC deposition rate is $0.012 \pm 0.002 \, g \, C \, m^{-2} \, day^{-1}$, which is two to three orders of magnitude smaller than the measured wetland NPP of $26 \pm 5 \, g \, C \, m^{-2} \, day^{-1}$[59] and $1.5 \, g \, C \, m^{-2} \, day^{-1}$ in the nearby, smaller Calado floodplain[60]. If considering the whole Janauacá floodplain, the estimated POC deposition rate is also considerably smaller than the derived NPP from the Moderate Resolution Imaging Spectroradiometer (MODIS) satellite ($4.4 \, g \, C \, m^{-2} \, day^{-1}$) and the Inter-Sectoral Impact Model Intercomparison Project (ISIMIP) multi-model ensemble ($2.5 \, g \, C \, m^{-2} \, day^{-1}$) (Fig. 4b; and see "Method" section).

With the measured oxygen levels and high-resolution water level and POC deposition simulations, we delineated a detailed map of oxygen exposure time (OET) of the deposited POC in the floodplain (Fig. 4c; and see "Method" section for the OET calculation). Within the lake zone (as defined in Fig. 1a), over 80% of the deposited POC has OET less than 10 days and over 90% has OET less than 1 month (Fig. 4d). The short OETs can be attributed to the concentration of POC deposition in deep lake waters (Fig. 2) as well as the shallow depths of oxycline in the Janauacá Lake[33]. Therefore, when sediment deposition occurs in the Janauacá Lake, the associated POC would likely experience an oxygen-deficient environment subsequently which favors carbon burial and anaerobic carbon oxidation, such as methanogenesis[61,62]. Conversely, outside of the lake zone, such as in wetlands and rivers, the deposited POC has much longer OET. For instance, over 60% of the deposited POC experiences longer than 100 days of OET in the non-lake area (Fig. 4d), an environment favoring aerobic carbon oxidation[62].

## Discussion

### Sediment deposition in Amazon/Solimões floodplains

The hydrological cycle of the Amazon River is a critical driver of sediment flux to Janauacá[40,63], but it is noteworthy that no significant correlation exists between the water stage of the Solimões River and sediment deposition in the floodplain at both annual and seasonal scales. This deviates from the observed sediment deposition behavior in the Amazonian floodplains near the Andes[14]. The contrasting results suggest that sediment deposition in these two floodplain systems is controlled by distinct drivers. For the Amazon/Solimões floodplains where fine sediments are dominant, our investigation finds that river SSC and floodplain hydrodynamics are more influential for sediment deposition. They also do not synchronize with the temporal and spatial variations of the water stage. Notably, floodplain hydrodynamics play a crucial role in spatial distribution of sediments, keeping them suspended in the Amazon/Solimões floodplains. As a result, only a small fraction of intruded sediments are deposited in the floodplains and the rest flow back to the Amazon River during FW. The hotspot area of sediment deposition in the floodplains during different hydrological periods correlates closely with the area where kinetic energy of inundated water is greatly attenuated. This result highlights that floodplain lakes are a major sink of water kinetic energy in the Amazon Basin and thus largely responsible for the trapping of ~11.7% of sediment influx from the Amazon River. A similar phenomenon has been observed in other Amazon Basin floodplains which possess numerous lake systems[64], which implies the broad applicability of the mechanism. Conversely, for the floodplains deficient in open waters, such as river deltas, most sediment would be deposited over the vegetated area[65,66]. Given the challenge to measure floodplain dynamics from space[32], our

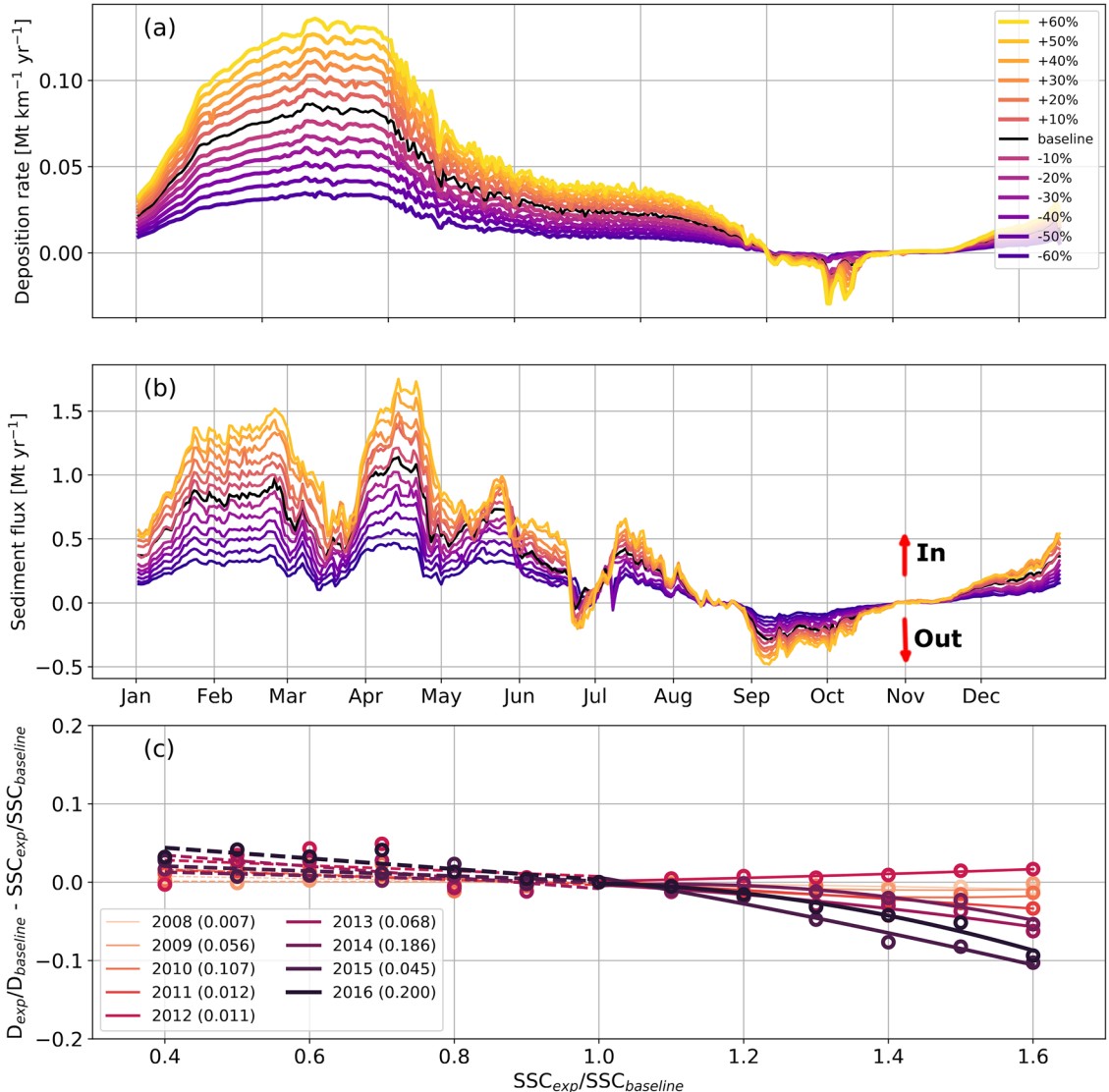

**Fig. 3 | Anthropogenic impacts on sediment deposition and flux.** Monthly variations of the simulated daily mean deposition rate (**a**) and daily mean suspended sediment flux through the open boundaries (**b**) on different SSC perturbation levels. **c** The change ratio of the simulated annual mean sediment deposition rate on different SSC perturbation levels for each simulation year. In (**a**) and (**b**), the baseline is the simulation without SSC perturbation and different percentages represent the SSC perturbation levels in the Amazon River. In (**c**), $D_{exp}$ is the simulated sediment deposition rate in the perturbed SSC simulations ($SSC_{exp}$) and $D_{baseline}$ is the simulated sediment deposition rate without SSC perturbation ($SSC_{baseline}$). For each year in (**c**), the simulations with reduced SSC are fitted using a linear relationship, and the simulations with increased SSC are fitted with second-order polynomial functions. The number after each labeled year is the coefficient of the second-order terms that measures the nonlinearity. Source data are provided as a Source Data file.

findings underscore the crucial value of high-fidelity hydrodynamic-sediment coupled models for the research of sediment deposition in Amazonian floodplains.

Human activities in the Amazon Basin continue to exert asymmetric impacts on floodplain sediment deposition. We show that the increase in river dams along the main course and tributaries of the Amazon River will reduce SSC in the downstream systems[22] and curtail sediment deposition in a proportional manner. The trapping efficiency of the floodplains for the Amazon River sediment therefore will not change. To the contrary, the impacts of expansion of cropland, rangeland, and deforestation in the Amazon Basin on sediment deposition are nonlinear. The surge of river SSC due to the dramatic increase of soil erosion (e.g., over 40%)[26] will overwhelm the capacity of the floodplains to trap sediment. This is because the progressive increase of floodplain bed elevation, especially in deeper water areas, would affect the magnitude and timing of flow and sediment exchanges, as well as the spatial distribution of deposition adjusted (Supplementary

Figs. S13 and S14), hindering a linear increase of deposition rates. With sediment deposition saturated, the trapping efficiency of Amazonian floodplains will decline sharply and higher proportions of sediment will remain in the Amazon River main channel. As deforestation increases sediment supply, the decreased floodplain trapping efficiency would increase the likelihood that the sediment transport capacity of the Amazon River is exceeded. Consequently, more sediments would be deposited in downstream river channels, exacerbating channel aggradation and reducing flood capacity[67,68]. River reaches may also undergo various geomorphological adjustments, including more rapid channel migration, channel steepening, or avulsion. These processes, which help rivers manage excessive sediment loads, can significantly alter floodplain morphology and hydrodynamics[69], but more research is needed to elucidate the impacts of such dynamics.

The modeled sediment deposition rate ($1.33 \pm 0.24$ kg m$^{-2}$ yr$^{-1}$) in the Janauacá floodplain is in line with the estimates for other Amazonian floodplains, including ~$0.5 \pm 0.1$ kg m$^{-2}$ yr$^{-1}$[56] at Lago Grande de

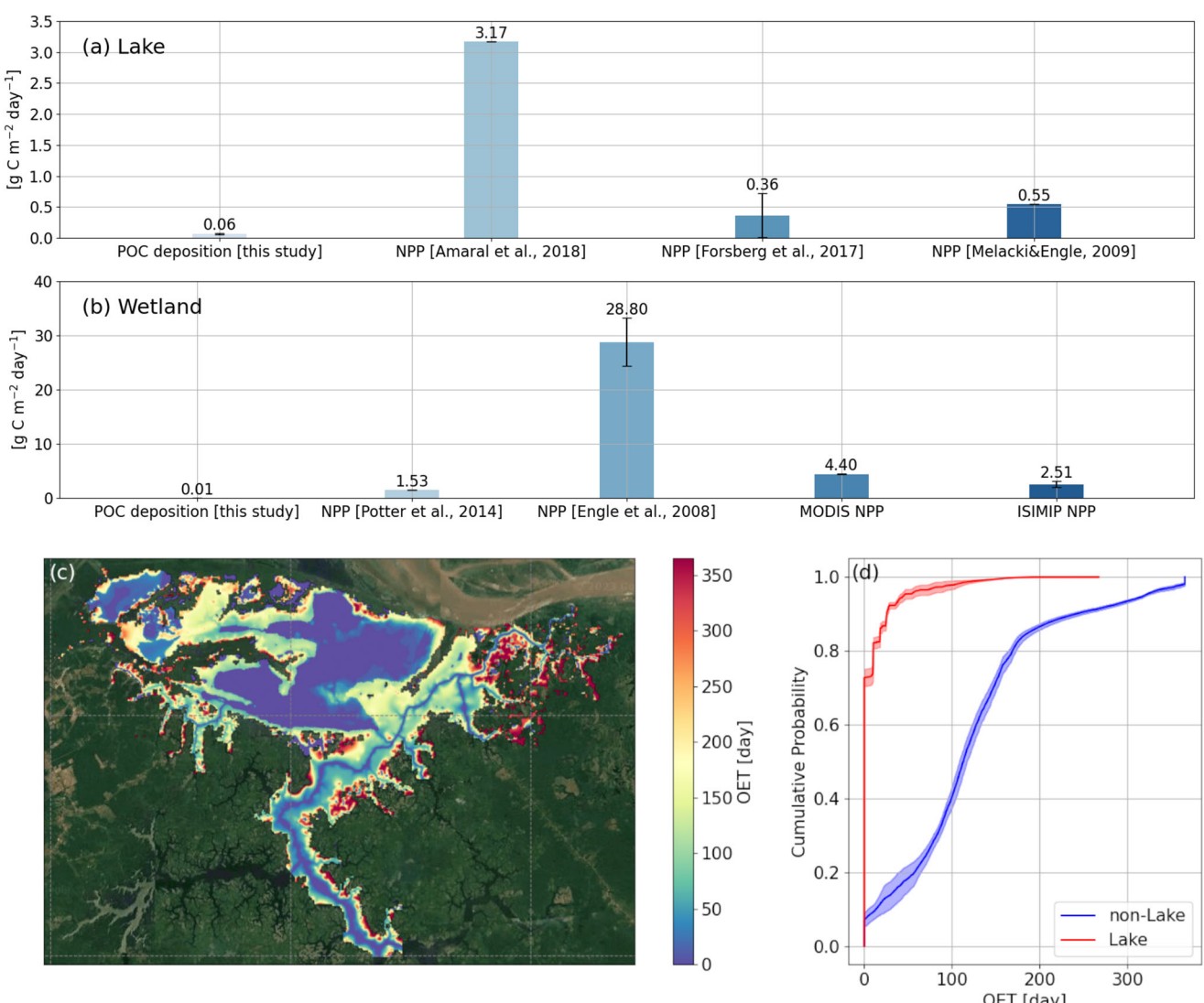

**Fig. 4 | Comparison of simulated POC deposition with NPP and oxygen exposure time (OET) in the Janauacá floodplain. a** Comparison of the simulated mean annual POC deposition rate with the measured NPP (i.e., Amaral et al., 2018[33], Forsberg et al., 2017[13] and Melack& Engle, 2009[58]) in the lake area of floodplains. Due to the large spread of NPP data in ref. 33, the mean value is shown here, above each bar. **b** Comparison of the simulated mean annual POC deposition rate with the measured (i.e., Potter et al., 2014[60] and Engle et al., 2008[59]) or derived NPP in the wetland area of floodplains. The error bars in (**a**) and (**b**) represent standard deviation. **c** Multi-year mean OET of deposited POC in Janauacá. **d** The cumulative distribution function of multi-year mean OET (weighted by the deposition rate) of grid cells in the lake area and non-lake area of Janauacá. The lake area and the wetland area are defined in Fig. 1a and Supplementary Fig. S1, respectively. The background maps in (**c**) are from Google Satellite Map (Map data © 2024 Google). Source data are provided as a Source Data file.

Curuai, 0.84 kg m$^{-2}$ yr$^{-1}$ at Lake Calado[34], 1.2 kg m$^{-2}$ yr$^{-1}$ at Lake Paca[57], and 8.8 kg m$^{-2}$ yr$^{-1}$ at the floodplains along the 390-km long reach between Itacoatiara and Óbidos[6]. If estimated by the floodplain length, the sediment deposition rate is 0.033 ± 0.006 Mt km$^{-1}$ yr$^{-1}$, which is significantly lower than a previous estimate of 0.3 Mt km$^{-1}$ yr$^{-1}$ in the reach between Itacoatiara and Óbidos[6]. However, it should be noted that both sides of the river banks were accounted for in their estimate[6].

By upscaling the sediment deposition rate to the floodplain area within the drainage basin of the Amazon/Solimões River mainstem (i.e., 58,103 km$^2$), we estimate that the Amazon/Solimões floodplains could trap a total amount of 77.3 ± 13.9 Mt of sediment per year and 0.98 ± 0.18 Mt of POC per year using the POC/sediment mass ratio of 1.27%[70]. Given 1200 Mt sediment drained to the Atlantic Ocean by the Amazon River per year[3], sediment deposition in the floodplains could account for a 6.1 ± 1% of sediment discharge by the Amazon River. Despite a small percentage of the total sediment load retained in the Amazon floodplains annually, this sediment can be subject to extensive biogeochemical processing due to prolonged floodplain storage, as indicated by the observed long transit times[71].

The estimated deposition is influenced by several uncertainties, such as the variability in floodplain trapping efficiency and the potential overestimation of the floodplain area[72]. Moreover, Janauacá may not represent all floodplains along the Solimões River, as there are sites with varying ratios of local drainage extent to lake area[73,74]. Our upscaling estimation focuses on the mainstem of the Solimões/Amazon River but excludes all tributaries, assuming that Janauacá is only representative of floodplains adjacent to the mainstem. This figure likely underestimates the total sediment deposition in the basin, because the floodplains of the tributaries could also trap a substantial amount of sediment[75]. For comparison, using the simulated river-length-specific deposition rate of 0.033 Mt km$^{-1}$ yr$^{-1}$ and the total mainstem length of 3349 km for the Solimões/Amazon River from HydroRIVERS[76], an alternative river-length-based upscaling approach yields a much higher total deposition estimate of 211 Mt per year.

However, this figure is likely an overestimate because floodplains are not always present along the river and on both sides.

## Implications for carbon cycling

Amazonian floodplains play an important role in the global $CO_2$ and $CH_4$ cycles[77]. Due to the enormous amount of sediment deposition in the floodplains, it is reasonable to speculate that the deposition of sediment-associated POC could be important for the regional $CO_2$ and $CH_4$ emissions[78,79]. However, our results do not support this hypothesis. Compared to the wetland soil and lake sediment carbon that are sourced from ecosystem primary production, the soil/sediment carbon input due to POC deposition is considerably smaller. Therefore, POC deposition in Amazon floodplains is unlikely to contribute to the high $CO_2$ and $CH_4$ emissions observed in the region, despite its crucial role in the supply of critical nutrients to the floodplain soils[80]. Furthermore, because Amazonian floodplains are one of several sediment deposition hotspots in the world, we suspect that POC deposition also only plays a minor role in fueling $CO_2$ and $CH_4$ emissions in other productive floodplains across the globe. However, for low-productivity floodplains in high latitudes, fluvial POC deposition remains important for regional $CO_2$ and $CH_4$ emissions[81].

Although sediment deposition in Amazonian floodplains is not an important driver of regional $CH_4$ emissions, due to its strong spatial heterogeneity and seasonal variations, this process can influence $CH_4$ emissions under specific conditions (i.e., deep lake areas during HW) within each floodplain. For instance, we find that the POC deposition rate in the lake area is approximately five times higher than that in the wetland area. For the deepest lake area, the POC deposition rate during HW can reach $0.57\,\mathrm{g\,C\,m^{-2}\,day^{-1}}$, which becomes comparable to the measured in-lake NPP. With limited oxygen exposure, a substantial fraction of these biologically active POC could be oxidized to $CH_4$ during the HW period through methanogenesis[61,62]. However, since deep lake areas only comprise a small fraction of the entire floodplains and the HW period only lasts 2–3 months, the overall contribution of POC deposition to regional annual $CH_4$ emissions is minor.

## Methods

### Hydrodynamic-sediment coupled model

A numerical model is developed based on the TELEMAC-MASCARET modeling system that couples hydrodynamics and suspended sediment transport processes. TELEMAC-MASCARET is an open-sourced integrated suite of solvers for use in the field of free-surface flow[82]. The system tightly couples a shallow water module (TELEMAC-2D) and a new sediment transport module (GAIA)[83] at every time step (i.e., 30 s). TELEMAC-2D solves 2-dimensional (2-D) shallow water equations (SWE) and advection diffusion equations (ADE) for suspended sediment transport. GAIA computes the bed evolution and the erosion ($E$) and deposition ($D$) fluxes ($\mathrm{kg\,s^{-1}\,m^{-2}}$) following[84,85]:

$$E = M\left(\frac{\tau_b - \tau_{ce}}{\tau_{ce}}\right), \tag{1}$$

$$D = w_s C_b \left[1 - \left(\frac{\tau_b}{\tau_{cd}}\right)\right], \tag{2}$$

where $M$ is the erodibility coefficient or Krone-Partheniades erosion law constant ($\mathrm{kg\,s^{-1}\,m^{-2}}$), $\tau_{ce}$ and $\tau_{cd}$ are the critical shear stress for erosion and deposition ($\mathrm{N\,m^{-2}}$), repsectively. $\tau_b = 0.5\rho C_f(U^2 + V^2)$ is the bottom shear stress ($\mathrm{N\,m^{-2}}$), where $C_f$ is the friction coefficient, and U and V are velocities in the Cartesian system. $w_s$ is the settling velocity of the sediment particles and $C_b$ is defined as the sediment concentration at the interface between suspended load and bedload, which is the SSC in our case since bedload is not considered. Both $E$ and $D$ contribute to the change of bed elevation.

The hydrodynamic component of the coupled model was previously configured by Pinel[32] based on the 30-m resolution digital elevation model (DEM) derived from Shuttle Radar Topographic Mission (SRTM)[50]. The bias over permanently inundated areas in the SRTM DEM was addressed by using the elevation data from the NASA's Ice, Cloud, and Land Elevation Satellite (ICEsat) Geoscience Laser Altimeter System (GLAS). The SRTM correction also used bathymetric data from high water levels in June 2012, collected with an ADCP and GPS-linked echo sounder, along with sampling from May 2008 and August 2006. Meanwhile, vegetation-induced biases were corrected using a wetland map from ref. 7 and a MODIS-derived vegetation height dataset from ref. 86. These corrected ground elevations were integrated using the ANUDEM v5.3 algorithm, constrained by a drainage channel network identified from Landsat imagery. For a detailed methodology, see ref. 50. The computational domain of Janauacá is delineated as the floodable region with an elevation lower than 29 m (4.6 m above the observed maximum water level)[32]. The mesh was developed with a horizontal resolution of 23–77 m[32]. The domain includes 8 inflow boundaries (BCF) and 3 water level boundaries (WLBC), the latter of which connect to the mainstem of the Amazon River (Fig. 1a). The delineation of WLBCs is based on ALOS-1/PALSAR images. The model domain is classified into various topological regions based on the dual-season wetlands map[7] (Supplementary Fig. S1), where the corresponding Manning's roughness coefficients are defined following[87] and are calibrated to ensure minimum bias in the hydrodynamic simulation[32]. The upland discharge and water level boundary data are obtained from a hydrologic model[51] and the observation gauge at Manacapuru (60°33′13.8″, 03°18′51.0″) from the ORE-HYBAM database[4], (https://hybam.obs-mip.fr/), respectively. The water level at the WLBCs is bias-corrected, using the difference in bed elevation from the Manacapuru gauge. Wind was excluded from the simulation due to the unavailability of vegetative sub-canopy wind, which is essential for accurate modeling in this context. This hydrodynamic model, as calibrated and validated against observations of water level, velocity, and flood extent over multiple hydrological years, provides a prerequisite for coupling with a sediment transport model that preserves the full dynamics.

Our simulation considers depth-integrated SSC, as the water depth in the Janauacá floodplain is typically much shallower than that in the Amazon River mainstem. Even in the deeper lake areas, peak water depths reach about 10 m during HW and reduce to less than 5 m during LW (Fig. S3), with SSC showing minimal variation across the shallower water column[40,88]. The daily SSC at the water level boundaries (Supplementary Fig. S2a) was derived from the near-surface observation at Manacapuru from the ORE-HYBAM database[4], with sample gaps filled using satellite-derived estimates following[40]. Outliers that are 4 standard deviations away from the mean were removed from the satellite-derived SSC estimates. A Lowess filter was applied with specified parameters (fraction=0.02, iterations=0, degree of polynomial=1) to ensure a smoothed SSC boundary condition and eliminate unrealistic variations that may cause numerical instability issues. There are no significant tributaries between the Manacapuru gauge and the WLBCs that would alter the sediment load along this stretch of the river. The satellite data[39] also implies no significant difference between SSC at Manacapuru and at WLBCs (Supplementary Fig. S18). However, due to the lack of direct SSC measurements at WLBCs, there is an inherent uncertainty in the SSC boundary condition. While there are no SSC samples collected from the upstream boundaries, the lake's upland catchment drains black water streams with low SSC ($<20\,\mathrm{mg\,L^{-1}}$)[33]. Thus, the SSC values are assumed as $20\,\mathrm{mg\,L^{-1}}$ at the BCF1 and BCF8 and $40\,\mathrm{mg\,L^{-1}}$ at the rest BCFs. The water level data is used to define 4 hydrological periods of low water (LW), rising water (RW), high water (HW), and falling water (FW): LW is when water level <15 m, HW is when water level >20 m, and RW and

FW are the periods with water level between 15 m and 20 m (Supplementary Fig. S2b).

In-situ samples collected at Manacapuru along the Solimẽs River[88] and in a similar floodplain, i.e., Lake Grande de Curuai[89], imply that the composition of suspended sediment is dominated by silt and clay ($< 63\,\mu m$), with the median particle diameter (D50) of $12 \sim 49\,\mu m$. Thus, we define 3 cohesive sediment classes in GAIA. The first two classes correspond to clay or fine silt from the Amazon/Solimões River, and the third class is sourced from the upstream catchment. We use the default sediment density of $2650\,\mathrm{kg\,m^{-3}}$ for all classes. Please note that since significant flocculation only occurs where the Amazon sediments meet the fine sediments from brackish-water tributaries[27], our model does not account for flocculation processes, which are observed to enhance the settling velocity of fine sediments in many river systems[90].

GAIA allows the change of lake morphology from sedimentation processes. The cumulative bed evolution is computed internally in the model computation using the deposition obtained in Equation (2) and a default bottom layer density of $50\,\mathrm{kg\,m^{-3}}$. Since the model does not output deposition rate directly, we estimated the deposition per unit area from the computed bed evolution. The deposition rate per unit length is the total deposition of Janauacá over the length of its open boundaries connecting to the Amazon River (~5 km). The sediment trapping ratio is defined as the ratio of the Amazon River sediment trapped by the floodplain, which is calculated by dividing the deposited sediment in the floodplain by the river sediment load measured at Manacapuru.

The modeled hydrodynamics in the Janauacá floodplain has been previously validated in ref. 50. This study re-validates the model against the water level observation at two gauges. The validation of modeled sediment dynamics in Amazonian floodplains remains challenging due to the dearth of both SSC and sediment deposition data. Here, the modeled sediment dynamics are validated using the SSC in-situ samples collected at three sites, which are all 0.5 m below the water surface, within Janauacá (Fig. 1a)[33,54,55], and the measured sediment deposition rate in floodplains with similar geographic settings[6,34,89], as there is no direct measurement of sediment deposition in the local region. At the downstream gauge, the Amazon River dominates the SSC variation, whereas in the open lake and upstream regions, wind effects may lead to sediment resuspension during low-flow seasons. The coupled hydrodynamic and sediment transport simulation is performed from 2007 to 2016. The first year is considered as model spin-up and is excluded from the analysis. To upscale the modeled deposition, the total floodplain area within the Amazon/Solimões River mainstem drainage basin is computed using the flooded wetland extent during high flood conditions[7,91] with the basin boundary delineated in ref. 92. The amount of sediment deposition in the floodplains within the drainage basin of Amazon/Solimões River mainstem can be estimated using the total floodplain area and the modeled deposition rate per unit area.

## Anthropogenic disturbance

To quantitatively assess the impact of anthropogenic disturbance from dam construction[19–21] and land use change[23,26] on sediment deposition, we performed multiple numerical experiments by perturbing the SSC value at the WLBCs. In general, dams trap large amounts of sediment, resulting in decreased SSC in the downstream reaches[30,93,94,95] reported a 20–30% reduction of SSC in the Madeira River downstream of the Santo Antônio Dam.[22] implies that the dams, built or proposed in the central Amazon Basin present a trapping efficiency of 0–60%. Following their estimate, the potential impact of dam construction is quantified by experiments with the boundary SSC reduced by 10%, 20%, 30%, 40%, 50% and 60%. Furthermore, excessive agricultural expansion and livestock activities cause deforestation in the Amazon Basin. The resultant land use and land cover change intensifies soil losses from erosion, increasing downstream SSC[23]. The impact of

deforestation is estimated by increasing the boundary SSC by 10–60% based on the increased soil erosion and sediment delivery rate in Solimões River[26]. While it is not entirely clear how anthropogenic disturbances affect the seasonal variations in SSC, these potential impacts were assessed through experiments, including reducing variation amplitude by 60% and enhancing the SSC magnitude by 60% during peak periods. Although the timing of the SSC peak is influenced by climatic rather than anthropogenic forcing, supplementary experiments were conducted by advancing the SSC peaks 30, 60, and 90 days towards the water level peak to assess how interactions between climate and human activities might affect the sediment deposition in Amazonian floodplains.

## Carbon dynamics

The impact of sediment deposition on carbon dynamics of Amazonian floodplains is assessed through the estimation of particulate organic carbon (POC) deposition and oxygen exposure time (OET) of deposited POC. While the present TELEMAC-GAIA model does not directly simulate POC deposition, it can be reasonably estimated using simulated sediment deposition rate and measured POC concentration in suspended sediment of the Amazon River.

$$D_{POC} = D \times f_{POC}, \qquad (3)$$

where $f_{POC}$ is the POC concentration measured from suspended load samples collected from channel depth profiles at Manacapuru during two sampling campaigns in June 2005 and March 2006 that respectively correspond to HW and RW periods[70]. The value is $1.27\% \pm 0.21\%$. It should be noted that due to the dependence of POC adherence on sediment grain size[96], the use of a unified POC ratio across all sediment classes carries substantial uncertainty.

OET is the major driver of POC burial efficiency and its mineralization pathways. High OET corresponds to fast mineralization of deposited POC and strong $CO_2$ production through aerobic oxidation. To the contrary, low OET corresponds to slow mineralization of deposited POC and strong $CH_4$ production through anaerobic oxidation. For deposited POC, OET is estimated using the modeled water depth and measured depth-resolved dissolved oxygen (DO) levels following the steps below. First, we identify the oxycline−the depth below which DO concentrations fall below $2\,\mathrm{mg\,L^{-1}}$−using DO data collected during the four hydrological periods by Amaral[33] at the wind-exposed (WE) site of Janauacá. The estimated oxycline depths are 0.5 m, 2 m, 3.5 m, 1.2 m for LW, RW, HW, and FW, respectively. We assume that the measured DO is uniformly distributed horizontally across the WE site and the identified depths separate aerobic and anaerobic sediment. Second, we compare the modeled water depth at each grid cell to the estimated low DO depths and define OET of each grid cell as the amount of days in a year that the sediment is above the low DO depths. The estimated OET is calculated based solely on the pre-burial oxidation condition for the settled POC and does not account for post-deposition oxidation processes that can deplete oxygen in the sediment layers. As a result, our estimation tends to be conservative.

For Amazonian floodplains, sediment/soil OC has two sources: autochthonous OC from primary production within the floodplain and allochthonous OC from POC deposition. We estimate autochthonous OC input in the floodplain from published net primary production (NPP) data for the lake area (Fig. 1a) and wetland area (non-lake flooded area) of Janauacá. The wetland area is covered by macrophytes and flooded forest (Supplementary Fig. S1). For the wetland area, the NPP estimates are available from in-situ data at Janauacá[33], from a comparable and smaller floodplain[58–60], satellite-derived MODIS product[97] and ISIMIP multimodel ensemble[98]. It should be noted that due to their coarse resolutions, the MODIS and ISIMIP estimates would include primary production from upland forests. We used the NPP of CARAIB,

DLEM, JULES-B1, LPJmL, ORCHIDEE, VEGAS, and VISIT from ISIMIP2a simulations that are forced by the GSWP3 climate data. The ISIMIP model outputs have the resolution of 0.5° and are archived monthly from 1971 to 2010. The annual NPP at the grid nearest to our domain since 2000 is used for comparison. The monthly NPP derived from MODIS from 2008 to 2016, with the spatial resolution of 500 m, is averaged annually over the model's computational domain. For the lake area, the NPP estimates are available from local measurements[33] and similar floodplain lakes[13,34,58].

## Data availability

Source data are provided with this paper. The baseline simulation results have been deposited in Zenodo at https://doi.org/10.5281/zenodo.14855431. Due to size constraints, additional simulation data are available upon request. Data obtained from third parties and used in figures include satellite imagery from Google Static Maps API (Google Earth, https://www.google.com/maps) and the Environmental Systems Research Institute (ESRI) National Geographic basemap (https://www.esri.com/en-us/arcgis/products/arcgis-location-platform/services/basemaps). These datasets are available under restricted access for third party rights. The access can be obtained by either perceiving proper permission from the data providers or providing appropriate credit. Source data are provided with this paper.

## Code availability

The Telemac source code is accessible at https://gitlab.pam-retd.fr/otm/telemac-mascaret/-/releases/v8p3r1(last access: Feb 2025). The scripts used to process and analyze the simulation results have been deposited in Zenodo at https://doi.org/10.5281/zenodo.14855431.

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

## Acknowledgements

This work was supported by the Scientific Discovery through Advanced Computing 5 (Capturing the Dynamics of Compound Flooding in E3SM), funded by the U.S. Department of Energy, Office of Science, Office of Biological and Environmental Research. The Pacific Northwest National Laboratory is operated by Battelle for the U.S. Department of Energy under Contract DE-AC05-76RLO1830. João Henrique Fernandes Amaral acknowledges the support of Conselho Nacional de Pesquisa e Desenvolvimento—Ministério da Ciência Tecnologia (CNPq/MCTI); CNPq/LBA-Edital.68/2013, processo, 458036/2013-7, CNPq-Universal processo.482004/2012-6.

## Author contributions

D.F. and Z.T. designed the study, performed hydrodynamic-sediment modeling, conducted the analyses, and wrote the initial manuscript. S.P. and M.B. set up the hydrodynamic model. D.X. and G.B. investigated the results. J.A. and A.F. processed the sediment data. All authors participated in technical discussions and manuscript editing.

## Competing interests

The authors declare no competing interests.
