## [Peer Review file · Nature Communications]

Drivers and impacts of sediment deposition in Amazonian floodplains

Corresponding Author: Dr Dongyu Feng

Version 0:

Reviewer comments:

Reviewer #1

(Remarks to the Author)

This is a review of "Drivers and impacts of sediment deposition in Amazonian floodplains" by Feng et al. for Nature Communications (Manuscript#: NCOMMS-24-40333-T).

The authors use a hydrodynamic and sediment transport model to simulate sediment deposition on an Amazonian River floodplain. The methodology is generally sound. One of the key insights and impacts of this work is to inform decision makers about how floodplain sedimentation functions along the Amazon so that decisions about whether to construct dams or deforest are informed about the sedimentological consequences. Overall the manuscript was well written with good quality figures. I only have three comments:

The authors scale up their deposition rate to a stretch of the Solimoes/Amazon River. However, there is probably still a lot of deposition and exchange happening in smaller-order rivers. Can the authors further contextualize their results across all rivers in the Amazon basin or at least comment on how the estimate they are providing does not include other rivers and is thus an underestimate of the true amount of floodplain sediment deposition in this basin. This is an important point, because the deposition amount is not just 6% of the Amazon sediment flux, it is much higher if you account for the deposition happening in other floodplains.

The SRTM DEM using for model topography would have approximated a water surface to the floodplain lake with no bathymetry. Did the authors add bathymetry to the lake somehow (not mentioned)? And if not, you should discuss how this might affect your model results? That is, particularly with unsteady hydrodynamics, the amount of water stored in the floodplain lake would not be properly accounted for.

w_s and C_b in Equation 2 were not defined

Jonathan A. Czuba, Virginia Tech

Reviewer #2

(Remarks to the Author)

Review of "Drivers and impacts of sediment deposition in Amazonian floodplains" for consideration at Nature Communications

Thank you for the opportunity to review this paper, which describes a modeling effort to determine sedimentation in a floodplain on the Amazon River, Brazil. Overall, the paper is easy to follow and describes an interesting study on floodplain of a globally important river. The study seems interesting and is of importance to quantifying and constraining Amazonian sediment dynamics, but I do not believe it currently merits publishing in Nature Communications.

A primary limitation is comparison to field observations. With so many knobs to twist in hydrodynamic and sediment-transport models, getting a model to match a few sparse field measurements of SSC is not sufficient evidence of model performance, in my view. Limited information is provided regarding things like settling velocity, erosion parameters, and so on, and these have great influence on the modeled sedimentation patterns. Nearly all the sediment validation data are from

the mainstem of the river, and it is unclear to me where the lake SSC data are sourced from. There are also no field deposition results, which is a primary missing link in model confidence.

Much of the results and discussion are given over to describing how sedimentation is not related to water level and is instead related to SSC or sediment flux. This seems quite expected, and is one of the reasons I doubt the appropriateness of this study, in its present form, to Nature Communications. Much discussion is given to the hysteresis between SSC and discharge, but this is a common occurrence in fluvial systems and shouldn't really be a surprise, and at the least shouldn't be a main takeaway from the paper. What is new that is being presented here?

The treatment of the potential future changes in SSC (Fig 4) also seems limited. Rather than an increase/decrease in SSC, wouldn't it be more useful and likely to consider a change that varies with? For example, smaller peaks in SSC (a smoothing-out of the signal), or alternatively, larger peaks (an intensification of sediment release during the wet season), or a shift in the timing of the SSC peak relative to the water peak.

Overall, I find the results to be interesting but not appropriate for this journal. In addition to the concerns above, the results are not sufficiently tied into a larger contextual framework, or when they are, this feels unsatisfying. For example, burial of sediment-associated POC is shown to be a very small player in the overall budget. But then the claim is made that this is evidence for why POC needs to be accounted for in lake CH₄ models... I don't follow this argument.

Finally, although perfectly readable as-is, editing for English usage by a sharp-eyed editor would be improve the manuscript.

I have other concerns listed below in the page-by-page and figure comments that follow. (As an aside, line numbers would have been very useful in the original submission.)

Page 1

Check capitalization of Amazon River. Overall, English language editing would be useful throughout the paper.

Page 2

15% of river sediment: this number appears to be roughly double what you estimate in the results. Can you describe why the numbers don't match?

Page 4

Please add a figure reference after each statement or claim, rather than grouping reference to three figures together at the end of the paragraph.

Page 5

This is intuitive, although I could not find a figure showing the main-stem sediment flux ($Q \cdot SSC$) anywhere.

Figure 1

I would like to see a clearer comparison of modeled vs observed SSC, and not just a timeseries as currently provided in (b). It is very hard to see any dynamics in the timeseries with the different scales. Why is the open lake in-situ timeseries not in the same graph as the other sites? The inset in (b) also should be a scatter plot and not a simplified timeseries, which may mask important variability that should be discussed. Why do you not show the NRMSE for the open lake SSC in the caption?

Page 6

The correlations (-0.35, 0.94, etc.), or plots where one could interpret them, are not shown anywhere in figure 2, despite a reference to this figure.

Here you say SSC "topped" (reached a maximum?) during RW but on page 4 you stated it peaked during LW.

"more widespread" is this simply because the lake is deeper at HW?

I only see a maximum value of about 0.07% for the trapping ratio.

I do not understand the sentence beginning with "For 0.34 Mt yr⁻¹ ..."

Figure 2

Using bar charts (in (c)) that do not start the y-axis at zero is misleading and should not be used. If you keep a bar chart in this figure you must start them at zero.

It would be very useful to print the total deposition (on the figure itself, as text) for each period in subplot d, e, f, g to ease comparison.

Page 8

Although the increase/decrease in SSC does drive a change in deposition, this is not a strongly nonlinear response.

The linear increase/decrease in deposition in response to linear changes in SSC is intuitive, yes? If deposition is thought of as a response to the product of SSC and inundation time, and SSC changes, then so does deposition. The sentence

beginning “Interestingly” makes it seem like a surprise.

More detail regarding this apparent decrease in deposition is required. Where is the erosion occurring, and is it physically realistic? Could this just be a numerical artifact? Also, don't these scenarios leave out the possibility that the river channel itself will change, potentially altering the dynamics of the sediment movement between mainstem and floodplain in an unexpected way?

Page 9

Given that sediment deposition plays a negligible role in the carbon budget of this lake, how relevant is it to this paper, unless connected to a larger conceptual framework or understanding of carbon budgets in Amazon floodplains?

Figure 3

Help the reader understand the timing by showing the average river sediment flux to compare with the deposition rates plotted in (a)

Page 11

Rather than referring to their estimate of 0.3 Mt km⁻¹ yr⁻¹, why not normalize it to a single side of the floodplain from the beginning?

“The magnitude of”... I do not understand the point of this sentence. It seems to describe a two order of magnitude variability in sediment deposition on river floodplains. It is not surprising this should vary given the likely wide diversity of floodplain environments in the different studies, to say nothing of the drainage basins for each of these river systems.

Figure 4:

It is hard to know which one [32] refers to in the legend, since they are listed as Author, Year in the plot labels. Make it easier for the reader to interpret either by reference to color, name, or something else. Also, add (this study) to the label for your results. In (c), tell us what OET means [You define it elsewhere in the paper, but the captions should stand on their own].

Page 12

What is the meaning of the +/- ranges in these measurements? That is, what components go into determining the range? I did not see this in the Methods.

How does this estimate square with the comparison made on page 11: 0.03 vs 0.3 Mt/km/yr? If that higher estimate were applied with an average floodplain width, would it result in ~60% of the sediment discharge?

“it is surprising to note” is this actually surprising? Given the 2-4x change in yearly river discharge but approximately order of magnitude change in SSC, the sediment flux is an obvious parameter to compare against.

“Notably, floodplain hydrodynamics play a crucial role in distributing sediment in Amazonian floodplains” How else would sediment be distributed across the floodplain?

“could trap a dominant fraction” this sentence needs to be more quantitative to be meaningful. Same with the following sentence. Right now it reads as a statement about how floodplains slow the speed of water flow compared to the mainstem. That seems rather obvious, yes? But then at the end of page 12 it seems like you may be comparing unvegetated to vegetated floodplains? I am somewhat confused by the argument here.

Page 13

It is not clear that the lack of floodplains will result in river channel aggradation.

A clarification of the definition of lake vs wetland areas is needed, as it has not really been introduced prior to here. Even though it is briefly mentioned in the methods, to a reader starting from the beginning they will not be able to interpret the difference between the two.

Page 14

Given how small the POC deposition is, wouldn't your study actually suggest that it's not important to include in lake CH₄ models?

Page 15

How can water depth be zero when the minimum flooded area is 23 km²?

I read through reference [32] and its supplementary material but could not find any TSS/SSC data, even though “TSS” was mentioned several times in that article.

How were E and D determined, and what are their values? Are they physically reasonable?

Page 16

SRTM data would not provide information in flooded areas of the lake. How were water-covered regions addressed in the bathymetry used to create the model domain?

You mention wind being important to SSC at low water. From what I can tell by looking at [45], wind is not implemented in the model being used here. Is that correct? If not, this seems like it could be an important missing component of the model.

Clarify: the WLBCs do not connect to the mainstem

Page 17

It is also possible that the SSC at Manacapuru is not the same as the SSC at the WLBCs. Can you constrain the uncertainty here?

Figure S1, S2

Why is the extent of open lake in S1 different from the land cover type in S2?

Figure S3

Water level is okay in (b), but water discharge (and also a plot of sediment flux) would be a lot more relevant here.

Figure S6

It would be very useful to have the conditions at Manacapuru on these figures since the main-stem Amazon is so important to the dynamics within the lake.

Figures in general

I was not able to find the location of Manacapuru on any of the figures. Please add.

Reviewer #3

(Remarks to the Author)

Reviewer comments for "Drivers and impacts of sediment deposition in Amazonian floodplains"

This manuscript uses a numerical model that couples 2D shallow-water hydrodynamics with cohesive sediment entrainment and deposition to predict how much sediment is deposited on the floodplain of the Amazon River near Lake Janauacá. The results indicate that most suspended sediment comes from the Amazon River and flows through a secondary channel before being deposited in the lake, and that there is a 4-month delay between peak floods in the Amazon and peak sediment deposition. In addition, the results indicate that a small fraction of particulate OC is deposited as part of SSC, but it is unclear if this has a significant effect on regional biogeochemical cycling since NPP by algae in the lake greatly outpaces POC deposition.

Overall, I found the model results interesting, but I did not think the manuscript described the modeling in sufficient detail to fully interpret the results. This study builds on significant previous work at the site, but the manuscript does not stand on its own as written. The introduction did not lay out clear knowledge gaps or testable hypotheses for the study, so it is not clear to me how significant or transferable the results are to other floodplains. Predicting fine-scale patterns of sedimentary deposition is at the frontier of modeling for river floodplains, but the paper requires significant rewriting to make this study's contribution and broader impacts clear. Given these considerations, I recommend this manuscript for major revisions followed by re-review.

General comments:

1) The paper needs significant revisions to the introduction and results so that the reader can fully understand the study. Currently, the motivations for the study, particularly the connection between sediment transport and floodplain outgassing, are not thoroughly described. It is also not clear why the authors selected this site as representative of the Amazon floodplain or how the numerical model works. The journal format has the methods presented at the end, so it is important to help the reader understand the results without reading the methods or referring to the supplement. In particular, this study builds on multiple prior studies on this site from the same research group, which should be summarized in the introduction to instill confidence in the field measurements and modeling presented in this paper and allow it to stand alone.

2) The model produced an interesting result, where increasing SSC 40% above the base case caused additional floodplain incision and decreased sediment storage. This is a surprising result, and it would be worth digging into the mechanics of processes that define this threshold value to understand if this result can be generalized beyond the study site. This requires a more detailed discussion of spatial and temporal variability in floodplain sediment entrainment and deposition rates as well as some sensitivity or stability analysis to ensure the model is making reasonable predictions.

Specific comments:

Abstract: "Biogeochemical cycling" is mentioned but the abstract needs more of your quantitative results for these processes. Please clarify the relevant state of the fluxes (particulate, dissolved) and the spatiotemporal scales (regional? 100 years?) presented in the results.

Page 2, "fueling emissions of carbon dioxide (CO₂) and methane (CH₄) [12, 13].": The introduction needs to include more

prior work on floodplain outgassing and clarify that you focus on particulate OC. Rivers, wetlands, and dry floodplain soils can all release CO₂ and CH₄ but this can come from oxidation of POC or DOC. For instance, I would expect DOC to be more readily emitted as CO₂ or CH₄ from rivers and wetlands while POC should track with fine sediment deposition and is more likely to be sequestered in floodplains and avoid oxidation.

Page 3, “accurate estimation of sediment dynamics in Amazonian floodplains with sufficient spatial and temporal details still remains challenging.”: It is not clear to me what spatial and temporal resolution you require to understand biogeochemical cycling and what specific knowledge gap this study hopes to fill.

Page 4, “we used a recently developed and well validated model”: It would be very helpful to give more information about the model you used and where it has been validated. Stating here that the model was already validating for hydraulics at this field site is important since the reader does not get that information until the Methods.

Page 4, “the simulated water level in the floodplain follows four hydrological periods”: I was quite confused by the extent of the floodplain boundary versus the model domain, which should be defined and consistent with maps in the supplement.

Page 6, “the mean annual deposition rate per unit length of Amazon River, of 0.033 Mt km⁻¹ yr⁻¹”: What is the reference for this value?

Page 6, “years of extremely low water levels in 2010 and 2011”: It is unclear why the years evaluated in this paper were chosen. Is this limited by SSC data availability? Are these years anomalous due to storms or climate cyclicity? If satellite measurements are available that match field sampling well, why not examine sediment transport over a longer period of time to have more statistically robust results?

Page 6, “the deposition rate reached elevated values”: To get from sediment concentration to sediment deposition rates in a specific you multiply the SSC by an average settling velocity. Grain size and SSC both vary with depth in the channel, so how does your model account for this? Does the SSC from your datasets represent a surface value or do you do some normalization based on flow depth and shear stress?

Page 6, “deep water area of Janauaca Lake”: What are the constraints on lake bathymetry? The model appears to assume it is constant.

Page 6, “For 0.34 Mt yr⁻¹ of the mean annual sediment influx (Figure 2a), 11.7±7.2% was retained”: Where does the rest of the sediment go? Does it bypass the floodplain and return to the Amazon River or does it flow upstream on tributaries?

Page 7: I would appreciate a paragraph that includes discussion of the spatial and temporal patterns in SSC and deposition and discusses causes of their variability in detail. I feel like the results gloss over this and just discuss parameters calculated for the whole floodplain, like the sediment trapping ratio.

Page 8, “except in the case of elevating SSC by 60% in which excessive erosion would occur during HW”: I don't understand why elevating SSC would increase erosion. If input SSC is higher wouldn't that cause increased deposition? This is an interesting and non-intuitive result that deserves more discussion so the reader can understand this effect.

Page 9, “measured in-lake net planktonic primary production (NPP)”: It is worth separating out different types of carbon. SSC can contain petrogenic OC, which is more recalcitrant and could be important for long-term OC storage in floodplain deposits, while the biospheric OC produced by NPP could be more labile.

Top of page 10: This paragraph should be motivated in the introduction. Stating that you want to determine if sediment POC deposition matters in floodplains and particularly in lakes would help to focus the introductory paragraphs.

Page 10, “POC deposition at the Januaca floodplain is also considerably smaller than the derived NPP”: This paragraph makes it sound like the POC deposition flux doesn't matter. It's important to be very clear if it does, and if so, over what time and spatial scales.

Page 10, “the shallow depths of the oxycline”: Be specific. How shallow? Does the oxycline depth vary throughout the year?

Page 12, “tropical floodplains”: I would assume the magnitude of deposition should be set more by tectonics than climate, since subsiding basins with aggrading rivers should have significant floodplain deposition rates regardless of if they are in the tropics. Please clarify this point.

Page 12, “sediment deposition in the floodplains could account for a 6.1%±1% of sediment discharge by Amazon River”: How does this compare to the rates of sediment transport and storage computed by Repasch et al. (2020)? Other work by Repasch and Scheingross could provide insight into OC oxidation during POC storage in the Amazonian floodplain. Repasch, M., Wittmann, H., Scheingross, J. S., Sachse, D., Szupiany, R., Orfeo, O., et al. (2020). Sediment transit time and floodplain storage dynamics in alluvial rivers revealed by meteoric 10Be. *Journal of Geophysical Research: Earth Surface*, 125(7), e2019JF005419. <https://doi.org/10.1029/2019JF005419>

Page 12, “river SSC and floodplain hydrodynamics, do not synchronize with the temporal and spatial variations of the water stage”: Based on the Exner equation you should get most rapid erosion and deposition when you have a spatial or temporal

change in SSC so this offset makes sense to me. I think setting up a testable hypothesis in the Introduction based on sediment transport conservation of mass would help the reader to understand this temporal offset more clearly.

Page 13, "sediment would be deposited over the vegetated area [60, 61]": This should vary depending on the delta and the magnitudes of riverine versus tidal and storm-driven sediment fluxes. I don't think the references here are particularly convincing, since the Mississippi River deposits enough sediment within its channel that it is very superelevated.

Page 13, "exacerbating channel aggradation and reducing flood capacity": This is an important point but requires some more detail. Reaches could gradually adjust to increased SSC in other ways such as migrating more rapidly, steepening, or avulsing.

Page 14, "The POC deposition rate on the wetland area of Amazonian floodplains is relatively small": Again, I am not sure how impactful the carbon story is. The authors argue that the methane flux could be important, unlike the CO₂ fluxes. However, they do not offer any specific numbers for Amazon CH₄ from other sources or make their own measurements of CO₂ or CH₄ outgassing. This is important because CH₄ could be produced at depth in lakes but then be oxidized to CO₂ by methanotrophs, in which case the flux may not matter. In addition, CO₂ and CH₄ have different residence times in the atmosphere so it's important to say what timescales these fluxes occur over.

Page 14, "input of deposited POC to the soil carbon pools": The authors seem to assume that adding POC will increase CH₄ production. Is this the case for floodplain lakes in the Amazon? Other limitations could be reaction kinetics, microbial abundance, and type of POC input. Since the lakes have such high biospheric POC input from photosynthesis I do not expect that adding a small amount of recalcitrant POC would significantly affect lake CH₄ emissions.

Page 14, "This Amazonian floodplain is a typical subsystem of Amazon River,": There is some justification for why you picked this floodplain, but it would be helpful to know how common lakes are along the Amazon. Was this motivated by findings in your prior work? Perhaps the entire "Study site" section could be moved into the main text and describe more of the earlier results at this site.

Page 16, "Shuttle Radar Topographic Mission (SRTM)": SRTM data is for the top of vegetation. I went through the references and found that you corrected for this in prior work, but that should be mentioned in this manuscript as well.

Page 16, "Manning's roughness coefficient": How did you delineate these regions and pick a Manning's roughness value for each? Did you pick a value independently based on field observations or iterate on different values to find a good fit of the model to the data? How sensitive was the model to delineating these domains and picking different Manning's n?

Page 16, "A Lowess filter": Please specify which filter parameters were used.

Page 17, "...high water (HW) and falling water (FW)": How did you pick these periods? Was it arbitrary or was there a quantitative threshold?

Page 17, "we define 3 cohesive sediment classes in GAIA": Do you include the effects of flocculation here? Mud is rarely transported as isolated particles in rivers and this would increase settling velocities and likely decrease the time spent in oxidating environments.

Page 17, "The deposition rate per unit area is estimated from the bed evolution rate and the default bottom layer density of 50 kg m³.": I did not understand how the deposition rate was calculated from the bed evolution rate since the equations presented have it being calculated from sediment concentrations and settling velocity. Is the bottom layer density the water + sediment in the bedload layer? Or is that the bottom of the river?

Page 17, "SSC in-situ samples": How were these samples collected? Using a Niskin bottle, Van Dorn sampler, etc.

Page 19, equation 3: Part of the results in Bouchez et al. referenced here is that POC tends to stick to finer grained sediment. Since you are using 3 different grain size classes and sources for sediment, it makes sense to assign each a different f_{POC} if they are statistically significant.

Page 19, "the identified depths separate aerobic and anaerobic sediment.": Rapid burial can also produce anaerobic environments under sediment since oxygen will have to diffuse down into the layers of lake mud. I am unclear if the calculations consider post-deposition oxidation or only consider oxidation while POC is settling out of the water column.

Page 20, "macrophytes": Please define the difference between macrophytes and forest.

Figure 1: I would like panels added to a figure showing the locations of the floodplain and where the gages are placed, similar to Figure S1 but in the main text. I am not familiar with this region and the geometry of the rivers and lake is important to understand the results. In addition, please put the circles and triangles on the figure legend. From the figure, it looks like there was a big pulse of sediment from upstream in January 2016 that the model did not capture very well... why is that? Does this affect the results?

Figure 2: I don't see these transects mentioned in the caption marked on Figure S1. Also, this caption is the first mention of POC versus floodplain carbon in general. Be sure to define POC and distinguish it from DOC.

Figure 3: It is not clear to me why the authors used a polynomial fit, and in particular a cubic equation. Comparing the linear and polynomial fits requires at least reporting R^2 values for each fit to all the data. Attributing the change in behavior for very high SSC inputs could inform what relationship you expect here and motivate picking a more complicated function than a linear fit. For example, deposition is proportional to $SSC \cdot w_s$. If the settling velocity is somehow proportional to SSC (maybe you pick up larger sediment at higher SSC?) then you could propose that $w_s \sim SSC$ and $D \sim SSC^2$.

Figure S1: How did the authors select this computational domain? Please put an arrow on the figure to indicate the direction each river is flowing. It would be helpful to have a topographic map showing the boundaries of the floodplain and have the areas shaded in.

Figure S2: This image is still confusing to try and understand what is considered the floodplain versus the model domain. Much of the area appears to be a lake (permanent water), though the boundaries of the lake does not agree with Figure S1. It is not clear how the areas for forest, inundated forest, etc. were delineated and the methods need to be explained in the caption or Methods section. One possible source could be inundation frequencies maps accessible at <https://global-surface-water.appspot.com/map>.

Figure S3: I am not sure how well the satellite measurements and field sampling agree, so a cross-plot of field versus satellite measurements is needed to validate this approach.

Figure S5: There are gaps visible between Amazon riverbank and the start of the model domain that seem important since most of the water and sediment is coming from the Amazon. Some mention of how much water flows through the secondary channel connected to the main channel versus overbank sheet flow would be helpful in addition to the delineation of fluxes in through each boundary already done in the main text.

Figure S6: Be clear which of these lines is the sediment and water from the Amazon. The main text mentions that flow into the modeling domain is positive while out is negative but it would be good to annotate the figure to remind the reader. Also, it appears that there is a net transport of water and sediment upstream (flowing backwards) into the tributaries to the lake – is that correct?

Figure S9: It's not clear how this map was produced compared to Figure S2. Is the floodplain here equivalent to the wetlands in that caption?

Version 1:

Reviewer comments:

Reviewer #1

(Remarks to the Author)

The authors have addressed my previous comments and I have no further comments.

Reviewer #2

(Remarks to the Author)

This manuscript is much improved after what appears to be careful attention and response to the comments of the three reviewers. Many of my primary concerns have been satisfied, although the following relatively minor issues remain.

Abstract: Andes mountains (missing an "s" on mountains)

Abstract: the state encompassing Manaus is Amazonas, which might be a better fit than Amazônia in this context. Elsewhere you use anglicized words to refer to the greater Amazon area (e.g., Amazonian) so it's a bit of a divergence to see the Portuguese usage here. This is admittedly a minor point, and one of the authors is Brazilian, so perhaps it's fine.

L6: I brought up the 15% vs the 6.1% you reference in the abstract in my last review. It would be useful here to hedge this number at this location by pointing out the difference in domain for these two numbers, especially since you use the exact same phrasing (Amazonian floodplains) in the abstract and on this line to refer to two different areal domains. (I do appreciate the expanded discussion later in the paper about the potential limitations/uncertainties regarding your new estimate.)

Figure 1: I appreciate the addition of (e) but remain suspect of the goodness of fit between modeled and observed SSCs since this is on a log-log scale. Please indicate if the black line is a best fit or 1:1 line in both (c) and (e).

L133: I suggest "reached a maximum" instead of "topped" here

L152: I'm confused by the numbers here. Is this the same 15% referenced from [6] in the intro? As written, they appear to refer to two different quantities, that is, fractional deposition rate of mainstem sediment on floodplains vs retention of sediment influx to the floodplain itself. Is the 15% just a coincidence on both of these numbers? I don't see 11.7% in Figure

2B anywhere.

Figure 3: Please double check the y-axis label of subplot c. Do the units work?

L232: I remain confused by the use of 11.7 here and 6.1 in the abstract. I believe they are referring to different quantities (see comment on L152) but as written it is hard to distinguish and keep track of the ratio of deposition to the mainstem flux and ratio of retention of the inflow to the floodplain. I suggest clarifying to indicate which is which when numbers are used.

Figure S7: What is the meaning of the dashed lines in the figures? Are they best fit lines? If so I am surprised by their orientations, especially in (a) and (c), since the r values are negative and the slopes should be, as well.

Finally, the figshare link includes Python pickle (.pkl) files. As described at <https://docs.python.org/3/library/pickle.html> the pickle module is not secure, and users should only unpickle data they trust, since pickle files can execute arbitrary code on a user's machine. This makes it an inappropriate format for supplemental files for the manuscript, and I strongly suggest providing data in a more standard format like netCDF which does not have the security implications of pickle files. For this reason, I have not opened the files, so it's possible something even simpler than netCDF like CSV or JSON would be appropriate.

Reviewer #3

(Remarks to the Author)

Review comments

The authors made numerous changes to address earlier comments from myself (Reviewer 3) and the other reviewers, resulting in a much-improved manuscript. The re-submitted product presents a much more detailed and nuanced picture of floodplain sedimentation and POC dynamics in the Amazon.

Overall comments

The authors argue that POC deposition does not matter when compared to NPP, except where it is deposited under anoxic conditions, where it might be oxidized to CH₄. The degree that methane production in Amazonian floodplain lakes occurs and is significant in the regional and global climate is still not clear to me. I would be fine with the authors just reporting a negative result – that floodplain POC is not significant compared to organic input from primary productivity in the Amazon – to ensure that the OC interpretations are well-supported and described consistently. Right now the paper has two main takeaways (drivers of floodplain sediment deposition and potential OC emissions as methane) and I think the paper works well as a sedimentation study without flashy results about methane emissions.

Line comments

Line 20: Unclear what "sediment attrition" means. Maybe "sediment abrasion?"

Line 32: I don't fully understand how dams would cause a 100% decrease in sediment transport. Does that mean no sediment would come from the Amazon to the ocean? This could be rephrased to be more clear.

Line 65: "Local floodplains" – local to the Amazon or somewhere else? I found this wording confusing.

Figure 1: The context imagery in the Panel A insets are really hard to see. I would use outlines of the continent and basin rather than imagery. In Panel B, it seems like there is an error in your boundary conditions to cause these flat sections of model results. Please explain why this is and if it matters in the main text. Panel E – please add error bars to the points. This will make the green points look like they have a much better fit because the SSC error bars are quite large. Caption – add a first sentence describing the figure overall before going into each panel.

Line 131: It might be helpful to set the reader's expectations for what deposition rates should scale to. Each of the reviewers had slightly different ideas about this, so reminding the reader what you expect here might help.

Figure 2: Please make the color bars go to negative values to capture the full range of deposition rates from your model outputs.

Lines 272-294: This is a long paragraph with two different estimations of sediment trapping in floodplains. Integrating the two estimates, with one as an upper and the other as a lower bound, would help condense this paragraph and improve the flow of ideas.

Line 355: Why did you pick 29 m as the elevation bound of the floodplain? Please say why and/or reference prior work.

Line 481: Why did you pick these values for the oxycline? Please briefly describe why.

Figure S2: I appreciate that the values for these thresholds were added to the main text, but I still don't understand why these values were selected. Is there a quantitative justification for the thresholds, for instance splitting up the annual hydrograph

into 4 equal time periods?

Reviewer 1

The authors use a hydrodynamic and sediment transport model to simulate sediment deposition on an Amazonian River floodplain. The methodology is generally sound. One of the key insights and impacts of this work is to inform decision makers about how floodplain sedimentation functions along the Amazon so that decisions about whether to construct dams or deforest are informed about the sedimentological consequences. Overall the manuscript was well written with good quality figures. I only have three comments:

Response:

We thank the reviewer for the valuable comments and recommendations. We have carefully addressed the reviewer's suggestions as follows:

R1C1:

The authors scale up their deposition rate to a stretch of the Solimoes/Amazon River. However, there is probably still a lot of deposition and exchange happening in smaller-order rivers. Can the authors further contextualize their results across all rivers in the Amazon basin or at least comment on how the estimate they are providing does not include other rivers and is thus an underestimate of the true amount of floodplain sediment deposition in this basin. This is an important point, because the deposition amount is not just 6% of the Amazon sediment flux, it is much higher if you account for the deposition happening in other floodplains.

Response:

Thank you for your insightful comment regarding the scaling of our deposition rate estimates across the Amazon basin. We initially attempted to upscale our deposition estimates using the modeled deposition rate per unit length by multiplying it with the length of major rivers. If we use the river length that matches the same basin (i.e., 3349 km estimated from HydroRIVERS (Lehner and Grill, 2013)) and deposition rate per unit length (i.e., $0.033 \text{ Mt km}^{-1} \text{ yr}^{-1}$), the total deposition is 211 Mt per year given the likely presence of floodplains on both banks of the river. However, this approach led to an overestimation of sediment deposition as the width of floodplains varies significantly and this method inaccurately assumes continuous floodplain presence along all river stretches, which is not the case.

Recognizing this limitation, we opted for a more conservative estimate by focusing on the deposition rate per unit area and scaling up based on the satellite-derived total floodplain area (Hess et al., 2015b). This dataset does include tributaries. However, we believe it is more appropriate for Janauacá to exclusively represent the floodplains adjacent to the main stem of the Amazon/Solimões River. We refrain from extending it to a broader domain because of the associated uncertainties. Our analysis thus excludes the smaller-order rivers, which likely causes the underestimation of the total deposition across the entire basin as significant deposition can also occur in floodplains associated with those small rivers (Trigg et al., 2012).

Correspondingly, in the revised manuscript, we provided a more detailed discussion on the limitations of our approach and clarify that our estimates primarily cover the major river floodplains,

neglecting the potential additional deposition from smaller rivers and tributaries. This will ensure a clearer understanding of the scope and implications of our findings (P18L276-P19L291):

“The estimated deposition is influenced by several uncertainties, such as the variability in floodplain trapping efficiency and the potential overestimation of the floodplain area (Fleischmann et al., 2022). Moreover, Janauacá may not represent all floodplains along the Solimões River, as there are sites with varying ratios of local drainage extent to lake area (Forsberg et al., 1988; Sobrinho et al., 2016). Given 1200 Mt sediment drained to the Atlantic Ocean by the Amazon River per year (Martinelli et al., 1989), sediment deposition in the floodplains could account for a $6.1 \pm 1\%$ of sediment discharge by the Amazon River. Our upscaling estimation focuses on the mainstem of the Solimões/Amazon River but excludes all tributaries, assuming that Janauacá is only representative of floodplains adjacent to the mainstem. This figure likely underestimates the total sediment deposition in the basin, because the floodplains of the tributaries could also trap a substantial amount of sediment (Trigg et al., 2012). Alternatively, the upscaling can be based on the river-length-specific deposition rate of $0.033 \text{ Mt km}^{-1} \text{ yr}^{-1}$. If using a river length of 3349 km from HydroRIVERS (Lehner and Grill, 2013), the river-length-based upscaling would yield the total sediment deposition of 211 Mt per year, a significantly larger figure. Given that floodplains are not always present along the river and on both sides, this figure is likely to carry a positive bias.”

R1C2:

The SRTM DEM using for model topography would have approximated a water surface to the floodplain lake with no bathymetry. Did the authors add bathymetry to the lake somehow (not mentioned)? And if not, you should discuss how this might affect your model results? That is, particularly with unsteady hydrodynamics, the amount of water stored in the floodplain lake would not be properly accounted for.

Response:

Thanks for the reviewer comment. The bias of SRTM DEM over permanently inundated regions is corrected using ICESat GLAS data and field sampling data. The details are provided in Pinel et al. (2015). The SRTM correction also included bathymetric data acquired during a June 2012 field trip, when the water level was exceptionally very high (absolute level of 24.3 m). Data were acquired using an Acoustic Doppler Profiler Current (Teledyne RD Instruments, ADCP 1200 Hz) and an echo sounder linked to a GPS station. Other in situ bathymetric data were also acquired in May 2008 and in August 2006.

In the revision (P21L345), we have briefly described the bias correction method used for SRTM DEM and refer readers to Pinel et al. (2015) for more detailed description.

“The bias over permanently inundated areas in the SRTM DEM was corrected by using the elevation data from the NASA’s Ice, Cloud, and land Elevation Satellite (ICESat) Geoscience Laser Altimeter System (GLAS). The SRTM correction also used bathymetric data from high water levels in June 2012, collected with an ADCP and GPS-linked echo sounder, along with sampling from May 2008 and August 2006. Meanwhile, vegetation-induced biases were corrected using a wetland map from Hess et al. (2015b) and a MODIS-derived vegetation height dataset

from Simard et al. (2011). These corrected ground elevations were integrated using the ANUDEM v5.3 algorithm, constrained by a drainage channel network identified from Landsat imagery. For a detailed methodology, see Pinel et al. (2015).”

R1C3:

w_s and C_b in Equation 2 were not defined

Response:

We apologize for the missing of information, which is added to the revision (P20L338). w_s is the settling velocity of the sediment particles and C_b is the SSC at the interface between suspended load and bedload, which is the SSC of the suspended sediment in our case since bedload is not included.

Reviewer 2

Thank you for the opportunity to review this paper, which describes a modeling effort to determine sedimentation in a floodplain on the Amazon River, Brazil. Overall, the paper is easy to follow and describes an interesting study on floodplain of a globally important river. The study seems interesting and is of importance to quantifying and constraining Amazonian sediment dynamics, but I do not believe it currently merits publishing in Nature Communications.

Response:

We sincerely thank the reviewer for the critical feedback and comments. We have carefully addressed the reviewer’s recommendations as follows:

R2C1:

A primary limitation is comparison to field observations. With so many knobs to twist in hydrodynamic and sediment-transport models, getting a model to match a few sparse field measurements of SSC is not sufficient evidence of model performance, in my view. Limited information is provided regarding things like settling velocity, erosion parameters, and so on, and these have great influence on the modeled sedimentation patterns. Nearly all the sediment validation data are from the mainstem of the river, and it is unclear to me where the lake SSC data are sourced from. There are also no field deposition results, which is a primary missing link in model confidence.

Response:

Thank you for the feedback on our validation. We acknowledge the complexities inherent in sediment model validation, which is a recognized challenge across this field of study due to sparse and uneven data distribution, especially in environments like Amazonian floodplains. Our study indeed leverages the data available from a few key sites within the Janauacá floodplain rather than in the mainstem of the Amazon River. Notably, similar data are typically unavailable in

other Amazonian floodplains. But we acknowledge the limitation of the available data in the Open lake site. More details are provided in our response to R2C10.

The modeled deposition rate is compared against the measurements in similar Amazonian floodplains, which shows a consistent order of magnitude. This comparative approach, while not perfect, provides a practical method to validate our model given the lack of coordinated measurements for such a complex system.

Moreover, this study represents a significant advancement by coupling hydrodynamic simulations with sediment dynamic modeling at an unprecedented level of spatial and temporal detail for this region. This integration itself is a substantial contribution, not only demonstrating the intricate interplay between floodplain dynamics and sediment processes but also highlighting the urgent need for more comprehensive data collection efforts in such regions. In the revision, we have provided more details in terms of configurations of deposition and erosion parameters. Please see our response to R2C35.

In addition to the model setup, we have conducted sensitivity analyses by perturbing the boundary SSC conditions to quantify the impacts of human activities such as damming and deforestation. These analyses suggest that our model's outcomes are robust and offer reasonable predictions under varied scenarios, further reinforcing the model's validity. Please see our enhanced analyses on the sensitivity experiments in our responses to R2C3, R3C12, R3C13.

The aforementioned points illustrate the significant progress our research has made in understanding sediment dynamics in the Amazonian floodplains, while also recognizing the need for ongoing enhancements in data collection and model refinement. In the revised method section (P24L421), we have clarified the existing modeling challenges in terms of model validation to ensure a balanced view of our study's contributions and limitations:

“The validation of modeled sediment dynamics in Amazonian floodplains remains challenging due to the dearth of both SSC and sediment deposition data. Here, the modeled sediment dynamics are validated using the SSC in-situ samples collected at three sites, which are all 0.5 m below the water surface, within Janauacá (Figure 1a) (Amaral et al., 2018, 2022; Barbosa et al., 2020), and the measured sediment deposition rate in floodplains with similar geographic settings (Smith et al., 2003; Moreira-Turcq et al., 2004; Mangiarotti et al., 2013), as there is no direct measurement of sediment deposition in the local region.”

R2C2:

Much of the results and discussion are given over to describing how sedimentation is not related to water level and is instead related to SSC or sediment flux. This seems quite expected, and is one of the reasons I doubt the appropriateness of this study, in its present form, to Nature Communications. Much discussion is given to the hysteresis between SSC and discharge, but this is a common occurrence in fluvial systems and shouldn't really be a surprise, and at the least shouldn't be a main takeaway from the paper. What is new that is being presented here?

Response:

We respectfully disagree with the reviewer at this point. While hysteresis between SSC and discharge is well-documented in fluvial systems, how this hysteresis interacts with floodplain hy-

drodynamics to influence sediment deposition in floodplains has been rarely explored. Particularly, our findings highlight a unique aspect of sediment dynamics within the Janauacá floodplain, distinct from conventional hydroclimatic drivers generalized across Amazonian floodplains. Previous research by Aalto et al. (2003) emphasizes the governing of inundation magnitude on sedimentation of sediment-laden water, observed near the Andes. Our study, however, challenges this view. We demonstrate that in the Amazon/Solimões floodplains, sediment deposition is not controlled by inundation level from the Amazon River but is significantly influenced by local floodplain hydrodynamics and SSC variations. This nuanced understanding addresses a gap in regional sedimentation models and underscores the complexity of sediment deposition mechanisms beyond traditional hydrological drivers. We have enhanced our manuscript to more clearly clarify this distinction and highlight the novelty of our findings in the context of existing literature (P2L13): “Despite its importance, our understanding of sediment deposition drivers in Amazonian floodplains remains limited. Existing studies suggest that the Amazon River inundation controls sediment deposition in the floodplains near the Andes (Aalto et al., 2003), where coarser sands are rapidly deposited during high water periods. However, this observation may not be applicable to the downstream floodplains, such as the Amazon/Solimões, which comprise the majority of Amazonian floodplain area. As the river progresses downstream, sediment composition transitions to finer sediments due to preferential deposition of coarse sediments, sediment attrition, and dilution by finer sediments from black-water tributaries (Vauchel et al., 2017). Thus, sediment deposition in the Amazon/Solimões floodplains is less likely to be influenced by river inundation magnitude and more so by local factors, such as inundation phases, river sediment concentration, and floodplain hydrodynamics.”

The discussion has also been elaborated to reflect our findings (P16L216):

“The hydrological cycle of the Amazon River is a critical driver of sediment flux to Janauacá (Junk et al., 1989; Rudorff et al., 2018), but it is noteworthy that no significant correlation exists between the water stage of the Solimões River and sediment deposition in the floodplain at both annual and seasonal scales. This deviates from the observed sediment deposition behavior in the Amazonian floodplains near the Andes (Aalto et al., 2003). The contrasting results suggest that sediment deposition in these two floodplain systems are controlled by distinct drivers. For the Amazon/Solimões floodplains where fine sediments are dominant, our investigation finds that river SSC and floodplain hydrodynamics are more influential for sediment deposition. They also do not synchronize with the temporal and spatial variations of the water stage. Notably, floodplain hydrodynamics play a crucial role in spatial distribution of sediments, keeping them suspended in the Amazon/Solimões floodplains. As a result, only a small fraction of intruded sediments are deposited in the floodplains and the rest flow back to the Amazon River during FW. The hotspot area of sediment deposition in the floodplains during different hydrological periods correlates closely with the area where kinetic energy of inundated water is greatly attenuated. This result highlights that floodplain lakes are a major sink of water kinetic energy in the Amazon Basin, trapping as much as ~11.7% of sediment sourced from the Amazon River. A similar phenomenon has been observed in other Amazon Basin floodplains which possess numerous lake systems (Wohl, 2021), which implies the broad applicability of the mechanism. Conversely, for the floodplains deficient in open waters, such as river deltas, most sediment would be deposited

over the vegetated area (Li et al., 2021; Soler et al., 2017). Given the challenge to measure floodplain dynamics from space (Pinel et al., 2020), our findings underscore the crucial value of high-fidelity hydrodynamic-sediment coupled models for the research of sediment deposition in Amazonian floodplains.”

R2C3:

The treatment of the potential future changes in SSC (Fig 3) also seems limited. Rather than an increase/decrease in SSC, wouldn't it be more useful and likely to consider a change that varies with? For example, smaller peaks in SSC (a smoothing-out of the signal), or alternatively, larger peaks (an intensification of sediment release during the wet season), or a shift in the timing of the SSC peak relative to the water peak.

Response:

Thanks for the reviewer's insights. While it is challenging to predict the potential influence of damming and deforestation on SSC, we have performed additional numerical experiments following your suggestions, i.e., reducing variation amplitude by 60% and enhancing the SSC magnitude by 60% during peak periods. Although the timing of SSC peaks can only be influenced by climatic forcing rather than anthropogenic factors, we conducted experiments that advanced the SSC peaks by 30, 60, and 90 days towards the peak water levels to explore the potential interactions between climate and human activities that might affect the variability in sediment deposition.

The design of the new experiments is described in the Method section: “While it is not entirely clear how anthropogenic disturbances affect the seasonal variations in SSC, these potential impacts were assessed through experiments, including reducing variation amplitude by 60%, and enhancing the SSC magnitude by 60% during peak periods. Although the timing of the SSC peak is influenced by climatic rather than anthropogenic forcing, supplementary experiments were conducted by advancing the SSC peaks 30, 60, and 90 days towards the water level peak to assess how interactions between climate and human activities might affect the sediment deposition in Amazonian floodplains.”

Consistent with our original sensitivity experiment which increases SSC uniformly across seasons by 60%, the new experiment that increases SSC by 60% over the peak period would also cause the decrease of the trapping efficiency of the floodplain (Figure S15). This consistency enhances our conclusion that a substantial increase of SSC due to deforestation would cause the reduction of the floodplain's capability to trap sediments. Additionally, we also conducted several numerical experiments that explore the interactions between the impacts of climate change and human disturbance. In these experiments, we shifted the SSC peaks 30, 60, and 90 days towards the water level peak.

The new experiments imply the significant influence of the SSC seasonal variation and dynamic response of deposition to multiple factors, further supporting our argument in R2C2. The temporal shift in SSC peaks results in a remarkable change in sediment influx, especially when the SSC peak occurs during RW (Figure S14e) that results in a higher deposition rate (Figure S14a). The smoothing of SSC peak by 60% (-60% smooth) did not result in a deposition rate reduced as much as the experiment that decrease the entire SSC time series (-60%) (Figure S14b). This

is because the smoothed SSC signal has higher values than the baseline simulation during RW and HW. Moreover, in Figure S14b, we found that the daily averaged SSC that only increase the values over peak days (+60% peak) is not as high as the experiment that increase the entire time series (+60%), which results in reduced sediment flux and deposition rate (Figure S14b and f).

These experiments clearly show the substantial influence of seasonal variation in SSC and the dynamic response of sediment deposition rate to multiple factors, reinforcing our arguments presented in R2C2. Notably, temporal shifts in SSC peaks led to significant changes in sediment influx, particularly when peaks coincide with the RW period, resulting in higher deposition rates, as shown in Figure S14e. Conversely, smoothing the SSC peak by 60% did not reduce the deposition rate as much as reducing the entire SSC time series by the same percentage (Figure S14b). This occurs because the smoothed SSC maintains higher values during both RW and HW compared to the baseline scenario. Moreover, we found that increasing SSC by 60% during its peak days (Figure S14d) does not lead to as high a deposition rate as increasing the entire time series by 60% (Figure S14b). This is because the SSC peak period slightly shifts among the simulation years, leading to a lower overall sediment flux and deposition rate (Figures S14b and f).

In the revised manuscript, the results of the new experiments are briefly referenced in the main context at P12L178: “The seasonal change of SSC implies a more dynamic response of the deposition rate, with shifts in SSC peaks causing notable increases in sediment influx during HW (Supplementary Figure S14a, c and e).” More details were reported in the caption Supplementary Figure S14 and S15.

Figure S14: The daily averaged deposition rate (a), suspended sediment concentration (SSC) at the WLBCs (c) and sediment flux at the WLBCs (e) of the baseline simulation and the experiments that shift the SSC by 30 days, 60 days and 90 days towards the HW period. The daily averaged deposition rate (b), suspended sediment concentration (SSC) at the WLBCs (d) and sediment flux at the WLBCs (f) of the baseline simulation and the experiments that increases the SSC peak by 60% (+60% peak), increases the SSC by 60% (+60%), smooths the SSC variation by 60% (-60% smooth) and smooths the SSC by 60%. The number in front of each experiment name in the labels is the corresponding deposition rate. These experiments show the substantial influence of intraannual variation in SSC and the dynamic response of sediment deposition rate. Notably, temporal shifts in SSC peaks led to significant changes in sediment influx, particularly when peaks coincide with the RW period, resulting in higher deposition rates, as shown in (e). Conversely, smoothing the SSC peak by 60% did not reduce the deposition rate as much as reducing the entire SSC time series by the same percentage (b). This occurs because the smoothed SSC maintains higher values during both RW and HW compared to the baseline scenario. Moreover, we found that increasing SSC by 60% during its peak days (d) does not lead to as high a deposition rate as increasing the entire time series by 60% (b). This is because the SSC peak period slightly shifts among the simulation years, leading to a lower overall sediment flux and deposition rate (b and f).

Figure S15: Annual averaged sediment trapping ratio in 2016 of the baseline simulation and the experiments that increases the SSC peak by 60% (+60% peak), increases the SSC by 60% (+60%), smooths the SSC variation by 60% (-60% smooth) and smooths the SSC by 60%.

R2C4:

Overall, I find the results to be interesting but not appropriate for this journal. In addition to the concerns above, the results are not sufficiently tied into a larger contextual framework, or when they are, this feels unsatisfying. For example, burial of sediment-associated POC is shown to be a very small player in the overall budget. But then the claim is made that this is evidence for why POC needs to be accounted for in lake CH₄ models. . . I don't follow this argument.

Response:

Thank you for your insightful comment. We apologize for any confusion caused by the tone and presentation of our findings regarding POC deposition in relation to methane (CH₄) models in our manuscript. We acknowledge that the comparison between the modeled POC deposition and NPP may have conveyed an unintended emphasis on the significance of POC deposition. It is indeed true that our results show POC deposition to be a relatively minor component of the overall carbon budget when compared to carbon fixation from ecosystem primary production.

Our intent is not only to report on the quantities of POC but also to highlight its ecological and biogeochemical relevance, providing a comprehensive understanding that supports future research and modeling efforts in similar settings. In particular, this study highlights that due to its strong heterogeneity, the impact of sediment deposition on carbon cycling varies considerably between lakes and wetlands. To improve clarity and avoid future confusion, we have revised our manuscript to better contextualize the role of sediment-associated POC within the larger carbon cycle and its implications for methane emission modeling.

In the revision, we first rephrased our statement and clarified the scale of POC deposition relative to NPP, subtly emphasizing its presence without overstating its comparative significance:

In the abstract, “Additionally, sediment deposition imports a large amount of biologically active carbon from Amazon River to Janauacá, albeit not comparable to carbon fixation from

ecosystem primary production. As experiencing short oxygen exposure, the deposited carbon would likely be sequestered or fuel methane emissions.” is rephrased to “Additionally, we show that the deposition of sediment-associated organic carbon only plays a minor role in fueling carbon dioxide and methane emissions from Amazonian floodplains.”

In the result section (P14L181), “Sediment deposition in the floodplain is shown to import a large amount of biologically active particulate organic carbon (POC) from Amazon River to the lake sediment in Janauacá” is rephrased to “With sediment deposition, considerable biologically active POC from the Amazon River was also deposited in the floodplain. However, quantitatively, the deposited POC per year is markedly lower than the annual soil organic carbon input from ecosystem primary production (Figure 4a, b).”

The discussion section (P19L296-P20L322) has also been rewritten to improve clarity and link to a larger conceptual framework:

Amazonian floodplains play an important role in the global CO₂ and CH₄ cycles (Murguía-Flores et al., 2023). Due to the enormous amount of sediment deposition in the floodplains, it is reasonable to speculate that the deposition of sediment-associated POC could be important for the regional CO₂ and CH₄ emissions (Saunois et al., 2019; Guilhen et al., 2020). However, our results do not support this hypothesis. Compared to the wetland soil and lake sediment carbon that are sourced from ecosystem primary production, the soil/sediment carbon input due to POC deposition is considerably smaller. Therefore, POC deposition in Amazon floodplains is unlikely to contribute to the high CO₂ and CH₄ emissions observed in the region, despite its crucial role in the supply of critical nutrients to the floodplain soils (Petsch et al., 2023). Furthermore, because Amazonian floodplains are one of several sediment deposition hotspots in the world, we suspect that POC deposition also only plays a minor role in fueling CO₂ and CH₄ emissions in other productive floodplains across the globe. However, for low-productivity floodplains in high latitudes, fluvial POC deposition remains important for regional CO₂ and CH₄ emissions (Herbst et al., 2024).

Although sediment deposition in Amazonian floodplains is not strongly tied to regional CH₄ emissions, due to its strong spatial heterogeneity and temporal variations it can influence the spatial and temporal variability of CH₄ emissions within each floodplain. For instance, we find that the POC deposition rate in the lake area is approximately five times higher than that in the wetland area. For the deepest lake area, the POC deposition rate during HW can reach 0.57 g C m⁻² day⁻¹, which becomes comparable to the measured in-lake NPP. With limited oxygen exposure, a substantial fraction of these biologically active POC could be oxidized to CH₄ during the HW period through methanogenesis (Segers, 1998; Sobek et al., 2009). As lake CH₄ models rarely include this carbon source, they would likely fail to capture the related CH₄ emission pulses from floodplain lakes in the Amazon (Tan et al., 2024). Therefore, our results underscore the importance to couple floodplain hydrodynamic, POC deposition, and CH₄ dynamics for more realistic CH₄ modeling in floodplain lakes.

We believe these modifications will make the significance of our findings clearer and enhance the manuscript’s contribution to the field.

R2C5:

Finally, although perfectly readable as-is, editing for English usage by a sharp-eyed editor would be improve the manuscript.

Response:

Thanks to the reviewer comment. We have asked communication service of our lab for English editing of the introduction and discussion sections.

R2C6:

Page 1 Check capitalization of Amazon River. Overall, English language editing would be useful throughout the paper.

Response:

Sorry for the typos. We make sure “Amazon River” is capitalized throughout the revised manuscript. As mentioned in R2C5, we have improved the English language editing.

R2C7:

Page 2 15% of river sediment: this number appears to be roughly double what you estimate in the results. Can you describe why the numbers don't match?

Response:

The figure of 15% represents a broader estimate that includes sediment deposition across all Amazonian floodplains from the foot of the Andes to the river outlet. In contrast, our model specifically focuses on the floodplains along the mainstem of the Amazon/Solimões River, the major component of Amazonian floodplains. The domain difference is the major reason for the discrepancy.

Moreover, we acknowledge the possibility that our estimate is underestimated. Our estimate focuses on the deposition rate per unit area and scaling up based on the satellite-derived total floodplain area (Hess et al., 2015b). This method, while more precise in our analysis, likely underestimates the total deposition across the entire basin as it does not account for smaller-order rivers and their floodplains where significant deposition can also occur. We acknowledge that excluding these smaller-order rivers and their associated floodplains likely leads to an underestimation of the actual sediment deposition within the Amazon basin. However, this method is more conservative and has much less uncertainties compared to the upscaling method using the deposition rate per unit length. Please see our response to R1C1.

In the revised manuscript (P18L276), we provided a more detailed discussion on the limitations of our approach and clarify that our estimates primarily cover the major river floodplains, neglecting the potential additional deposition from smaller rivers and tributaries. This will ensure a clearer understanding of the scope and implications of our findings:

“The estimated deposition is influenced by several uncertainties, such as the variability in floodplain trapping efficiency and the potential overestimation of the floodplain area (Fleischmann et al., 2022). **Moreover, Janauacá may not represent all floodplains along the Solimões River, as there are sites with varying ratios of local drainage extent to lake area (Forsberg et al., 1988; Sobrinho et al., 2016).** Given 1200 Mt sediment drained to the Atlantic Ocean by the Amazon

River per year (Martinelli et al., 1989), sediment deposition in the floodplains could account for a $6.1 \pm 1\%$ of sediment discharge by the Amazon River. Our upscaling estimation focuses on the mainstem of the Solimões/Amazon River but excludes all tributaries, assuming that Janauacá is only representative of floodplains adjacent to the mainstem. This figure likely underestimates the total sediment deposition in the basin, because the floodplains of the tributaries could also trap a substantial amount of sediment (Trigg et al., 2012). Alternatively, the upscaling can be based on the river-length-specific deposition rate of $0.033 \text{ Mt km}^{-1} \text{ yr}^{-1}$. If using a river length of 3349 km from HydroRIVERS (Lehner and Grill, 2013), the river-length-based upscaling would yield the total sediment deposition of 211 Mt per year, a significantly larger figure. Given that floodplains are not always present along the river and on both sides, this figure is likely to carry a positive bias.”

R2C8:

Page 4 Please add a figure reference after each statement or claim, rather than grouping reference to three figures together at the end of the paragraph.

Response:

This is corrected in the revision (P6).

R2C9:

Page 5 This is intuitive, although I could not find a figure showing the main-stem sediment flux ($Q \cdot SSC$) anywhere.

Response:

Thanks for the reviewer comment. We added a new figure to the supplement (Figure S8) showing the main-stem discharge and sediment flux measured at Manacapuru.

Figure S8: Time series of Amazon River discharge and sediment load measured at Manacapuru.

R2C10:

Figure 1 I would like to see a clearer comparison of modeled vs observed SSC, and not just a timeseries as currently provided in (b). It is very hard to see any dynamics in the timeseries

with the different scales. Why is the open lake in-situ timeseries not in the same graph as the other sites? The inset in (b) also should be a scatter plot and not a simplified timeseries, which may mask important variability that should be discussed. Why do you not show the NRMSE for the open lake SSC in the caption?

Response:

Thank you for the comment. Scatter plots of modeled water level and SSC against observations have been provided in Figure 1c and 1e in corresponding to timeseries plots in Figure 1b and 1d. The comparison of the mean SSC over the four hydrological periods is also added to the scatter plot in Figure 1e. Due to data availability constraints in the Open lake site, direct time series comparison and the evaluation metrics are not possible for this specific location. Instead, we had to use the mean and standard deviation values reported in Amaral et al. (2022), which provided a viable method to validate our model indirectly. With 4 values only, it is less reasonable to calculate NRMSE for the Open lake site. We have clarified this methodological decision in the caption of Figure 1 to ensure transparency about the data usage and validation process.

Figure 1: (a) The model’s computational domain (limited to the low-lying floodable regions with bottom elevation < 29 m). White labels show 8 inflow boundaries (BCF) and 3 water level open boundaries (WLBC) connecting the main channel of the Amazon River. Magenta transects at WLBCs are used to calculate the flow and sediment exchange flux between the floodplain and the Amazon River main channel. Red labels show two sampling sites of water level at WL_A and WL_B (red triangles) and three sampling sites for SSC at Upstream, Open lake and Downstream (black squares). The red dashed line outlines the lake area, which is used to estimate the lake-specific POC deposition and the corresponding oxygen exposure time (OET). The arrows in the upstream region indicate the river flow direction. The inset shows the minimized view of the domain boundary and the location of the Manacapuru gauge at the mainstem of the Amazon River. Comparison of the modeled water level (b and c) and SSC (d and e) (solid and dashed lines) against the measurements (circles and triangles) in the Janauacá floodplain. The inset in (d) shows the model-data comparison at Open lake, where model validation is based on the comparison with published mean and standard deviation values (error bar) (Amaral et al., 2022). The normalized root-mean-square error (NRMSE) is 0.04, 0.03, 0.29 and 0.21 for water level at WL_A and WL_B and for SSC at Upstream and Downstream sites, respectively.

R2C11:

Page 6 The correlations (-0.35, 0.94, etc.), or plots where one could interpret them, are not shown anywhere in figure 2, despite a reference to this figure.

Response:

Thanks for the reviewer comment. A scatter plot (Fig. S7) that better interpret the correlation among variables is now provided in the supplementary. A reference to this figure is added to the caption of Figure 2: “Corresponding scatter plots of annual mean deposition rate, water level and SSC are provided in Supplementary Figure S7”.

Figure S7: Scatter plots of annual mean deposition rate, water level and SSC.

R2C12:

Here you say SSC “topped” (reached a maximum?) during RW but on page 4 you stated it peaked during LW.

Response:

Sorry for the typo. We note that SSC starts increasing during LW and reaches its maximum during RW. We have corrected it on page 6 of the revision (P6L111) that: “Specifically, high and low SSC concentrations occur during RW and HW periods, respectively.”

R2C13:

“more widespread” is this simply because the lake is deeper at HW?

Response:

Thanks for the comment. Sediment deposition being “more widespread during HW” is a result of joint impact from inundation, SSC in the Amazon River and floodplain hydrodynamics. While a substantial amount of sediments were supplied from the Amazon River during RW, they could not be deposited due to the high flow momentum. When inundated water flowed to the deeper lake area during the subsequent HW and slowed down (see Supplementary Figure S6c), the condition to favor sediment deposition was formed. In response to this comment, we have elaborated this process in the revision (P9L138-L145):

“Particularly, the model shows that the rising water supplied a dominant amount of sediments to the floodplain (Supplementary Figure S4b). These intruded sediments were only deposited to the deep water area of Janauacá Lake during the subsequent HW, several months after RW (Figure 2f), when the slowing of water movement favored sediment deposition (Supplementary Figure S6c). In regions adjacent to WLBC3 where flows are persistently fast during RW and HW (Supplementary Figure S6b and S6c), the deposition rates are reduced due to continuous sediment resuspension and transport (Figure 2e and 2f).”

R2C14:

I only see a maximum value of about 0.07% for the trapping ratio.

Response:

Thanks for the comment. We note that the maximum value of the trapping ratio is $\sim 0.09\%$ (upper bound of the shade in Figure 2b) but its multi-year averaged value is up to $\sim 0.07\%$ (dashed line in Figure 2b).

R2C15:

I do not understand the sentence beginning with “For 0.34 Mt yr⁻¹ . . .”

Response:

This sentence describes that within 0.34 Mt sediment influx per year, the floodplain traps $11.7 \pm 7.2\%$, which number is consistent to that estimated in Mangiarotti et al. (2013). In response, we rephrased this sentence (P9L150): “Of the 0.34 Mt mean annual sediment influx (Figure 2a), $11.7 \pm 7.2\%$ was retained in the floodplain and the rest was returned to the Amazon

River during FW (Figure 2b), which is in line with the previous estimate (about 15%) of flood-plain retention that was conducted over the 390-km Amazon reaches from Itacoatiara to Óbidos (Mangiarotti et al., 2013). Like sediment deposition, the sediment trapping ratio also reached the peak values in the RW period (Figure 2b)."

R2C16:

Figure 2 Using bar charts (in (c)) that do not start the y-axis at zero is misleading and should not be used. If you keep a bar chart in this figure you must start them at zero.

It would be very useful to print the total deposition (on the figure itself, as text) for each period in subplot d, e, f, g to ease comparison.

Response:

We made modifications to Figure 2 according to the reviewer's suggestion. The y-axis in the bar chart (Figure 2c) now starts from zero. The averaged deposition rate per unit area over each hydrological period is provided on the figure as text.

Figure 8: (a) Daily averaged exchanging rate of **monthly** flow and suspended sediment through the open boundaries (transects in Figure 1a). The positive and negative values represent the flux into and out of the Janauacá floodplain, respectively. (b) Daily averages of sediment deposition rate and sediment trapping ratio. The sediment trapping ratio is the ratio of the sediment deposition in the floodplain versus the sediment load of the Amazon River at the Manacapuru gauge. The negative value of deposition rate indicates bed erosion during FW and LW. (c) Annual mean sediment deposition rate of Janauacá, annual mean water level and annual SSC of the Amazon River from 2008 to 2016. The annual SSC each year is the number over the bar. **Corresponding scatter plots of annual mean deposition rate, water level and SSC are provided in Supplement Figure S7.** Maps of the modeled deposition rate averaged over four hydrological periods: (d) LW, (e) RW, (f) HW and (g) FW. **The number behind each hydrological period is the deposition rate.**

R2C17:

Page 8 Although the increase/decrease in SSC does drive a change in deposition, this is not a strongly nonlinear response.

Response:

Thank you for the reviewer's insights. To address this comment, we have performed more detail analyses to understand the spatiotemporal variation of the deposition change.

In the updated Figure 3, we added the daily averaged sediment flux computed from each experiment as Figure 3b according to your suggestion in R2C21. In Figure 3c, instead of using box plots to represent mean and standard deviation across multiple years, we explicitly show the relationship for each simulation year, which highlights the increasing nonlinearity over time. This nonlinear change is closely related with the floodplain bed evolution. We changed the y-axis from $D_{exp}/D_{baseline}$ to $D_{exp}/D_{baseline} - SSC_{exp}/SSC_{baseline}$ to highlight the detailed change.

We've also added Figure S11 in the Supplementary, showing significant changes in bed elevation due to sediment deposition, particularly near the domain's open boundary and in deeper water regions where more deposition occurs. Comparing the simulation results between 2008 and 2016, we found a clear difference in deposition rate and the exchange rate of flow and sediment (Figure S12). In particular, even with only boundary SSC perturbed, we observed alterations in flow flux magnitude and timing at varying SSC levels in 2016 (Figure S12d), impacting sediment flux (Figure S12f). Our experiments responding to R2C3 reveal a dynamic response of sediment deposition to the shift in relative timing of SSC and water level peaks, suggesting that minor flow rate changes can significantly affect deposition rates.

Moreover, the spatial distribution $D_{exp}/D_{baseline}$ further supports our findings, particularly in deeper regions where bed evolution is significant (Figure S12). In these regions, $D_{exp}/D_{baseline}$ aligned with $SSC_{exp}/SSC_{baseline}$ in 2008 (Figure S13a). However, in 2016, this consistency decreased, especially at higher SSC levels in Figure S13b, where the $D_{exp}/D_{baseline}$ typically fell below the corresponding $SSC_{exp}/SSC_{baseline}$.

In summary, deforestation-induced increases in SSC levels in the Amazon River lead to a

nonlinear rise in deposition rates, and this rate of increase diminishes over time, reflecting the complex interplay of sediment deposition, floodplain bed evolution, and hydrodynamics.

In response to this comment, we slightly reframed our statement in the abstract: “Our model experiment further demonstrates that widespread deforestation would reduce the trapping efficiency of the floodplains **over time**, exacerbating downstream river aggradation.”

The result section was rephrased to improve the accuracy of the description (P11L164):

“However, if the SSC level in the Amazon River increases substantially due to deforestation, the gain of sediment deposition in the floodplain would gradually lose to the surge of river sediment influx over time (Figure 3c and Supplementary Figure S15), causing higher fractions of river sediment to transport downstream instead of being deposited in the floodplain. Our analysis shows that this reduction of the sediment retention in the floodplain is a result of floodplain bed evolution (Supplementary Figure S11~S13). Under the extreme scenario of agricultural expansion (60% increase in SSC), the trapping efficiency of the floodplain is predicted to decline by 7% by the tenth year of the simulation, as the floodplain elevation near its open boundary rises by more than 0.5 m (Supplementary Figure S11). It is worth noting that the human-induced alteration of SSC levels will make little changes to the seasonal and inter-annual variabilities of sediment deposition and sediment exchange flux (Figure 3a and 3b and Supplementary Figure S10b), although their seasonal variations are intensified.”

The discussion section was also elaborated to incorporate the new findings (P17L248):

“This is because the progressive increase of floodplain bed elevation, especially in deeper water areas, would affect the magnitude and timing of flow and sediment exchanges, as well as the spatial distribution of deposition adjusted (Supplementary Figure S12 and S13), hindering a linear increase of deposition rates. With sediment deposition saturated, the trapping efficiency of Amazonian floodplains will decline sharply and higher proportions of sediment will remain in the Amazon River main channel. As deforestation increases sediment supply, the decreased floodplain trapping efficiency would increase the likelihood that the sediment transport capacity of the Amazon River is exceeded. Consequently, more sediments would be deposited in downstream river channels, exacerbating channel aggradation and reducing flood capacity (Pfeiffer et al., 2019; Parrinello and Kondolf, 2021).”

Figure 3: Monthly variations of the simulated daily mean deposition rate (a) and **daily mean suspended sediment flux through the open boundaries** (b) on different SSC perturbation levels. (c) The change ratio of the simulated annual mean sediment deposition rate on different SSC perturbation levels **for each simulation year**. In (a) and (b), the baseline is the simulation without SSC perturbation and different percentages represent the SSC perturbation levels in Amazon River. In (c), D_{exp} is the simulated sediment deposition rate in the perturbed SSC simulations (SSC_{exp}) and $D_{baseline}$ is the simulated sediment deposition rate without SSC perturbation ($SSC_{baseline}$). **For each year in (c), the simulations with reduced SSC are fitted using a linear relationship, and the simulations with increased SSC are fitted with second-order polynomial functions. The number after each labeled year is the coefficient of the second-order terms that measures the nonlinearity.**

Figure S11: The difference of the total change of floodplain bed elevation from 2008 to 2016 between an experiment simulation and the baseline simulation (ΔBed elevation).

Figure S12: Time series of deposition rate in 2008 (a) and in 2016 (b) on different SSC perturbation levels. The corresponding exchanging rate of flow and sediment in 2008 (c and e) and in 2016 (d and f). Comparing the simulation results between 2008 and 2016, there is a clear difference in deposition rate and the exchange rate of flow and sediment. Even with only boundary SSC perturbed, alterations are observed in flow flux magnitude and timing at varying SSC levels in 2016 (d), impacting sediment flux (f).

Figure S13: Spatial distribution of $D_{exp}/D_{baseline}$ in 2008 (a) and 2016 (b). In deeper regions near the floodplain's open boundary, $D_{exp}/D_{baseline}$ aligned with $SSC_{exp}/SSC_{baseline}$ in 2008 (a). However, in 2016, this consistency decreased, especially at higher SSC levels in (b), where the $D_{exp}/D_{baseline}$ typically fell below the corresponding $SSC_{exp}/SSC_{baseline}$.

R2C18:

The linear increase/decrease in deposition in response to linear changes in SSC is intuitive, yes? If deposition is thought of as a response to the product of SSC and inundation time, and SSC changes, then so does deposition. The sentence beginning “Interestingly” makes it seem like a surprise.

Response:

Thank you for the comment. The reviewer is right that the relationship between SSC and sediment deposition is typically linear. The use of “Interestingly” was intended to highlight the nonlinear response observed in deposition rates under land use change scenarios. To avoid confusion, we have removed “Interestingly” in the manuscript accordingly.

R2C19:

More detail regarding this apparent decrease in deposition is required. Where is the erosion occurring, and is it physically realistic? Could this just be a numerical artifact? Also, don't these scenarios leave out the possibility that the river channel itself will change, potentially altering the dynamics of the sediment movement between mainstem and floodplain in an unexpected way?

Response:

We followed the reviewer's suggestion and found that these scenarios indeed alter the floodplain bed evolution, resulting in the nonlinear increase of the deposition rate over time. Please refer to our response to R2C17 for detailed explanation.

The erosion typically occurred near WLBC3 and within the channels connecting to WLBC1 (see Supplementary Figure S11). However, we have revised the discussion concerning "excessive erosion" in the case of elevating SSC by 60%, because the result only occurred when we ran the scenario on a different machine with a different compiler. Once we used the same machine to run all the simulations, the simulation does not show 'excessive erosion'. This discussion has thus been removed from the revised manuscript. The re-evaluation has verified that the discrepancies were minor and do not affect the main conclusions of our study.

R2C20:

Page 9 Given that sediment deposition plays a negligible role in the carbon budget of this lake, how relevant is it to this paper, unless connected to a larger conceptual framework or understanding of carbon budgets in Amazon floodplains?

Response:

Thanks for the comment. In the revision, we clarified the scale of POC deposition relative to NPP and elaborated the implication of this study for carbon cycling in the discussion. Please see our response to R2C4.

R2C21:

Figure 3 Help the reader understand the timing by showing the average river sediment flux to compare with the deposition rates plotted in (a)

Response:

A subplot Figure 3b was added that shows the averaged river sediment flux.

R2C22:

Page 11 Rather than referring to their estimate of 0.3 Mt km⁻¹ yr⁻¹, why not normalize it to a single side of the floodplain from the beginning?

Response:

Thanks for the reviewer comment. We prefer to keeping the original estimated number of 0.3 Mt km⁻¹ yr⁻¹, as a simple normalization to a single side may distort their estimate given the complex geographic conditions in the Amazon River, as mentioned in R1C1, resulting uncertainties in such estimates.

R2C23:

"The magnitude of"... I do not understand the point of this sentence. It seems to describe a two order of magnitude variability in sediment deposition on river floodplains. It is not surprising this should vary given the likely wide diversity of floodplain environments in the different studies, to say nothing of the drainage basins for each of these river systems.

Response:

This sentence was intended to argue that the modeled floodplain sediment deposition rate is comparable to not only those in Amazonian floodplains but also in other tropical floodplains. We agree with the reviewer that the tectonic factors may be even more important for the spatial variability of sediment deposition in the world. Therefore, to avoid the potential confusion, we only focus on Amazonian floodplains in the revision, which has already encompassed a large domain and drawn great interests. We have removed the discussion regarding the other tropical floodplains.

R2C24:

Figure 4: It is hard to know which one [32] refers to in the legend, since they are listed as Author, Year in the plot labels. Make it easier for the reader to interpret either by reference to color, name, or something else. Also, add (this study) to the label for your results. In (c), tell us what OET means [You define it elsewhere in the paper, but the captions should stand on their own].

Response:

In response to the reviewer comment, we added “[this study]” to our results in the xtick labels. In the figure caption, we provided references to all measured NPP results and the full name of OET. The mean value is annotated above the corresponding bar plot.

Figure 4: (a) Comparison of the simulated mean annual POC deposition rate with the measured NPP (i.e., Amaral et al., 2018 [32], Forsberg et al., 2017 [13] and Melack&Engle, 2009 [47]) in the lake area of floodplains. Due to the large spread of NPP data in [32], the mean value is shown here, above each bar. (b) Comparison of the simulated mean annual POC deposition rate with the measured (i.e., Potter et al., 2014 [49]) and Engle et al., 2008 [48]) or derived NPP in the wetland area of floodplains. (c) Multi-year mean oxygen exposure time (OET) of deposited POC in Janauacá. (d) The cumulative distribution function of multi-year mean OET (weighted by the deposition rate) of grid cells in the lake area and non-lake area of Janauacá. The lake area and the wetland area is defined in Figure 1a and Supplementary S1, respectively.

R2C25:

Page 12 What is the meaning of the +/- ranges in these measurements? That is, what components go into determining the range? I did not see this in the Methods.

Response:

The amount of sediment deposition (77.3 ± 13.9 Mt) is estimated using the floodplain area within the drainage basin of the Amazon/Solimões River mainstem (i.e., $58,103 \text{ km}^2$) and the modeled sediment deposition rate per unit area ($1.33 \pm 0.24 \text{ kg m}^{-2} \text{ yr}^{-1}$). The POC deposition is computed by applying the mean POC ratio of 1.27% (Page 19). Given the 1200 Mt yr^{-1} sediment drained to the Atlantic Ocean by Amazon River, the mean, upper bound and lower bound of the fraction that sediment deposition in the floodplains account for are respectively estimated as: $77.3/(77.3+1200)$, $(77.3+13.9)/(77.3+13.9+1200)$, and $(77.3-13.9)/(77.3-13.9+1200)$, which yields $6.1\% \pm 1\%$.

In response to the comment, we clarified the estimation of sediment deposition in the Methods section (P24L434): “**The amount of sediment deposition in the floodplains within the drainage basin of the Amazon/Solimões River mainstem can be estimated using the total floodplain area and the modeled deposition rate per unit area.**”. We also mentioned the POC ratio used in the estimation of the POC deposition in the Results section (P18L275): “**using the POC/sediment mass ratio of 1.27% (Bouchez et al., 2014)**”.

R2C26:

How does this estimate square with the comparison made on page 11: 0.03 vs 0.3 Mt/km/yr? If that higher estimate were applied with an average floodplain width, would it result in 60% of the sediment discharge?

Response:

Thanks for the reviewer comment. The upscaled amount of sediment deposition is estimated using the floodplain area within the drainage basin of the Amazon/Solimões River mainstem and the modeled deposition rate per unit area. The floodplain area is carefully derived from the flooded wetland extent during high flood conditions (Hess et al., 2015a) and basin boundaries (Mayorga, E. and Logsdon, M.G. and Ballester, M.V.R. and Richey, J.E., 2012). We believe that this upscaling method is more accurate than using the Amazon River length and the deposition rate per unit length (either our estimate 0.033 vs $0.3 \text{ Mt km}^{-1} \text{ yr}^{-1}$ by Mangiarotti et al. (2013)).

We initially attempted to upscale our deposition estimates using the modeled deposition rate per unit length by multiplying it with the length of major rivers. If we use the river length that matches the same basin (i.e., 3349 km estimated from HydroRIVERS (Lehner and Grill, 2013)) and deposition rate per unit length (i.e., $0.033 \text{ Mt km}^{-1} \text{ yr}^{-1}$), the total deposition is 211 Mt per year given the likely presence of floodplains on both banks of the river. However, this approach led to an overestimation of sediment deposition as the width of floodplains varies significantly and this method inaccurately assumes continuous floodplain presence along all river stretches, which is not the case.

R2C27:

“it is surprising to note” is this actually surprising? Given the 2-4x change in yearly river discharge but approximately order of magnitude change in SSC, the sediment flux is an obvious parameter to compare against.

Response:

We appreciate the reviewer comment. The floodplain deposition is indeed closely correlated to sediment flux as indicated by the intraannual variation of sediment flux and deposition in Figure 2. Here, we disentangle the control of sediment flux into the SSC and river stage to highlight our finding that the deposition is not governed by the river stage, challenging a conventional understanding in Aalto et al. (2003). We have addressed this in our response to R2C2 and clarified our hypothesis in the revised introduction. In response to this comment, we emphasized the decoupling of sediment deposition from river stage and its implications for understanding floodplain dynamics (P16L216):

“The hydrological cycle of the Amazon River is a critical driver of sediment flux to Janauacá (Junk et al., 1989; Rudorff et al., 2018), but it is **noteworthy** that no significant correlation exists between the water stage of the Solimões River and sediment deposition in the floodplain at both annual and seasonal scales. **This deviates from the observed sediment deposition behavior in the Amazonian floodplains near the Andes (Aalto et al., 2003). The contrasting results suggest that sediment deposition in these two floodplain systems are controlled by distinct drivers.**”

R2C28:

“Notably, floodplain hydrodynamics play a crucial role in distributing sediment in Amazonian floodplains” How else would sediment be distributed across the floodplain?

Response:

We agree that the spatial distribution of sediments in Amazonian floodplains is controlled by floodplain hydrodynamics, but the change of spatial distribution across a hydrological cycle is also impacted by the river SSC and water stage that determine the amount of boundary flux and the floodplain hydrodynamics. These factors are tightly connected in terms of the floodplain sediment dynamics. In response to this comment, we rephrased this sentence to avoid confusions (P16L225):

“Notably, floodplain hydrodynamics play a crucial role in spatial distribution of sediments, **keeping them suspended in the Amazon/Solimões floodplains**. As a result, only a small fraction of intruded sediments are deposited in the floodplains and the rest flow back to the Amazon River during FW. The hotspot area of sediment deposition in the floodplains **during different hydrological periods correlates closely with the area where kinetic energy of inundated water is greatly attenuated.**”

R2C29:

“could trap a dominant fraction” this sentence needs to be more quantitative to be meaningful. Same with the following sentence. Right now it reads as a statement about how floodplains slow the speed of water flow compared to the mainstem. That seems rather obvious, yes? But then at the end of page 12 it seems like you may be comparing unvegetated to vegetated floodplains? I am somewhat confused by the argument here.

Response:

Thanks for the comment. This sentence has been rephrased to include the exact number of

fraction: “floodplain lakes in the Amazon Basin could trap ~11.7% sediment sourced from the Amazon River.”

Regarding the contrasting behaviors in sediment deposition across floodplain environments, our statement aimed to illustrate how floodplain characteristics, such as the presence of lakes or extensive vegetation, distinctly influence sediment deposition patterns. Specifically, open waters play a unique role in the sediment dynamics of the lake-abundant floodplains, such as the Amazon/Solimões floodplains. This differentiation underscores the applicability scope of our findings. Moreover, the diverse mechanisms of sediment retention observed across various floodplain configurations necessitate detailed hydrodynamic and sediment transport modeling to accurately capture these dynamics.

R2C30:

Page 13 It is not clear that the lack of floodplains will result in river channel aggradation.

Response:

Thanks for the comment.

The increased SSC in the Amazon River mainstem corresponds to the increased sediment supply from the upstream of the Amazon River. With reduced sediment trapping ratios in floodplains, the likelihood that the transported sediments exceed the sediment transport capacity in the downstream channels increases. As a result, sediments would become more likely to deposit in river channels, causing aggradation (Pfeiffer et al., 2019). We have rephrased our statement to increase clarity (P17L252):

“With sediment deposition saturated, the trapping efficiency of Amazonian floodplains will decline sharply and higher proportions of sediment will remain in the Amazon River main channel. As deforestation increases sediment supply, the decreased floodplain trapping efficiency would increase the likelihood that the sediment transport capacity of the Amazon River is exceeded. Consequently, more sediments would be deposited in downstream river channels, exacerbating channel aggradation and reducing flood capacity (Pfeiffer et al., 2019; Parrinello and Kondolf, 2021). River reaches may also undergo various geomorphological adjustments, including more rapid channel migration, channel steepening, or avulsion. These processes, which help rivers manage excessive sediment loads, can significantly alter floodplain morphology and hydrodynamics (Popović et al., 2021), but more research is needed to elucidate the impacts of such dynamics. ”

R2C31:

A clarification of the definition of lake vs wetland areas is needed, as it has not really been introduced prior to here. Even though it is briefly mentioned in the methods, to a reader starting from the beginning they will not be able to interpret the difference between the two.

Response:

We appreciate this comment and clarified the definition of lake and wetland areas in the caption of Figure 1a: “The red dashed line outlines the lake area, which is used to estimate the lake-specific POC deposition and the corresponding oxygen exposure time (OET)” and Supplementary Figure S1: “The lake area in Figure 1a is extracted from the permanent water region of the Janauacá

floodplain. Macrophytes are aquatic plants growing near water, while forest refers to a large area densely packed with trees and undergrowth on land. Wetlands consist of Macrophytes and Flooded forest.”. The reference to these figures were also added to the caption of Figure 4: “The lake area and the wetland area is defined in Figure 1a and Supplementary Figure S1, respectively.”

R2C32:

Page 14 Given how small the POC deposition is, wouldn't your study actually suggest that it's not important to include in lake CH₄ models?

Response:

Thank you for the comment. The whole paragraph (P19L310) has been rewritten to improve clarity:

Although sediment deposition in Amazonian floodplains is not strongly tied to regional CH₄ emissions, due to its strong spatial heterogeneity and temporal variations it can influence the spatial and temporal variability of CH₄ emissions within each floodplain. For instance, we find that the POC deposition rate in the lake area is approximately five times higher than that in the wetland area. For the deepest lake area, the POC deposition rate during HW can reach 0.57 g C m⁻² day⁻¹, which becomes comparable to the measured in-lake NPP. With limited oxygen exposure, a substantial fraction of these biologically active POC could be oxidized to CH₄ during the HW period through methanogenesis (Segers, 1998; Sobek et al., 2009). As lake CH₄ models rarely include this carbon source, they would likely fail to capture the related CH₄ emission pulses from floodplain lakes in the Amazon (Tan et al., 2024). Therefore, our results underscore the importance to couple floodplain hydrodynamic, POC deposition, and CH₄ dynamics for more realistic CH₄ modeling in floodplain lakes.

R2C33:

Page 15 How can water depth be zero when the minimum flooded area is 23 km²?

Response:

We apologize for the confusion. A large portion of the Janauacá floodplain is dried during LW, resulting in a much reduced flooded area compared with that in HW. By “water depth”, we intentionally refer to the water depth in our modeled domain of the floodplain. The water depth is zero in the dry area. We rephrased this sentence to avoid confusions (P5L84): “The water stage in Janauacá changes accordingly between below 11 m to 24 m.”

R2C34:

I read through reference [32] and its supplementary material but could not find any TSS/SSC data, even though “TSS” was mentioned several times in that article.

Response:

TSS in Amaral et al. (2018) is equivalent to SSC, which data collected in that sampling campaign is provided by one coauthor who conducted the sampling.

R2C35:

How were E and D determined, and what are their values? Are they physically reasonable?

Response:

The erosion and deposition fluxes (E and D) are internally computed in GAIA (Tassi et al., 2023), which use the established physical formulas in dynamic sediment transport (Wu, 2007). The erosion rate (Eq. 1) is a linear function of the dimensionless excess shear stress and the deposition rate (Eq. 2) is the product of settling velocity (w_s) and the sediment concentration at the interface between suspended load and bedload (C_b , SSC in our case without bedload considered). The information has been added to the revision (P20L338): " w_s is the settling velocity of the sediment particles and C_b is the SSC at the interface between suspended load and bedload, which is the SSC of the suspended sediment in our case since bedload is not included." Given the typical challenge of data availability for validating sedimentation, we compared our modeled deposition rate against the documented values from similar Amazonian floodplains. The consistency in terms of magnitudes implies the modeled erosion and deposition fluxes are reasonable.

R2C36:

Page 16 SRTM data would not provide information in flooded areas of the lake. How were water-covered regions addressed in the bathymetry used to create the model domain?

Response:

Thanks for the comment. This information is detailed in Pinel et al. (2015) that the bias over nonvegetated regions is corrected using ICESat GLAS data. The SRTM correction also included bathymetric data acquired during a June 2012 field trip, when the water level was exceptionally very high (absolute level of 24.3 m). Data were acquired using an Acoustic Doppler Profiler Current (Teledyne RD Instruments, ADCP 1200 Hz) and an echo sounder linked to a GPS station. Other in situ bathymetric data were also acquired in May 2008 and in August 2006. In the revision (P21L345), we have briefly described the bias correction method used for SRTM DEM and refer readers to Pinel et al. (2015) for more detailed description.

"The bias over permanently inundated areas in the SRTM DEM was addressed by using the elevation data from the NASA's Ice, Cloud, and land Elevation Satellite (ICESat) Geoscience Laser Altimeter System (GLAS). The SRTM correction also used bathymetric data from high water levels in June 2012, collected with an ADCP and GPS-linked echo sounder, along with sampling from May 2008 and August 2006. Meanwhile, vegetation-induced biases were corrected using a wetland map from Hess et al. (2015b) and a MODIS-derived vegetation height dataset from Simard et al. (2011). These corrected elevations were integrated using the ANUDEM v5.3 algorithm, constrained by a drainage channel network identified from Landsat imagery. For a detailed methodology, see Pinel et al. (2015)."

R2C37:

You mention wind being important to SSC at low water. From what I can tell by looking

at [45], wind is not implemented in the model being used here. Is that correct? If not, this seems like it could be an important missing component of the model.

Response:

The reviewer is correct in observing that our current model does not explicitly incorporate wind effects. This decision was based on the limitations of the available wind data provided by the Brazilian national meteorological institute (INMET) (<https://mapas.inmet.gov.br/>). Although the nearest meteorological station is at Manacapuru, the wind is measured at a height of 10 meters above land. This height does not accurately reflect wind conditions at the water surface of a nearby floodplain, especially given the influence of the canopy and other local geographical features. Implementing a spatially uniform wind speed, derived from this data, could introduce significant biases and uncertainties in a model focused on the smaller scale dynamics of a floodplain. Moreover, after re-evaluating the referenced article (Moreira-Turcq et al., 2004), we found it may not be an appropriate reference because this work is a geologic scale study that is not able to evaluate the effects of wind on sediments. We removed this reference from the revision.

Finally, our primary aim is to ensure that the hydrodynamic simulation reliably captured flow speed and water level, which has already been validated in the previous hydrodynamic simulation (Pinel et al., 2020). Thus, wind was not included in our simulation setup. We appreciate this point and has clarified in the revised manuscript the reasons for excluding wind from the model (P22L367).

“Wind was excluded from the simulation due to the unavailability of vegetative sub-canopy wind, which is essential for accurate modeling in this context.”

R2C38:

Clarify: the WLBCs do not connect to the mainstem

Response:

The three water level open boundaries (WLBC) are connected to the mainstem of Amazon River. This is clarified in the model description (P21L356) and in the caption of Figure 1a (previous Figure S1): “The domain includes 8 inflow boundaries (BCF) and 3 water level boundaries (WLBC), the latter of which connect to the mainstem of the Amazon River (Figure 1a).”

R2C39:

Page 17 It is also possible that the SSC at Manacapuru is not the same as the SSC at the WLBCs. Can you constrain the uncertainty here?

Response:

We recognize that the spatial differences between the gauge and the model's boundary that could introduce uncertainties into our simulation. In the map of Figure 1a (previous Figure S1), we marked the Manacapuru station to demonstrate its distance relative to the model's open boundary. The water level at the WLBCs is bias-corrected, using the difference in bed elevation from the Manacapuru gauge. This information is added to Page 16.

Furthermore, we have added to our manuscript a note about the geographical layout of the river system relevant to our study site (P22L383): “There are no significant tributaries between the Manacapuru gauge and the WLBCs that would alter the sediment load along this stretch of the river.” This geographical fact supports our use of Manacapuru SSC data as a reasonable approximation for the boundary conditions.

Additionally, the SSC derived from satellite data implies that there is no significant difference between SSC levels at Manacapuru and at WLBCs (P22L385): “The satellite data (Fassoni-Andrade and de Paiva, 2019) also implies no significant difference between SSC at Manacapuru and at WLBCs (Supplementary Figure S17).”

Figure S17: Time series of suspended sediment concentration (SSC) over the simulation period (2007-2016) at Manacapuru (blue) and at the Janauacá's open boundary (red). The SSC is extracted from the MODIS data processed in Fassoni-Andrade and de Paiva (2019).

Finally, given the challenge in in-situ sediment sampling and limited samples available, we believe that the attempt of further constraining this uncertainty may only introduce additional layers of uncertainty. In the revision, we acknowledged the existence of such uncertainty in P22L386: “However, due to the lack of direct SSC measurements at WLBCs, there is an inherent uncertainty in the SSC boundary condition.”

R2C40:

Figure S1, S2 Why is the extent of open lake in S1 different from the land cover type in S2?

Response:

Figure S2 (now Figure S1) describes the topological classification of the computational domain, which is used to (a) define the Manning's roughness coefficients in the separate regions (Page 16) and (b) delineate the wetland area to estimate the corresponding POC deposition (Figure 4). The open lake in Figure 1a (previous Figure S1) is extracted from the permanent water region in Figure S2 (now Figure S1) to reflect the lake zone within the Janauacá floodplain, which is used to estimate the lake POC deposition and the corresponding oxygen exposure time (OET) (Page 14).

In the revision, we further clarified the definition of the open lake in the caption of Figure 4: “The lake area and the wetland area are defined in Figure 1a and Supplementary S1, respectively.”, in

the caption of Figure 1a: “The red dashed line outlines the lake area, which is used to estimate the lake POC deposition and the corresponding oxygen exposure time (OET).”, and in the caption of Figure S1: “The lake area in Figure 1a is extracted from the permanent water region of the Janauacá floodplain.”

R2C41:

Figure S3 Water level is okay in (b), but water discharge (and also a plot of sediment flux) would be a lot more relevant here.

Response:

Figure S2 (previous Figure S3) is used to demonstrate the boundary conditions of both water and sediment that drive the simulation, while water and sediment fluxes are computed in the simulation where the first year is taken as model spin-up. We followed the reviewer suggestion and created a new figure (Figure S9) that illustrates the net water and sediment flux at the model’s boundary corresponding to Figure 2a and 2b, respectively.

Figure S9: Time series of monthly exchanging rate of flow and suspended sediment through the open boundaries (transects in Figure 1a). The positive and negative values represent the flux into and out of the Janauacá floodplain, respectively.

R2C42:

Figure S6 It would be very useful to have the conditions at Manacapuru on these figures since the main-stem Amazon is so important to the dynamics within the lake.

Response:

Thanks for the reviewer comment. In the revised Figure S5, we added the discharge and sediment flux at the Manacapuru gauge to panel c and d, respectively.

Figure S5: (a) The daily averaged flux of exchanging **water** flow between Lake Janauacá and the Amazon River mainstem at the three open boundaries WLBC1, WLBC2 and WLBC3 (Fig. 1a), their net total (**black**) and the total input from 8 upstream inflow (BCF) boundaries (**red**). (b) The daily averaged SSC at the WLBCs. (c) and (d) are the daily averaged discharge and sediment flux, respectively, at the Manacapuru gauge. The positive and negative values represent the flux into and out of the Janauacá floodplain, respectively. Light shades indicate the variation over multiple simulation years. The channelized flow and overflow are delineated at WLBC1 and WLBC2/WLBC3, respectively, where the bottom elevation at WLBC1 is below 12 m and at WLBC2 and WLBC3, it is over 20 m. From the late LW period to the mid-RW, channelized flow predominantly influences water and sediment exchange at WLBC1, transitioning to overflow in WLBC2 and WLBC3 when the water level exceeds the levee crest.

R2C43:

Figures in general I was not able to find the location of Manacapuru on any of the figures. Please add.

Response:

We added the location of the Manacapuru gauge to the inset of Figure 1a (previous Figure S1).

Reviewer 3

This manuscript uses a numerical model that couples 2D shallow-water hydrodynamics with cohesive sediment entrainment and deposition to predict how much sediment is deposited

on the floodplain of the Amazon River near Lake Janauacá. The results indicate that most suspended sediment comes from the Amazon River and flows through a secondary channel before being deposited in the lake, and that there is a 4-month delay between peak floods in the Amazon and peak sediment deposition. In addition, the results indicate that a small fraction of particulate OC is deposited as part of SSC, but it is unclear if this has a significant effect on regional biogeochemical cycling since NPP by algae in the lake greatly outpaces POC deposition.

Overall, I found the model results interesting, but I did not think the manuscript described the modeling in sufficient detail to fully interpret the results. This study builds on significant previous work at the site, but the manuscript does not stand on its own as written. The introduction did not lay out clear knowledge gaps or testable hypotheses for the study, so it is not clear to me how significant or transferable the results are to other floodplains. Predicting fine-scale patterns of sedimentary deposition is at the frontier of modeling for river floodplains, but the paper requires significant rewriting to make this study's contribution and broader impacts clear. Given these considerations, I recommend this manuscript for major revisions followed by re-review.

Response:

We sincerely thank the reviewer for the feedback and comments. We have carefully addressed the reviewer's recommendations as follows:

R3C1:

General comments:

1) *The paper needs significant revisions to the introduction and results so that the reader can fully understand the study. Currently, the motivations for the study, particularly the connection between sediment transport and floodplain outgassing, are not thoroughly described. It is also not clear why the authors selected this site as representative of the Amazon floodplain or how the numerical model works. The journal format has the methods presented at the end, so it is important to help the reader understand the results without reading the methods or referring to the supplement. In particular, this study builds on multiple prior studies on this site from the same research group, which should be summarized in the introduction to instill confidence in the field measurements and modeling presented in this paper and allow it to stand alone.*

2) *The model produced an interesting result, where increasing SSC 40% above the base case caused additional floodplain incision and decreased sediment storage. This is a surprising result, and it would be worth digging into the mechanics of processes that define this threshold value to understand if this result can be generalized beyond the study site. This requires a more detailed discussion of spatial and temporal variability in floodplain sediment entrainment and deposition rates as well as some sensitivity or stability analysis to ensure the model is making reasonable predictions.*

Response:

Thanks for the reviewer comment. We have revised the introduction to better articulate our research motivation and to more thoroughly explain the link between sediment transport and

floodplain outgassing. Additionally, we moved the background and study domain descriptions from the Methods section to the introduction to provide a clearer context based on earlier studies. Please refer to our response to R3C3, R3C4, R3C5, R3C8, R3C15, R3C20 and R3C25.

Moreover, we have performed additional analyses on the spatiotemporal variability in the Janauacá floodplain under different scenarios. The new analyses imply that deforestation-induced increases in SSC levels in the Amazon River lead to a nonlinear rise in deposition rates, and this increasing rate diminishes over time, reflecting the complex interplay of sediment deposition, floodplain bed evolution, and hydrodynamics.

In the updated Figure 3c, instead of using box plots to represent mean and standard deviation across multiple years, we explicitly show the relationship for each simulation year, which highlights the increasing nonlinearity over time. This nonlinear change is closely related to the floodplain bed evolution.

We've also added Figure S11 in the Supplementary materials, showing significant changes in bed elevation due to sediment deposition, particularly near the domain's open boundary and in deeper water regions where more deposition occurs. Comparing the simulation results between 2008 and 2016, we found a clear difference in deposition rate and the exchange rate of flow and sediment (Figure S12). In particular, even with only boundary SSC perturbed, we observed alterations in flow flux magnitude and timing at varying SSC levels in 2016 (Figure S12d), impacting sediment flux (Figure S12f).

Moreover, the spatial distribution $D_{exp}/D_{baseline}$ further supports our findings, particularly in deeper regions where bed evolution is significant (Figure S12). In these regions, $D_{exp}/D_{baseline}$ aligned with $SSC_{exp}/SSC_{baseline}$ in 2008 (Figure S13a). However, in 2016, this consistency decreased, especially at higher SSC levels in Figure S13b, where the $D_{exp}/D_{baseline}$ typically fell below the corresponding $SSC_{exp}/SSC_{baseline}$.

In response to this comment, we slightly reframed our statement in the abstract: "Our model experiment further demonstrates that widespread deforestation would reduce the trapping efficiency of the floodplains **over time**, exacerbating downstream river aggradation."

The result section was rephrased to improve the accuracy of the description (P11L164):

"However, if the SSC level in the Amazon River increases substantially due to deforestation, the gain of sediment deposition in the floodplain would gradually lose to the surge of river sediment influx over time (Figure 3c and Supplementary Figure S15), causing higher fractions of river sediment to transport downstream instead of being deposited in the floodplain. Our analysis shows that this reduction of the sediment retention in the floodplain is a result of floodplain bed evolution (Supplementary Figure S11~S13). Under the extreme scenario of agricultural expansion (60% increase in SSC), the trapping efficiency of the floodplain is predicted to decline by 7% by the tenth year of the simulation, as the floodplain elevation near its open boundary rises by more than 0.5 m (Supplementary Figure S11). It is worth noting that the human-induced alteration of SSC levels will make little changes to the seasonal and inter-annual variabilities of sediment deposition and sediment exchange flux (Figure 3a and 3b and Supplementary Figure S10b), although their seasonal variations are intensified. The seasonal change of SSC implies a more dynamic response of the deposition rate, with shifts in SSC peaks causing notable increases

in sediment influx during HW (Supplementary Figure S14a, c and e).”

The discussion section was also elaborated to incorporate the new findings (P17L246):

“The surge of river SSC due to the dramatic increase of soil erosion (e.g., over 40%) (Riquetti et al., 2023) will overwhelm the capacity of the floodplains to trap sediment. This is because the progressive increase of floodplain bed elevation, especially in deeper water areas, would affect the magnitude and timing of flow and sediment exchanges, as well as the spatial distribution of deposition adjusted (Supplementary Figure S12 and S13), hindering a linear increase of deposition rates.”

Figure 3: Monthly variations of the simulated daily mean deposition rate (a) and **daily mean suspended sediment flux through the open boundaries** (b) on different SSC perturbation levels. (c) The change ratio of the simulated annual mean sediment deposition rate on different SSC perturbation levels **for each simulation year**. In (a) and (b), the baseline is the simulation without SSC perturbation and different percentages represent the SSC perturbation levels in Amazon River. In (c), D_{exp} is the simulated sediment deposition rate in the perturbed SSC simulations (SSC_{exp}) and $D_{baseline}$ is the simulated sediment deposition rate without SSC perturbation ($SSC_{baseline}$). **For each year in (c), the simulations with reduced SSC are fitted using a linear relationship, and the simulations with increased SSC are fitted with second-order polynomial functions. The number after each labeled year is the coefficient of the second-order terms that measures the nonlinearity.**

Figure S11: The difference of the total change of floodplain bed elevation from 2008 to 2016 between an experiment simulation and the baseline simulation (ΔBed elevation).

Figure S12: Time series of deposition rate in 2008 (a) and in 2016 (b) on different SSC perturbation levels. The corresponding exchanging rate of flow and sediment in 2008 (c and e) and in 2016 (d and f). Comparing the simulation results between 2008 and 2016, there is a clear difference in deposition rate and the exchange rate of flow and sediment. Even with only boundary SSC perturbed, alterations are observed in flow flux magnitude and timing at varying SSC levels in 2016 (d), impacting sediment flux (f).

Figure S13: Spatial distribution of $D_{exp}/D_{baseline}$ in 2008 (a) and 2016 (b). In deeper regions near the floodplain's open boundary, $D_{exp}/D_{baseline}$ aligned with $SSC_{exp}/SSC_{baseline}$ in 2008 (a). However, in 2016, this consistency decreased, especially at higher SSC levels in (b), where the $D_{exp}/D_{baseline}$ typically fell below the corresponding $SSC_{exp}/SSC_{baseline}$.

To demonstrate the robustness of our results, We also performed additional numerical experiments that represent the change of the seasonal variations in SSC due to human activities. The design of the new experiments is described in the Method section (P25L452): “While it is not entirely clear how anthropogenic disturbances affect the seasonal variations in SSC, these potential impacts were assessed through experiments, including reducing variation amplitude by 60%, and enhancing the SSC magnitude by 60% during peak periods. Although the timing of the SSC peak is influenced by climatic rather than anthropogenic forcing, supplementary experiments were conducted by advancing the SSC peaks 30, 60, and 90 days towards the water level peak to assess how interactions between climate and human activities might affect the sediment deposition in Amazonian floodplains.”

Consistent with our original sensitivity experiment which increases SSC uniformly across seasons by 60%, the new experiment that increases SSC by 60% over the peak period would also cause the decrease of the trapping efficiency of the floodplain (Figure S15). This consistency enhances our conclusion that a substantial increase of SSC due to deforestation would cause the reduction of the floodplain's capability to trap sediments. Additionally, we also conducted several numerical experiments that explore the interactions between the impacts of climate change and human disturbance. In these experiments, we shifted the SSC peaks 30, 60, and 90 days towards the water level peak.

These experiments clearly show the substantial influence of seasonal variation in SSC and the dynamic response of sediment deposition rate to multiple factors (Figure S14), reinforcing our arguments presented in R2C2. Notably, temporal shifts in SSC peaks led to significant changes in sediment influx, particularly when peaks coincide with the RW period, resulting in higher deposition rates, as shown in Figure S14e. Conversely, smoothing the SSC peak by 60% did not reduce the deposition rate as much as reducing the entire SSC time series by the same percentage (Figure S14b). This occurs because the smoothed SSC maintains higher values during both RW and HW compared to the baseline scenario. Moreover, we found that increasing SSC by 60% during its peak days (Figure S14d) does not lead to as high a deposition rate as increasing the entire time series by 60% (Figure S14b). This is because the SSC peak period slightly shifts among the simulation years, leading to a lower overall sediment flux and deposition rate (Figures S14b and f).

In the revised manuscript, the results of the new experiments are briefly referenced in the main context at P12L177: “The seasonal change of SSC implies a more dynamic response of the deposition rate, with shifts in SSC peaks causing notable increases in sediment influx during HW (Supplementary Figure S14a, c and e).” More details were reported in the caption Supplementary Figure S14 and S15.

Figure S14: The daily averaged deposition rate (a), suspended sediment concentration (SSC) at the WLBCs (c) and sediment flux at the WLBCs (e) of the baseline simulation and the experiments that shift the SSC by 30 days, 60 days and 90 days towards the HW period. The daily averaged deposition rate (b), suspended sediment concentration (SSC) at the WLBCs (d) and sediment flux at the WLBCs (f) of the baseline simulation and the experiments that increases the SSC peak by 60% (+60% peak), increases the SSC by 60% (+60%), smooths the SSC variation by 60% (-60% smooth) and smooths the SSC by 60%. The number in front of each experiment name in the labels is the corresponding deposition rate. These experiments show the substantial influence of intraannual variation in SSC and the dynamic response of sediment deposition rate. Notably, temporal shifts in SSC peaks led to significant changes in sediment influx, particularly when peaks coincide with the RW period, resulting in higher deposition rates, as shown in (e). Conversely, smoothing the SSC peak by 60% did not reduce the deposition rate as much as reducing the entire SSC time series by the same percentage (b). This occurs because the smoothed SSC maintains higher values during both RW and HW compared to the baseline scenario. Moreover, we found that increasing SSC by 60% during its peak days (d) does not lead to as high a deposition rate as increasing the entire time series by 60% (b). This is because the SSC peak period slightly shifts among the simulation years, leading to a lower overall sediment flux and deposition rate (b and f).

Figure S15: Annual averaged sediment trapping ratio in 2016 of the baseline simulation and the experiments that increases the SSC peak by 60% (+60% peak), increases the SSC by 60% (+60%), smooths the SSC variation by 60% (-60% smooth) and smooths the SSC by 60%.

R3C2:

Abstract: “Biogeochemical cycling” is mentioned but the abstract needs more of your quantitative results for these processes. Please clarify the relevant state of the fluxes (particulate, dissolved) and the spatiotemporal scales (regional? 100 years?) presented in the results.

Response:

Thanks for the reviewer comment. The importance of sediment deposition to biogeochemical cycling is evaluated by comparing the deposition of sediment-associated organic carbon (particulate organic carbon) with the soil carbon input through ecosystem primary production. Because POC deposition is one to two orders of magnitude lower than NPP at the annual scale for the whole floodplain, our results actually show that POC deposition only plays a minor role in regional CO₂ and CH₄ emissions. We apologize for the confusion due to our vague presentation. In the revised abstract, we rephrased our statement:

“Additionally, we show that the deposition of sediment-associated organic carbon only plays a minor role in fueling carbon dioxide and methane emissions from Amazonian floodplains.”

However, it should be noted that POC deposition can play an important role in the CH₄ emissions of floodplain lakes in the Amazon during HW. Please refer to our response to R3C16 for more detailed discussions.

R3C3:

Page 2, “fueling emissions of carbon dioxide (CO₂) and methane (CH₄) [12, 13].”: The introduction needs to include more prior work on floodplain outgassing and clarify that you focus on particulate OC. Rivers, wetlands, and dry floodplain soils can all release CO₂ and CH₄ but this can come from oxidation of POC or DOC. For instance, I would expect DOC to be more readily emitted as CO₂ or CH₄ from rivers and wetlands while POC should track

with fine sediment deposition and is more likely to be sequestered in floodplains and avoid oxidation.

Response:

We agree that our initial discussion did not sufficiently address the nuances of how POC and DOC contribute differently to outgassing in various parts of the river-floodplain system. In response, we have emphasized our focus on POC, particularly in relation to its transformation pathways due to oxygen exposure levels (P4L56).

“Current measures of sediment-associated particulate organic carbon (POC) deposition also lack sufficient spatial and temporal details. In particular, they are too coarse to determine which landscape and oxic condition POC is dominantly deposited in Amazonian floodplains (McClain and Naiman, 2008), and this is a determinant factor for the transformation pathways of POC in floodplain soils (Ward et al., 2013; Sawakuchi et al., 2016).”

R3C4:

Page 3, “accurate estimation of sediment dynamics in Amazonian floodplains with sufficient spatial and temporal details still remains challenging.”: It is not clear to me what spatial and temporal resolution you require to understand biogeochemical cycling and what specific knowledge gap this study hopes to fill.

Response:

Thanks for the reviewer comment. Our study emphasizes the importance of high-resolution modeling to capture the dynamic processes of sediment deposition within Amazonian floodplains and provides implications for associated biogeochemical cycling.

While the existing network of gauge stations and remote sensing observations provides essential insights into the sediment budget of the Amazon River, these methods are often limited at capturing the fine-scale spatial and temporal variations that are critical for understanding sediment dynamics in the floodplain environment. For instance, gauge stations are typically spaced widely and provide discontinuous and even sparse time series data. Similarly, the temporal resolution of remote sensing data, often limited by cloud cover and the satellite pass schedule, may not adequately capture rapid changes during flooding events.

Our study seeks to address these gaps by employing a high-resolution hydrodynamic and sediment transport model that can simulate sediment dynamics over fine temporal scales (e.g., daily) and spatial scales (e.g., meters). This level of detail is crucial to accurately assess the the drivers and impacts of sediment deposition in Amazonian floodplains. By filling these gaps, our research aims to provide a more detailed understanding of the mechanisms driving sediment distribution and its ecological impacts.

In the revised introduction (P4L43), we rewrote this paragraph to elaborate the knowledge gap this study aims to fill:

“The sediment budget of the Amazon River is well measured via a network of gauge stations on the mainstream and major tributaries (Martinez et al., 2009) and remote sensing observations (Espinoza-Villar et al., 2018), but accurate representation of sediment dynamics within Amazonian floodplains remains challenging. Both in-situ data and estimates of SSC derived

from remote sensing imagery have limited ability to resolve the processes in sufficient spatial and temporal resolution. For example, though the underlying drivers of sediment dynamics, such as inundation, fluctuate usually on a daily basis and vary significantly in space across the complex landscapes at a 10-m scale (Pinel et al., 2020), in-situ sediment sampling was only occasionally conducted in specific floodplain locations for short time periods (Amaral et al., 2018; Smith et al., 2003). Meanwhile, despite the broad coverage from remote sensing technologies over the Amazon Basin floodplains (Villar et al., 2013; Martinez et al., 2015; Espinoza-Villar et al., 2018), the data resolution remains coarse and its uncertainty is often high due to the empirical nature of the methodology, signal degradation by cloud cover, macrophytes, and shallow sediment beds (de Moraes Novo et al., 2006; Montanher et al., 2014; Fassoni-Andrade and de Paiva, 2019; Rudorff et al., 2018). Current measures of sediment-associated particulate organic carbon (POC) deposition also lack sufficient spatial and temporal details. In particular, they are too coarse to determine which landscape and oxic condition POC is dominantly deposited in Amazonian floodplains (McClain and Naiman, 2008), and this is a determinant factor for the transformation pathways of POC in floodplain soils (Ward et al., 2013; Sawakuchi et al., 2016)."

R3C5:

Page 4, "we used a recently developed and well validated model": It would be very helpful to give more information about the model you used and where it has been validated. Stating here that the model was already validating for hydraulics at this field site is important since the reader does not get that information until the Methods.

Response:

Thanks for the reviewer comment. In the revised manuscript, we have moved the description of study site to the introduction and rephrased this sentence (P4L69) to: "Here, we used a recently developed hydrodynamic model, now coupled with sediment dynamics, to simulate high-resolution sediment deposition", "The hydrodynamic model has been rigorously validated in representing floodplain hydrodynamics, including both water stage and flow current (Pinel et al., 2020)."

R3C6:

Page 4, "the simulated water level in the floodplain follows four hydrological periods": I was quite confused by the extent of the floodplain boundary versus the model domain, which should be defined and consistent with maps in the supplement.

Response:

Sorry for the confusion. The computational domain of the Janauacá floodplain is delineated as the floodable region with an elevation lower than 29 m, which is defined in the Method section (P21L354). In this case, the floodplain boundary is consistent to the model domain in Figure 1a (previous Figure S1).

However, we understand the floodplain boundary may be defined differently elsewhere. To avoid confusion, we rephrased to "the simulated water level in our computational domain" (P6L103).

R3C7:

Page 6, “the mean annual deposition rate per unit length of Amazon River, of 0.033 Mt km⁻¹ yr⁻¹”: What is the reference for this value?

Response:

This value is the modeled deposition rate per unit length. We rephrased this sentence (P8L120) to improve the clarity: “or equivalently 0.033 Mt km⁻¹ yr⁻¹ of sediment was deposited per unit river length as the adjacent boundary of Janauacá with the Amazon River is about 5 km long.”

R3C8:

Page 6, “years of extremely low water levels in 2010 and 2011”: It is unclear why the years evaluated in this paper were chosen. Is this limited by SSC data availability? Are these years anomalous due to storms or climate cyclicity? If satellite measurements are available that match field sampling well, why not examine sediment transport over a longer period of time to have more statistically robust results?

Response:

This simulation period encompasses one drought event (year 2010), flood events and normal hydrological conditions. There are in-situ flow data samples for previous hydrological and hydrodynamic simulations (Bonnet et al., 2017; Pinel et al., 2020), and SSC data samples during 2015 and 2016 (Amaral et al., 2018) for sediment model validation in this study. The latter is extremely valuable and limited elsewhere. We agree with the reviewer that a longer-period simulation may provide more statistically robust results. However, the runoff boundary condition from the upland local watershed is provided by the LUMP-FP model (Bonnet et al., 2017), which only provides the data for our simulation period.

R3C9:

Page 6, “the deposition rate reached elevated values”: To get from sediment concentration to sediment deposition rates in a specific you multiply the SSC by an average settling velocity. Grain size and SSC both vary with depth in the channel, so how does your model account for this? Does the SSC from your datasets represent a surface value or do you do some normalization based on flow depth and shear stress?

Response:

In the 2D simulation, we assume a depth-integrated SSC value and do not account for the vertical concentration distribution as would have been done in the 3D simulation using a Rouse profile, because the Janauacá floodplain is much shallower than the Amazon River mainstem. Even in the deeper lake area during HW (outlined in Figure 1a), the water depth is ~10 m, which reduces to <5 m during LW (Figure S3). While SSC within Janauacá (Amaral et al., 2018) and at Manacapuru (Martinez et al., 2009) are both collected near the surface, there are little variations in the shallower regions (~10 m) of the water column in the Amazon River mainstem according to Rudorff et al. (2018) and Bouchez et al. (2011). In the revision (P22L372), we first justified the assumption of depth-integrated SSC in our simulations:

“Our simulation considers depth-integrated SSC, as the water depth in the Janauacá floodplain is typically much shallower than that in the Amazon River mainstem. Even in the deeper lake areas, peak water depths reach about 10 m during HW and reduce to less than 5 m during LW (Figure S3), with SSC showing minimal variation across the shallower water column (Rudorff et al., 2018; Bouchez et al., 2011).”

We then clarified the SSC samples are collected near the water surface (P22L377, P24L424):

“The daily SSC at the water level boundaries (Supplementary Figure S2a) was derived from the near-surface observation at Manacapuru from the ORE-HYBAM database”

“modeled sediment dynamics are validated using the SSC in-situ samples collected at three sites, which are all 0.5 m below the water surface, within Janauacá.”

R3C10:

Page 6, “deep water area of Janauaca Lake”: What are the constraints on lake bathymetry? The model appears to assume it is constant.

Response:

TELEMAC-2D solves depth-averaged 2-dimensional (2-D) shallow water equations, which accounts for the variation in lake bathymetry. The sediment deposition starts increasing as a substantial amount of sediment is supplied during RW (Figure S4b) and occurs only in the deep water area during HW as a result of the slowing of water momentum (Figure 2f). The deep water area is shown in Supplement Figure S3c and the change in momentum is provided in Supplement Figure S6c. The bathymetry data uses the SRTM DEM that is bias corrected for the deep water area. Please see our response to R3C26 for more details on the bathymetry.

In the revision (P9L138), we rephrased the statement to increase the clarity: “Particularly, the model shows that the rising water supplied a dominant amount of sediments to the floodplain (Supplementary Figure S4b). These intruded sediments were only deposited to the deep water area of Janauacá Lake during the subsequent HW, several months after RW (Figure 2f), when the slowing of water movement favored sediment deposition (Supplementary Figure S6c).”

R3C11:

Page 6, “For 0.34 Mt yr⁻¹ of the mean annual sediment influx (Figure 2a), 11.7±7.2% was retained”: Where does the rest of the sediment go? Does it bypass the floodplain and return to the Amazon River or does it flow upstream on tributaries?

Response:

Thanks to the reviewer comment. The extent of the sediment transport is limited to the low-lying regions of the floodplain (Supplement Figure S4b), so the rest of the sediment does not flow upstream on tributaries and is instead returned to Amazon River during FW. This explanation was added to the revision (P9L150): “Of the 0.34 Mt mean annual sediment influx (Figure 8a),

11.7±7.2% was retained in the floodplain and the rest was returned to the Amazon River during FW (Figure 8b)”.

R3C12:

Page 7: I would appreciate a paragraph that includes discussion of the spatial and temporal patterns in SSC and deposition and discusses causes of their variability in detail. I feel like the results gloss over this and just discuss parameters calculated for the whole floodplain, like the sediment trapping ratio.

Response:

Thanks for the reviewer comment. In the original manuscript, one paragraph outlined the spatiotemporal patterns of SSC and deposition. In this revision, we have elaborated these descriptions to provide a detailed spatial and temporal analysis (P9L131-146):

“Due to the effect of floodplain hydrodynamics, the simulated sediment deposition shows large and distinct spatial variability during the four hydrological periods (Figures 2d-2g). Unlike the modeled SSC that topped during RW, the deposition rate reached elevated values of $>2 \text{ kg m}^{-2} \text{ yr}^{-1}$ in both the RW and HW periods (Figures 2e and 2f). Between these two periods, the sediment deposition during HW is more widespread to the lake and wetland units in the floodplain (Figures 2e and 2f), implying its higher relevance to the sediment and carbon budgets of these landscape components. Particularly, the model shows that the rising water supplied a dominant amount of sediment to the floodplain (Supplementary Figure S4b). The intruded sediment was only deposited to the deep water area of Janauacá Lake during the subsequent HW, several months after RW (Figure 2f), when the slowing of water movement favors sediment deposition (Supplementary Figure S6c). In regions adjacent to WLBC3 where flows are persistently fast during RW and HW (Supplementary Figure S6b and S6c), the deposition rates are reduced due to continuous sediment resuspension and transport (Figure 2e and 2f). Overall, the annual sediment deposition can exceed $5 \text{ kg m}^{-2} \text{ yr}^{-1}$ in the deep water lake.”

Given that the sediment trapping ratio is derived directly from the deposition rates, the spatial pattern of the sediment trapping ratio would inherently mirror the deposition rate patterns already presented in Figure 2. More importantly, the sediment trapping ratio is intentionally computed as a gross measure, representing the overall trapping capacity of the floodplain. This ratio is calculated by comparing the total amount of sediment deposited within the floodplain to sediment load of Amazon River, as measured at Manacapuru (new Supplementary Figure S8). This approach is intended to provide a comprehensive overview of the floodplain’s capacity to retain sediment, highlighting its role as a critical sediment sink within the larger riverine ecosystem.

R3C13:

Page 8, “except in the case of elevating SSC by 60% in which excessive erosion would occur during HW”: I don’t understand why elevating SSC would increase erosion. If input SSC is higher wouldn’t that cause increased deposition? This is an interesting and non-intuitive result that deserves more discussion so the reader can understand this effect.

Response:

We apologize for any confusion caused by our earlier descriptions. Erosion primarily occurred

near WLBC3 and along the channels connecting to WLBC1, as illustrated in Supplementary Figure S11. Notably, in scenarios with elevated SSC, increased erosion during the FW period was observed, driven by exchange fluxes that resuspend deposited sediment. This phenomenon is reflected by the negative values in the deposition rate shown in Figure 3a.

Additionally, we have revisited the discussion concerning “excessive erosion” in the case of elevating SSC by 60%. It turns out that the simulated “excessive erosion” is an artifact because the related simulations were conducted in a different machine. Originally, two simulations (increases of 50% and 60% SSC shown) were unintentionally conducted on a different machine than the rest simulations. This resulted in minor discrepancies, potentially due to external library/compiler differences that could affect the stability of sediment transport simulations. We have corrected this by rerunning these simulations on a uniform platform to ensure consistent experimental conditions. The re-evaluation has verified that the discrepancies were minor and do not affect the main conclusions of our study. This discussion has been removed from the revised manuscript.

R3C14:

Page 9, “measured in-lake net planktonic primary production (NPP)”: It is worth separating out different types of carbon. SSC can contain petrogenic OC, which is more recalcitrant and could be important for long-term OC storage in floodplain deposits, while the biospheric OC produced by NPP could be more labile.

Response:

Thank you for the comment. In the revision, we have rephrased our statement and clarified the scale of POC deposition relative to NPP, subtly emphasizing its presence without overstating its comparative significance. Please see our more detailed response to R3C16.

We also removed the corresponding discussion here: “One explanation for the low POC deposition is the labile nature of the carbon fixed as NPP by planktonic algae and herbaceous plants, that are mainly decomposed within the floodplain, as is illustrated by a carbon budget computation in a nearby floodplain lake that shows that the majority of the NPP is recycled back as CO₂ to the atmosphere”.

R3C15:

Top of page 10: This paragraph should be motivated in the introduction. Stating that you want to determine if sediment POC deposition matters in floodplains and particularly in lakes would help to focus the introductory paragraphs.

Response:

The motivation is added to the revised introduction (P4L56): “Current measures of sediment-associated particulate organic carbon (POC) deposition also lack sufficient spatial and temporal details. In particular, they are too coarse to determine which landscape and oxic condition POC is dominantly deposited in Amazonian floodplains (McClain and Naiman, 2008), and this is a determinant factor for the transformation pathways of POC in floodplain soils (Ward et al., 2013; Sawakuchi et al., 2016).”

R3C16:

Page 10, “POC deposition at the Januaca floodplain is also considerably smaller than the derived NPP”: This paragraph makes it sound like the POC deposition flux doesn’t matter. It’s important to be very clear if it does, and if so, over what time and spatial scales.

Response:

We apologize for any confusion caused by the tone and presentation of our findings regarding POC deposition in relation to methane (CH₄) models in our manuscript. We acknowledge that our original statement may have conveyed an unintended emphasis on the significance of POC deposition. It is indeed true that our results show the minor role of POC deposition in regional CO₂ and CH₄ emissions when compared to the soil carbon input from ecosystem primary production.

Our intent is not only to report on the quantities of POC but also to highlight its ecological and biogeochemical relevance, providing a comprehensive understanding that supports future research and modeling efforts in similar settings. In particular, this study highlights that due to its strong heterogeneity, the impact of sediment deposition on carbon cycling varies considerably between lakes and wetlands and among different hydrological periods. To improve clarity and avoid future confusion, we have revised our manuscript to better contextualize the role of sediment-associated POC within the larger carbon cycle and its implications for methane emissions modeling.

In the revision, we first rephrased our statement and clarified the scale of POC deposition relative to NPP, subtly emphasizing its presence without overstating its comparative significance:

In the abstract, “Additionally, sediment deposition imports a large amount of biologically active carbon from Amazon River to Januacá, albeit not comparable to carbon fixation from ecosystem primary production. As experiencing short oxygen exposure, the deposited carbon would likely be sequestered or fuel methane emissions.” is rephrased to “**Additionally, we show that the deposition of sediment-associated organic carbon plays a minor role in fueling carbon dioxide and methane emissions from Amazonian floodplains.**”

In the result section (P14L181), “Sediment deposition in the floodplain is shown to import a large amount of biologically active particulate organic carbon (POC) from Amazon River to the lake sediment in Januacá” is rephrased to “**With sediment deposition, considerable biologically active POC from the Amazon River was also deposited in the floodplain. However, quantitatively, the deposited POC per year is markedly lower than the annual soil organic carbon input from ecosystem primary production (Figure 4a, b).**”

The discussion section (P19L296-P20L322) has also been rewritten to improve clarity and link to a larger conceptual framework:

Amazonian floodplains play an important role in the global CO₂ and CH₄ cycles (Murguía-Flores et al., 2023). Due to the enormous amount of sediment deposition in the floodplains, it is reasonable to speculate that the deposition of sediment-associated POC could be important for the regional CO₂ and CH₄ emissions (Saunois et al., 2019; Guilhen et al., 2020). However, our results do not support this hypothesis. Compared to the wetland soil and lake sediment carbon that are sourced from ecosystem primary production, the soil/sediment carbon input due to POC

deposition is considerably smaller. Therefore, POC deposition in Amazon floodplains is unlikely to contribute to the high CO₂ and CH₄ emissions observed in the region, despite its crucial role in the supply of critical nutrients to the floodplain soils (Petsch et al., 2023). Furthermore, because Amazonian floodplains are one of several sediment deposition hotspots in the world, we suspect that POC deposition also only plays a minor role in fueling CO₂ and CH₄ emissions in other productive floodplains across the globe. However, for low-productivity floodplains in high latitudes, fluvial POC deposition remains important for regional CO₂ and CH₄ emissions (Herbst et al., 2024).

Although sediment deposition in Amazonian floodplains is not strongly tied to regional CH₄ emissions, due to its strong spatial heterogeneity and temporal variations it can influence the spatial and temporal variability of CH₄ emissions within each floodplain. For instance, we find that the POC deposition rate in the lake area is approximately five times higher than that in the wetland area. For the deepest lake area, the POC deposition rate during HW can reach 0.57 g C m⁻² day⁻¹, which becomes comparable to the measured in-lake NPP. With limited oxygen exposure, a substantial fraction of these biologically active POC could be oxidized to CH₄ during the HW period through methanogenesis (Segers, 1998; Sobek et al., 2009). As lake CH₄ models rarely include this carbon source, they would likely fail to capture the related CH₄ emission pulses from floodplain lakes in the Amazon (Tan et al., 2024). Therefore, our results underscore the importance to couple floodplain hydrodynamic, POC deposition, and CH₄ dynamics for more realistic CH₄ modeling in floodplain lakes.

We believe these modifications will make the significance of our findings clearer and enhance the manuscript's contribution to the field.

R3C17:

Page 10, “the shallow depths of the oxycline”: Be specific. How shallow? Does the oxycline depth vary throughout the year?

Response:

We apologize for the confusion. This information is complemented in Method section of the revised manuscript (P26L481): “The estimated oxycline is 0.5 m, 2 m, 3.5 m, 1.2 m for LW, RW, HW and FW, respectively.”

R3C18:

Page 12, “tropical floodplains”: I would assume the magnitude of deposition should be set more by tectonics than climate, since subsiding basins with aggrading rivers should have significant floodplain deposition rates regardless of if they are in the tropics. Please clarify this point.

Response:

We appreciate the reviewer's point and agree that the tectonic factors may be even more important for the spatial variability of sediment deposition in the world. Therefore, to avoid the potential confusion, we only focus on Amazonian floodplains in the revision, which has already encompassed a large domain and drawn great interests. We have removed the discussion regarding the other tropical floodplains.

R3C19:

Page 12, “sediment deposition in the floodplains could account for a $6.1\% \pm 1\%$ of sediment discharge by Amazon River”: How does this compare to the rates of sediment transport and storage computed by Repasch et al. (2020)? Other work by Repasch and Scheingross could provide insight into OC oxidation during POC storage in the Amazonian floodplain. Repasch, M., Wittmann, H., Scheingross, J. S., Sachse, D., Szupiany, R., Orfeo, O., et al. (2020). Sediment transit time and floodplain storage dynamics in alluvial rivers revealed by meteoric ^{10}Be . *Journal of Geophysical Research: Earth Surface*, 125(7), e2019JF005419. <https://doi.org/10.1029/2019JF005419>

Response:

We appreciate this suggestion. Although the referenced study is specific to the Amazon foreland, we found it valuable to contextualize our findings within broader sediment dynamics research. Floodplains serve as significant but dynamic reservoirs of sediment and OC, undergoing considerable biogeochemical transformations that are influenced by both the physical aspects of the floodplain and the temporal scale of sediment storage. The implication is that, although a smaller percentage of the total sediment load is retained in the Amazon floodplains annually, this sediment can be subject to similar extensive biogeochemical processing due to prolonged floodplain storage, as indicated by the observed long transit times. We have added this point to the revised discussion (P19L291): “Despite a small percentage of the total sediment load retained in the Amazon floodplains annually, this sediment can be subject to extensive biogeochemical processing due to prolonged floodplain storage, as indicated by the observed long transit times (Repasch et al., 2020).”

R3C20:

Page 12, “river SSC and floodplain hydrodynamics, do not synchronize with the temporal and spatial variations of the water stage”: Based on the Exner equation you should get most rapid erosion and deposition when you have a spatial or temporal change in SSC so this offset makes sense to me. I think setting up a testable hypothesis in the Introduction based on sediment transport conservation of mass would help the reader to understand this temporal offset more clearly.

Response:

In response to the reviewer comment, we added discussions and a testable hypothesis to the revised introduction (P2L13):

“Despite its importance, our understanding of sediment deposition drivers in Amazonian floodplains remains limited. Existing studies suggest that the Amazon River inundation controls sediment deposition in the floodplains near the Andes (Aalto et al., 2003), where coarser sands are rapidly deposited during high water periods. However, this observation may not be applicable to the downstream floodplains, such as the Amazon/Solimões, which comprise the majority of Amazonian floodplain area. As the river progresses downstream, sediment composition transitions to finer sediments due to preferential deposition of coarse sediments, sediment attrition, and dilution by finer sediments from black-water tributaries (Vauchel et al., 2017). Thus, sediment

deposition in the Amazon/Solimões floodplains is less likely to be influenced by river inundation magnitude and moreso by local factors, such as inundation phases, river sediment concentration, and floodplain hydrodynamics. ”

R3C21:

Page 13, “sediment would be deposited over the vegetated area [60, 61]”: This should vary depending on the delta and the magnitudes of riverine versus tidal and storm-driven sediment fluxes. I don’t think the references here are particularly convincing, since the Mississippi River deposits enough sediment within its channel that it is very superelevated.

Response:

We agree with the reviewer. This reference has been replaced with Soler et al. (2017), which used laboratory experiments to demonstrate the impacts of vegetation on the enhancement of sediment deposition.

R3C22:

Page 13, “exacerbating channel aggradation and reducing flood capacity”: This is an important point but requires some more detail. Reaches could gradually adjust to increased SSC in other ways such as migrating more rapidly, steepening, or avulsing.

Response:

The reviewer correctly points out that floodplain river reaches could adapt to increased SSC through various geomorphological responses. We acknowledge the complexity of these potential adjustments and the multifaceted nature of river dynamics under higher sediment loads. To address this, we expanded our discussion to include these additional geomorphic responses and their potential implications for floodplain behavior (P17L258):

”River reaches may also undergo various geomorphological adjustments, including more rapid channel migration, channel steepening, or avulsion. These processes, which help rivers manage excessive sediment loads, can significantly alter floodplain morphology and hydrodynamics (Popović et al., 2021), but more research is needed to elucidate the impacts of such dynamics. ”

R3C23:

Page 14, “The POC deposition rate on the wetland area of Amazonian floodplains is relatively small”: Again, I am not sure how impactful the carbon story is. The authors argue that the methane flux could be important, unlike the CO₂ fluxes. However, they do not offer any specific numbers for Amazon CH₄ from other sources or make their own measurements of CO₂ or CH₄ outgassing. This is important because CH₄ could be produced at depth in lakes but then be oxidized to CO₂ by methanotrophs, in which case the flux may not matter. In addition, CO₂ and CH₄ have different residence times in the atmosphere so it’s important to say what timescales these fluxes occur over.

Response:

Thanks for the comment. This study focuses on POC deposition and its potential implications

for methane dynamics within Amazonian floodplain lakes. While we did not directly measure CH₄ or CO₂ outgassing in our study, recent detailed studies conducted in the same floodplain provide substantial context to support our assertions (Amaral et al., 2022; Barbosa et al., 2020). Methane produced in deeper lake sections can indeed be oxidized to CO₂ by methanotrophs before reaching the atmosphere, especially during periods when water levels change (Barbosa et al., 2020). The methane and carbon dioxide dynamics in Janauacá show significant temporal and spatial heterogeneity. This variability highlights the importance of including small-scale POC deposition processes in broader methane and carbon cycle models.

In the revision, we elaborated our discussion to clarify the importance of including POC deposition in methane models (P19L310):

“Although sediment deposition in Amazonian floodplains is not strongly tied to regional CH₄ emissions, due to its strong spatial heterogeneity and temporal variations it can influence the spatial and temporal variability of CH₄ emissions within each floodplain. For instance, we find that the POC deposition rate in the lake area is approximately five times higher than that in the wetland area. For the deepest lake area, the POC deposition rate during HW can reach 0.57 g C m⁻² day⁻¹, which becomes comparable to the measured in-lake NPP. With limited oxygen exposure, a substantial fraction of these biologically active POC could be oxidized to CH₄ during the HW period through methanogenesis (Segers, 1998; Sobek et al., 2009). As lake CH₄ models rarely include this carbon source, they would likely fail to capture the related CH₄ emission pulses from floodplain lakes in the Amazon (Tan et al., 2024). Therefore, our results underscore the importance to couple floodplain hydrodynamic, POC deposition, and CH₄ dynamics for more realistic CH₄ modeling in floodplain lakes.”

R3C24:

Page 14, “input of deposited POC to the soil carbon pools”: The authors seem to assume that adding POC will increase CH₄ production. Is this the case for floodplain lakes in the Amazon? Other limitations could be reaction kinetics, microbial abundance, and type of POC input. Since the lakes have such high biospheric POC input from photosynthesis I do not expect that adding a small amount of recalcitrant POC would significantly affect lake CH₄ emissions.

Response:

We appreciate the reviewer’s insight and agree that other factors such as reaction kinetics, microbial community structure, and the nature of the POC itself (labile vs. recalcitrant) play crucial roles in methane production. Also, our results support the reviewer’s speculation that POC deposition does not play a comparable role for CH₄ emissions to that of the soil carbon input from ecosystem primary production. However, due to the high spatial and temporal resolution of our simulations, we are able to show that POC deposition can become important for CH₄ emissions from floodplain lakes in the Amazon during HW. We have rephrased our discussion, please see the response to R3C23 above.

R3C25:

Page 14, “This Amazonian floodplain is a typical subsystem of Amazon River,”: There is

some justification for why you picked this floodplain, but it would be helpful to know how common lakes are along the Amazon. Was this motivated by findings in your prior work? Perhaps the entire “Study site” section could be moved into the main text and describe more of the earlier results at this site.

Response:

Thanks for the reviewer comment.

In response to the reviewer comment, we moved the entire “Study site” section to the last paragraph of the revised introduction and referenced to Figure 1a (previous Figure S1) when describing the study site.

R3C26:

Page 16, “Shuttle Radar Topographic Mission (SRTM)”: SRTM data is for the top of vegetation. I went through the references and found that you corrected for this in prior work, but that should be mentioned in this manuscript as well.

Response:

Thanks for the reviewer comment. In the revision, we have briefly described the bias correction method used for SRTM DEM and refer readers to Pinel et al. (2015) for more detailed description (P21L345).

“The bias over permanently inundated areas in the SRTM DEM was addressed by using the elevation data from the NASA’s Ice, Cloud, and land Elevation Satellite (ICESat) Geoscience Laser Altimeter System (GLAS). The SRTM correction also used bathymetric data from high water levels in June 2012, collected with an ADCP and GPS-linked echo sounder, along with sampling from May 2008 and August 2006. Meanwhile, vegetation-induced biases were corrected using a wetland map from Hess et al. (2015b) and a MODIS-derived vegetation height dataset from Simard et al. (2011). These corrected ground elevations were integrated using the ANUDEM v5.3 algorithm, constrained by a drainage channel network identified from Landsat imagery. For a detailed methodology, see Pinel et al. (2015).”

R3C27:

Page 16, “Manning’s roughness coefficient”: How did you delineate these regions and pick a Manning’s roughness value for each? Did you pick a value independently based on field observations or iterate on different values to find a good fit of the model to the data? How sensitive was the model to delineating these domains and picking different Manning’s n?

Response:

The delineation of bottom friction zones with different Manning’s n is detailed in Section 3.2.3 of Pinel et al. (2020), which uses the dual-season wetlands map (Hess et al., 2015b) that delineates four distinct topological classes and adjustments based on satellite data. The Manning’s n values are defined for each class following the numbers suggested by Arcement and Schneider (1989) followed by model calibrations. As discussed in Pinel et al. (2020), the hydrodynamic simulation in Janauacá is not sensitive to the choice of this parameter.

In the revision (P21L359), we briefly describe the delineation of Manning's n: "The model domain is classified into various topological regions based on the dual-season wetlands map (Hess et al., 2015b) (Supplementary Figure S1), where the corresponding Manning's roughness coefficients are defined following Arcement and Schneider (1989) and are calibrated to ensure minimum bias in the hydrodynamic simulation (Pinel et al., 2020)."

R3C28:

Page 16, "A Lowess filter": Please specify which filter parameters were used.

Response:

Thanks. We specified the filter parameters in the revision (P22L380): "A Lowess filter was applied with specified parameters (fraction=0.02, iterations=0, degree of polynomial=1)..."

R3C29:

Page 17, "...high water (HW) and falling water (FW)": How did you pick these periods? Was it arbitrary or was there a quantitative threshold?

Response:

This information is described in the caption of Figure S2: "The hydrological periods are defined by the water level elevation: LW is when water level < 15 m, HW is when water level > 20 m, and RW and FW are the periods with water level between 15 m and 20 m."

In response to the comment, we added to the information to the main context (P23L393): "The water level data is used to define 4 hydrological periods of low water (LW), rising water (RW), high water (HW) and falling water (FW): LW is when water level < 15 m, HW is when water level > 20 m, and RW and FW are the periods with water level between 15 m and 20 m (Supplementary Figure S2b)."

R3C30:

Page 17, "we define 3 cohesive sediment classes in GAIA": Do you include the effects of flocculation here? Mud is rarely transported as isolated particles in rivers and this would increase settling velocities and likely decrease the time spent in oxidating environments.

Response:

Thank you for your insightful comment regarding the exclusion of flocculation effects in our sediment model. You are correct in noting that flocculation can significantly influence the transport and deposition behaviors of mud in riverine systems. In this current version of the model, flocculation processes were not explicitly incorporated due to initial model simplifications aimed at focusing on broader hydrodynamic and sediment transport mechanisms. Additionally, significant flocculation only occurs where the Amazon sediments meet the fine sediments from brackish-water tributaries (Nittrouer et al., 2021). However, we recognize the importance of flocculation in accurately simulating sediment dynamics and acknowledged this as a limitation of the current study in the revision (P23L402):

“Please note that since significant flocculation only occurs where the Amazon sediments meet the fine sediments from brackish-water tributaries (Nittrouer et al., 2021), our model does not account for flocculation processes, which are observed to enhance the settling velocity of fine sediments in many river systems (Lamb et al., 2020).”

R3C31:

Page 17, “The deposition rate per unit area is estimated from the bed evolution rate and the default bottom layer density of 50 kg m³.”: I did not understand how the deposition rate was calculated from the bed evolution rate since the equations presented have it being calculated from sediment concentrations and settling velocity. Is the bottom layer density the water + sediment in the bedload layer? Or is that the bottom of the river?

Response:

We apologize for the confusion. Within the model, the sediment deposition rate is calculated using Equation 2, which is then converted to cumulative bed evolution as a model output variable assuming the constant bottom layer density of 50 kg m⁻³. Note that because the model does not output deposition rate directly, we computed the deposition rate from the computed bed evolution in the model output, which is equivalent to that from the model computation. The bottom layer is the river bottom, as we don’t consider the bedload.

In response to this comment, we clarified the computation of deposition rate (P23L407): “**The cumulative bed evolution is computed internally in the model computation using the deposition obtained in Equation 2 and a default bottom layer density of 50 kg m⁻³. Since the model does not output deposition rate directly, we estimated the deposition per unit area from the computed bed evolution.**”

R3C32:

Page 17, “SSC in-situ samples”: How were these samples collected? Using a Niskin bottle, Van Dorn sampler, etc.

Response:

Water for TSS was collected at both stations and stored in insulated boxes until processing. TSS was then determined by weighing particulates collected on pre-weighed Millipore HA filters (0.45 μm pore size). We would like to refer readers to the previous work (Amaral et al., 2018) for more detailed description.

R3C33:

Page 19, equation 3: Part of the results in Bouchez et al. referenced here is that POC tends to stick to finer grained sediment. Since you are using 3 different grain size classes and sources for sediment, it makes sense to assign each a different f_{POC} if they are statistically significant.

Response:

We agree with the reviewer that POC adherence depends on sediment grain size, which ideally suggests assigning different POC ratios to each sediment class. However, in our study, the

primary goal for estimating POC deposition was to provide a straightforward, comparative analysis of POC deposition rates across lake and wetland environments and to facilitate comparison with NPP data from other studies. Given the inherent uncertainties associated with POC ratios ($1.27\% \pm 0.21\%$), which can significantly vary not only due to particle size but also due to biogeochemical conditions and sampling methods, we opted to use a single average POC ratio. This approach simplifies our estimations and aligns with the modeled sediment deposition as a unified measure, as well as the overarching objective of presenting a broad landscape-scale perspective rather than detailed particle-specific deposition dynamics. We have clarified the simplification of using a unified POC ratio in the revised manuscript (P26L470):

“It should be noted that due to the dependence of POC adherence on sediment grain size (Bianchi et al., 2018), the use of a unified POC ratio across all sediment classes carries substantial uncertainty.”

R3C34:

Page 19, “the identified depths separate aerobic and anaerobic sediment.”: *Rapid burial can also produce anaerobic environments under sediment since oxygen will have to diffuse down into the layers of lake mud. I am unclear if the calculations consider post-deposition oxidation or only consider oxidation while POC is settling out of the water column.*

Response:

In our current modeling framework, we focus primarily on the initial oxidation processes while POC is settling out of the water column. This approach primarily considers the distribution of dissolved oxygen (DO) within the water column to estimate the oxygen exposure time (OET) before the POC is buried in the sediment. We acknowledge that this method does not account for post-deposition oxidation, which can indeed occur as oxygen diffuses into sediment layers. As this is primarily a hydrodynamic-sediment modeling study, our biogeochemical processes are not included in detail. The results we present are intended as an estimate to provide insights into the potential carbon pathways of deposited POC, rather than comprehensive predictions of specific biogeochemical outcomes.

We appreciate your insight and as a response, we clarified this limitation in our revised manuscript (P26L486) to ensure the scope and assumptions of our modeling approach are transparent: “The estimated OET is calculated based solely on the pre-burial oxidation condition for the settled POC, and does not account for post-deposition oxidation processes that can deplete oxygen in the sediment layers. As a result, our estimation tends to be conservative.”

R3C35:

Page 20, “macrophytes”: *Please define the difference between macrophytes and forest.*

Response:

Agree. We describe the difference between macrophytes and forest in the caption of Figure S2 (now Figure S1): “Macrophytes are aquatic plants growing near water, while forest refers to a large area densely packed with trees and undergrowth on land.”

R3C36:

Figure 1: I would like panels added to a figure showing the locations of the floodplain and where the gages are placed, similar to Figure S1 but in the main text. I am not familiar with this region and the geometry of the rivers and lake is important to understand the results. In addition, please put the circles and triangles on the figure legend. From the figure, it looks like there was a big pulse of sediment from upstream in January 2016 that the model did not capture very well... why is that? Does this affect the results?

Response:

Thanks for the comment. In response, we have changed Figure S1 to Figure 1a in the main text and put the circles and triangles on the figure legend. Please also see our response to R3C39 below.

The high SSC value observed in the data is unlikely sourced from the Amazon River in January 2016, given the limited influx from the Amazon River at the Upstream site, as illustrated by the flow momentum maps (Supplement Figure S6). There is also no sediment input from the upstream discharge at that time (Figure R1). Thus, this big pulse is likely an outlier and has not been used in the model evaluation. To prevent any confusion, this outlier has been removed in Figure 1d of the revised manuscript.

Figure R1: Time series of total river inflow during year 2015 and 2015.

R3C37:

Figure 2: I don't see these transects mentioned in the caption marked on Figure S1. Also, this caption is the first mention of POC versus floodplain carbon in general. Be sure to define POC and distinguish it from DOC.

Response:

Thanks for the reviewer comment. In the revised Figure 1a (previous Figure S1), we changed the color of the transect lines from black to magenta and increased the linewidth to make the

transects more clear. The mention of POC is now removed from the caption of Figure 2 to avoid confusion.

R3C38:

Figure 3: It is not clear to me why the authors used a polynomial fit, and in particular a cubic equation. Comparing the linear and polynomial fits requires at least reporting R^2 values for each fit to all the data. Attributing the change in behavior for very high SSC inputs could inform what relationship you expect here and motivate picking a more complicated function than a linear fit. For example, deposition is proportional to $SSC \cdot w_s$. If the settling velocity is somehow proportional to SSC (maybe you pick up larger sediment at higher SSC?) then you could propose that $w_s \sim SSC$ and $D \sim SSC^2$.

Response:

Thanks for the suggestion. In the revised Figure 3, we now display the relationship for each simulation year, as discussed in our response to R3C1. Given the limited data available per year, it is not suitable to calculate and report R^2 values. Instead of employing third-order polynomial fits for the land use change scenarios, we have switched to simpler second-order polynomial fits. We use the coefficient of the second-order term to evaluate nonlinearity; a coefficient of zero indicates linearity, while a higher magnitude signifies greater nonlinearity. This coefficient is noted in parentheses following each year.

R3C39:

Figure S1: How did the authors select this computational domain? Please put an arrow on the figure to indicate the direction each river is flowing. It would be helpful to have a topographic map showing the boundaries of the floodplain and have the areas shaded in.

Response:

Thanks for the reviewer comment. The computational domain of Janauacá floodplain is delineated as the floodable region with an elevation lower than 29 m, which is defined in the Method section (Page 21). The floodplain boundary is consistent to the model domain boundary in Figure 1a (previous Figure S1). We have added an inset of the topographic map in Figure 1a to better illustrate the domain boundary. A few arrows are also added in the upstream region to indicate the river flow direction.

Figure 1: (a) The model's computational domain (limited to the low-lying floodable regions with bottom elevation < 29 m). White labels show 8 inflow boundaries (BCF) and 3 water level open boundaries (WLBC) connecting the main channel of the Amazon River. Magenta transects at WLBCs are used to calculate the flow and sediment exchange flux between the floodplain and the Amazon River main channel. Red labels show two sampling sites of water level at WL_A and WL_B (red triangles) and three sampling sites for SSC at Upstream, Open lake and Downstream (black squares). The red dashed line outlines the lake area, which is used to estimate the lake-specific POC deposition and the corresponding oxygen exposure time (OET). The arrows in the upstream region indicate the river flow direction. The inset shows the minimized view of the domain boundary and the location of the Manacapuru gauge at the mainstem of the Amazon River. Comparison of the modeled water level (b and c) and SSC (d and e) (solid and dashed lines) against the measurements (circles and triangles) in the Janauacá floodplain. The inset in (d) shows the model-data comparison at Open lake, where model validation is based on the comparison with published mean and standard deviation values (error bar) (Amaral et al., 2022). The normalized root-mean-square error (NRMSE) is 0.04, 0.03, 0.29 and 0.21 for water level at WL_A and WL_B and for SSC at Upstream and Downstream sites, respectively.

R3C40:

Figure S2: This image is still confusing to try and understand what is considered the floodplain versus the model domain. Much of the area appears to be a lake (permanent water), though the boundaries of the lake does not agree with Figure S1. It is not clear how the areas for forest, inundated forest, etc. were delineated and the methods need to be explained in the caption or Methods section. One possible source could be inundation frequencies maps accessible at <https://global-surface-water.appspot.com/map>.

Response:

Thanks for the comment.

Figure S2 (now Figure S1) describes the topological classification of the computational domain, which is used to (a) define the Manning's roughness coefficients in the separate regions (Page 16) and (b) delineate the wetland area to estimate the corresponding POC deposition (Figure 4). The open lake in Figure 1a (previous Figure S1) is extracted from the permanent water region in Figure S2 (now Figure S1) to reflect the lake region within the Janauacá floodplain, which is used to estimate the lake POC deposition and the corresponding oxygen exposure time (OET) (Page 8).

In the revision, we further clarified the definition of the open lake in the caption of Figure 4: "The lake area and the wetland area are defined in Figure 1a and Supplementary Figure S1, respectively.", in the caption of Figure 1a: "The red dashed line outlines the lake area, which is used to estimate the lake-specific POC deposition and the corresponding oxygen exposure time (OET).", and in the caption of Figure S1: "The lake area in Figure 1a is extracted from the permanent water region of the Janauacá floodplain."

In our response to R3C27, the topological classes are defined using the dual-season wetlands map (Hess et al., 2015b) with more detailed descriptions referred to Pinel et al. (2020). The brief description was added to the Method section (P21L359): “The model domain is classified into various topological regions based on the dual-season wetlands map (Hess et al., 2015b) (Supplementary Figure S1)”.

R3C41:

Figure S3: I am not sure how well the satellite measurements and field sampling agree, so a cross-plot of field versus satellite measurements is needed to validate this approach.

Response:

The method of using satellite data collected at the same monitoring station to fill the data gaps is generally adopted given the limited sediment sampling (Rudorff et al., 2018). In our case, the data gaps are rather limited. The temporal variation is generally consistent between in-situ sampling and satellite data after applying Lowess filter (Figure S2).

In response to this comment, here we also provided the scatter plot of in-situ sampled SSC against satellite data-derived SSC. There are only ~ 70 dates when both data are available. Given the small sample size, the correlation coefficient of 0.7 indicates a reasonable comparison, especially when the unrealistic variations will be further damped with the Lowess filter.

Figure R2: Scatter plot of in-situ sampled SSC against satellite data-derived SSC. Their correlation coefficient is 0.7.

R3C42:

Figure S5: There are gaps visible between Amazon riverbank and the start of the model domain that seem important since most of the water and sediment is coming from the Amazon. Some mention of how much water flows through the secondary channel connected to the main channel versus overbank sheet flow would be helpful in addition to the delineation of fluxes in through each boundary already done in the main text.

Response:

Thank you for your insightful comment. We apologize for any confusion caused by the background satellite imagery in Figure S1 (now Figure 1a). The apparent gaps are due to the scaling of the map at different zoom levels, and do not reflect the actual modeling domain. The background map is intended to provide readers with a general understanding of the geographic location of the study area. The delineation of WLBCs is based on ALOS-1/PALSAR images (Pinel et al., 2020). We have clarified this in the revised manuscript.

Regarding the flow dynamics at the domain boundary, the reviewer is correct in noting the distinction between channelized flow and overflow. These flows are delineated at different boundaries: WLBC1 for channelized flow and WLBC2/WLBC3 for overflow, where the bottom elevation at WLBC1 is below 12 m and at WLBC2 and WLBC3, it is over 20 m. We have enhanced the figure caption to better explain these dynamics:

“The channelized flow and overflow are delineated at WLBC1 and WLBC2/WLBC3, respectively, where the bottom elevation at WLBC1 is below 12 m and at WLBC2 and WLBC3, it is over 20 m. From the late LW period to the mid-RW, channelized flow predominantly influences water and sediment exchange at WLBC1, transitioning to overflow in WLBC2 and WLBC3 when the water level exceeds the levee crest.”

R3C43:

Figure S6: Be clear which of these lines is the sediment and water from the Amazon. The main text mentions that flow into the modeling domain is positive while out is negative but it would be good to annotate the figure to remind the reader. Also, it appears that there is a net transport of water and sediment upstream (flowing backwards) into the tributaries to the lake – is that correct?

Response:

Thanks for the reviewer comment. In the revised Figure S6 (now Figure S5), we clarified that the lines in panel a represent the water flux and the meaning of positive and negative fluxes in the caption. Upward and downward arrows and texts are annotated in Figure S5 and Figure 3 to improve clarity. We also added the discharge and sediment flux from the Manacapuru gauge according to the request of Reviewer 2. We note that the red line represents the combined flux from the BCF upstream boundary. The line labeled “total” implies the net total flux from the three open boundaries. This is now clarified in the caption.

Figure S5: (a) The daily averaged flux of exchanging **water** flow between Lake Janauaca and the Amazon River mainstem at the three open boundaries WLBC1, WLBC2 and WLBC3 (Fig. 1a), their net total (**black**) and the total influx from 8 upstream inflow (BCF) boundaries (**red**). (b) The daily averaged SSC at the WLBCs. (c) and (d) are the daily averaged **discharge** and **sediment flux**, respectively, at the Manacapuru gauge. The positive and negative values represent the flux into and out of the Janauacá floodplain, respectively. Light shades indicate the variation over multiple simulation years.

R3C44:

Figure S9: It's not clear how this map was produced compared to Figure S2. Is the floodplain here equivalent to the wetlands in that caption?

Response:

The floodplain here is not exactly equivalent to the wetlands defined in Figure S1 (previous Figure S2) even though the same wetland extent product is used (Hess et al., 2015a). Figure S2 is used to demonstrate the different land cover types in Janauacá, which were used by Pinel et al. (2020) to delineate bottom friction for the hydrodynamic simulation. The different land cover zones are slightly adjusted according to satellite data (Pinel et al., 2015). This map here directly takes flooded wetlands at High Water Stage from Hess et al. (2015b). We have noted this in the caption of Figure S16 (previous Figure S9):

“Floodplain and open water are directly delineated from the flooded wetland extent product at High Water Stage (Hess et al., 2015) within the drainage basin of the Amazon/Solimões River mainstem.”

References

- Aalto, R., Maurice-Bourgoin, L., Dunne, T., Montgomery, D. R., Nittrouer, C. A., and Guyot, J.-L. (2003). Episodic sediment accumulation on Amazonian flood plains influenced by El Nino/Southern Oscillation. *Nature*, 425(6957):493–497.
- Amaral, J. H. F., Borges, A. V., Melack, J. M., Sarmiento, H., Barbosa, P. M., Kasper, D., de Melo, M. L., De Fex-Wolf, D., da Silva, J. S., and Forsberg, B. R. (2018). Influence of plankton metabolism and mixing depth on CO₂ dynamics in an Amazon floodplain lake. *Science of the Total Environment*, 630:1381–1393.
- Amaral, J. H. F., Melack, J. M., Barbosa, P. M., Borges, A. V., Kasper, D., Cortés, A. C., Zhou, W., MacIntyre, S., and Forsberg, B. R. (2022). Inundation, hydrodynamics and vegetation influence carbon dioxide concentrations in Amazon floodplain lakes. *Ecosystems*, 25(4):911–930.
- Arcement, G. J., J. and Schneider, V. R. (1989). Guide for selecting manning’s roughness coefficients for natural channels and flood plains. Technical Report FHWA-TS-84-204, Geological Survey Water-Supply, United States Government Printing Office, Washington, U.S.A.
- Barbosa, P. M., Melack, J. M., Amaral, J. H., MacIntyre, S., Kasper, D., Cortés, A., Farjalla, V. F., and Forsberg, B. R. (2020). Dissolved methane concentrations and fluxes to the atmosphere from a tropical floodplain lake. *Biogeochemistry*, 148:129–151.
- Bianchi, T. S., Cui, X., Blair, N. E., Burdige, D. J., Eglinton, T. I., and Galy, V. (2018). Centers of organic carbon burial and oxidation at the land-ocean interface. *Organic Geochemistry*, 115:138–155.
- Bonnet, M.-P., Pinel, S., Garnier, J., Bois, J., Resende Boaventura, G., Seyler, P., and Motta Marques, D. (2017). Amazonian floodplain water balance based on modelling and analyses of hydrologic and electrical conductivity data. *Hydrological Processes*, 31(9):1702–1718.
- Bouchez, J., Galy, V., Hilton, R. G., Gaillardet, J., Moreira-Turcq, P., Pérez, M. A., France-Lanord, C., and Maurice, L. (2014). Source, transport and fluxes of Amazon River particulate organic carbon: Insights from river sediment depth-profiles. *Geochimica et Cosmochimica Acta*, 133:280–298.
- Bouchez, J., Métivier, F., Lupker, M., Maurice, L., Perez, M., Gaillardet, J., and France-Lanord, C. (2011). Prediction of depth-integrated fluxes of suspended sediment in the amazon river: Particle aggregation as a complicating factor. *Hydrological processes*, 25(5):778–794.
- de Moraes Novo, E. M. L., de Farias Barbosa, C. C., de Freitas, R. M., Shimabukuro, Y. E., Melack, J. M., and Filho, W. P. (2006). Seasonal changes in chlorophyll distributions in Amazon floodplain lakes derived from MODIS images. *Limnology*, 7:153–161.

- Espinoza-Villar, R., Martinez, J.-M., Armijos, E., Espinoza, J.-C., Filizola, N., Dos Santos, A., Willems, B., Fraizy, P., Santini, W., and Vauchel, P. (2018). Spatio-temporal monitoring of suspended sediments in the Solimões River (2000–2014). *Comptes Rendus Geoscience*, 350(1-2):4–12.
- Fassoni-Andrade, A. C. and de Paiva, R. C. D. (2019). Mapping spatial-temporal sediment dynamics of river-floodplains in the Amazon. *Remote sensing of environment*, 221:94–107.
- Fleischmann, A. S., Papa, F., Fassoni-Andrade, A., Melack, J. M., Wongchuig, S., Paiva, R. C. D., Hamilton, S. K., Fluet-Chouinard, E., Barbedo, R., Aires, F., et al. (2022). How much inundation occurs in the Amazon River basin? *Remote Sensing of Environment*, 278:113099.
- Forsberg, B. R., Devol, A. H., Richey, J. E., Martinelli, L. A., and Dos Santos, H. (1988). Factors controlling nutrient concentrations in amazon floodplain lakes 1. *Limnology and Oceanography*, 33(1):41–56.
- Guilhen, J., Al Bitar, A., Sauvage, S., Parrens, M., Martinez, J.-M., Abril, G., Moreira-Turcq, P., and Sánchez-Pérez, J.-M. (2020). Denitrification and associated nitrous oxide and carbon dioxide emissions from the amazonian wetlands. *Biogeosciences*, 17(16):4297–4311.
- Herbst, T., Fuchs, M., Liebner, S., and Treat, C. C. (2024). Carbon stocks and potential greenhouse gas production of permafrost-affected active floodplains in the lena river delta. *Journal of Geophysical Research: Biogeosciences*, 129(1):e2023JG007590.
- Hess, L., Melack, J., Affonso, A., Barbosa, C., Gastil-Buhl, M., and Novo, E. (2015a). LBA-ECO LC-07 Wetland Extent, Vegetation, and Inundation: Lowland Amazon Basin. <https://doi.org/10.3334/ORNLDAAC/1284>.
- Hess, L. L., Melack, J. M., Affonso, A. G., Barbosa, C., Gastil-Buhl, M., and Novo, E. M. (2015b). Wetlands of the lowland Amazon basin: Extent, vegetative cover, and dual-season inundated area as mapped with JERS-1 synthetic aperture radar. *Wetlands*, 35:745–756.
- Junk, W. J., Bayley, P. B., Sparks, R. E., et al. (1989). The flood pulse concept in river-floodplain systems. *Canadian special publication of fisheries and aquatic sciences*, 106(1):110–127.
- Lamb, M. P., de Leeuw, J., Fischer, W. W., Moodie, A. J., Venditti, J. G., Nittrouer, J. A., Haught, D., and Parker, G. (2020). Mud in rivers transported as flocculated and suspended bed material. *Nature Geoscience*, 13(8):566–570.
- Lehner, B. and Grill, G. (2013). Global river hydrography and network routing: baseline data and new approaches to study the world’s large river systems. *Hydrological Processes*, 27(15):2171–2186.

- Li, F., Shan, Y., Huang, S., Liu, C., and Liu, X. (2021). Flow depth, velocity, and sediment motions in a straight widened channel with vegetated floodplains. *Environmental Fluid Mechanics*, 21:483–501.
- Mangiarotti, S., Martinez, J.-M., Bonnet, M.-P., Buarque, D. C., Filizola, N., and Mazzega, P. (2013). Discharge and suspended sediment flux estimated along the mainstream of the Amazon and the Madeira Rivers (from in situ and MODIS Satellite Data). *International Journal of Applied Earth Observation and Geoinformation*, 21:341–355.
- Martinelli, L. A., Victoria, R. L., Devol, A. H., Richey, J. E., and Forsberg, B. R. (1989). Suspended sediment load in the Amazon Basin: an overview. *GeoJournal*, 19:381–389.
- Martinez, J.-M., Espinoza-Villar, R., Armijos, E., and Silva Moreira, L. (2015). The optical properties of river and floodplain waters in the Amazon River Basin: Implications for satellite-based measurements of suspended particulate matter. *Journal of Geophysical Research: Earth Surface*, 120(7):1274–1287.
- Martinez, J.-M., Guyot, J.-L., Filizola, N., and Sondag, F. (2009). Increase in suspended sediment discharge of the Amazon River assessed by monitoring network and satellite data. *Catena*, 79(3):257–264.
- Mayorga, E. and Logsdon, M.G. and Ballester, M.V.R. and Richey, J.E. (2012). Lba-eco cd-06 amazon river basin land and stream drainage direction maps. Data set. Available online from Oak Ridge National Laboratory Distributed Active Archive Center, Oak Ridge, Tennessee, USA.
- McClain, M. E. and Naiman, R. J. (2008). Andean influences on the biogeochemistry and ecology of the amazon river. *BioScience*, 58(4):325–338.
- Montanher, O. C., Novo, E. M., Barbosa, C. C., Rennó, C. D., and Silva, T. S. (2014). Empirical models for estimating the suspended sediment concentration in Amazonian white water rivers using Landsat 5/TM. *International Journal of Applied Earth Observation and Geoinformation*, 29:67–77.
- Moreira-Turcq, P., Jouanneau, J., Turcq, B., Seyler, P., Weber, O., and Guyot, J.-L. (2004). Carbon sedimentation at Lago Grande de Curuai, a floodplain lake in the low Amazon region: insights into sedimentation rates. *Palaeogeography, Palaeoclimatology, Palaeoecology*, 214(1-2):27–40.
- Murguia-Flores, F., Jaramillo, V., and Gallego-Sala, A. (2023). Assessing methane emissions from tropical wetlands: uncertainties from natural variability and drivers at the global scale. *Global Biogeochemical Cycles*, 37(9):e2022GB007601.
- Nittrouer, C. A., DeMaster, D. J., Kuehl, S. A., Figueiredo Jr, A. G., Sternberg, R. W., Faria, L. E. C., Silveira, O. M., Allison, M. A., Kineke, G. C., Ogston, A. S., et al. (2021). Amazon sediment transport and accumulation along the continuum of mixed fluvial and marine processes. *Annual Review of Marine Science*, 13:501–536.

- Parrinello, G. and Kondolf, G. M. (2021). The social life of sediment. *Water History*, 13(1):1–12.
- Petsch, D. K., Cionek, V. d. M., Thomaz, S. M., and Dos Santos, N. C. L. (2023). Ecosystem services provided by river-floodplain ecosystems. *Hydrobiologia*, 850(12):2563–2584.
- Pfeiffer, A. M., Collins, B. D., Anderson, S. W., Montgomery, D. R., and Istanbuluoglu, E. (2019). River bed elevation variability reflects sediment supply, rather than peak flows, in the uplands of washington state. *Water Resources Research*, 55(8):6795–6810.
- Pinel, S., Bonnet, M.-P., S. Da Silva, J., Sampaio, T. C., Garnier, J., Catry, T., Calmant, S., Fragoso Jr, C. R., Moreira, D., Motta Marques, D., et al. (2020). Flooding dynamics within an Amazonian floodplain: Water circulation patterns and inundation duration. *Water Resources Research*, 56(1):e2019WR026081.
- Pinel, S., Bonnet, M.-P., Santos Da Silva, J., Moreira, D., Calmant, S., Satgé, F., and Seyler, F. (2015). Correction of interferometric and vegetation biases in the SRTMGL1 spaceborne DEM with hydrological conditioning towards improved hydrodynamics modeling in the Amazon Basin. *Remote Sensing*, 7(12):16108–16130.
- Popović, P., Devauchelle, O., Abramian, A., and Lajeunesse, E. (2021). Sediment load determines the shape of rivers. *Proceedings of the National Academy of Sciences*, 118(49):e2111215118.
- Repasch, M., Wittmann, H., Scheingross, J. S., Sachse, D., Szupiany, R., Orfeo, O., Fuchs, M., and Hovius, N. (2020). Sediment transit time and floodplain storage dynamics in alluvial rivers revealed by meteoric 10be. *Journal of Geophysical Research: Earth Surface*, 125(7):e2019JF005419.
- Riquetti, N. B., Beskow, S., Guo, L., and Mello, C. R. (2023). Soil erosion assessment in the Amazon basin in the last 60 years of deforestation. *Environmental Research*, 236:116846.
- Rudorff, C. M., Dunne, T., and Melack, J. M. (2018). Recent increase of river–floodplain suspended sediment exchange in a reach of the lower Amazon River. *Earth Surface Processes and Landforms*, 43(1):322–332.
- Saunois, M., Stavert, A. R., Poulter, B., Bousquet, P., Canadell, J. G., Jackson, R. B., Raymond, P. A., Dlugokencky, E. J., Houweling, S., Patra, P. K., et al. (2019). The global methane budget 2000–2017. *Earth System Science Data Discussions*, 2019:1–136.
- Sawakuchi, H. O., Bastviken, D., Sawakuchi, A. O., Ward, N. D., Borges, C. D., Tsai, S. M., Richey, J. E., Ballester, M. V. R., and Krusche, A. V. (2016). Oxidative mitigation of aquatic methane emissions in large amazonian rivers. *Global change biology*, 22(3):1075–1085.
- Segers, R. (1998). Methane production and methane consumption: a review of processes underlying wetland methane fluxes. *Biogeochemistry*, 41:23–51.

- Simard, M., Pinto, N., Fisher, J. B., and Baccini, A. (2011). Mapping forest canopy height globally with spaceborne lidar. *Journal of Geophysical Research: Biogeosciences*, 116(G4).
- Smith, L., Melack, J., Hammond, D., Smith, L. K., and Hammond, D. E. (2003). Carbon, nitrogen, and phosphorus content and Pb-derived burial rates in sediments of an Amazon floodplain lake. *Amazoniana*, 17:413–436.
- Sobek, S., Durisch-Kaiser, E., Zurbrügg, R., Wongfun, N., Wessels, M., Pasche, N., and Wehrli, B. (2009). Organic carbon burial efficiency in lake sediments controlled by oxygen exposure time and sediment source. *Limnology and Oceanography*, 54(6):2243–2254.
- Sobrinho, R. L., Bernardes, M. C., Abril, G., Kim, J.-H., Zell, C., Mortillaro, J.-M., Meziane, T., Moreira-Turcq, P., and Sinninghe Damsté, J. S. (2016). Spatial and seasonal contrasts of sedimentary organic matter in floodplain lakes of the central amazon basin. *Biogeosciences*, 13(2):467–482.
- Soler, M., Colomer, J., Serra, T., Casamitjana, X., and Folkard, A. M. (2017). Sediment deposition from turbidity currents in simulated aquatic vegetation canopies. *Sedimentology*, 64(4):1132–1146.
- Tan, Z., Yao, H., Melack, J., Grossart, H.-P., Jansen, J., Balathandayuthabani, S., Sargsyan, K., and Leung, L. R. (2024). A lake biogeochemistry model for global methane emissions: Model development, site-level validation, and global applicability. *Journal of Advances in Modeling Earth Systems*, 16(10):e2024MS004275.
- Tassi, P., Benson, T., Delinares, M., Fontaine, J., Huybrechts, N., Kopmann, R., Pavan, S., Pham, C.-T., Taccone, F., and Walther, R. (2023). GAIA-a unified framework for sediment transport and bed evolution in rivers, coastal seas and transitional waters in the TELEMAC-MASCARET modelling system. *Environmental Modelling & Software*, 159:105544.
- Trigg, M. A., Bates, P. D., Wilson, M. D., Schumann, G., and Baugh, C. (2012). Floodplain channel morphology and networks of the middle amazon river. *Water Resources Research*, 48(10).
- Vauchel, P., Santini, W., Guyot, J. L., Moquet, J. S., Martinez, J. M., Espinoza, J. C., Baby, P., Fuertes, O., Noriega, L., Puita, O., et al. (2017). A reassessment of the suspended sediment load in the madeira river basin from the andes of peru and bolivia to the amazon river in brazil, based on 10 years of data from the hybam monitoring programme. *Journal of Hydrology*, 553:35–48.
- Villar, R. E., Martinez, J.-M., Le Texier, M., Guyot, J.-L., Fraizy, P., Meneses, P. R., and de Oliveira, E. (2013). A study of sediment transport in the Madeira River, Brazil, using MODIS remote-sensing images. *Journal of South American Earth Sciences*, 44:45–54.

- Ward, N. D., Keil, R. G., Medeiros, P. M., Brito, D. C., Cunha, A. C., Dittmar, T., Yager, P. L., Krusche, A. V., and Richey, J. E. (2013). Degradation of terrestrially derived macromolecules in the amazon river. *Nature Geoscience*, 6(7):530–533.
- Wohl, E. (2021). An integrative conceptualization of floodplain storage. *Reviews of Geophysics*, 59(2):e2020RG000724.
- Wu, W. (2007). *Computational river dynamics*. Crc Press.

Reviewer 1

This manuscript is much improved after what appears to be careful attention and response to the comments of the three reviewers. Many of my primary concerns have been satisfied, although the following relatively minor issues remain.

Response:

We thank the reviewer for the valuable comments and recommendations. We have carefully addressed the reviewer's suggestions as follows:

R1C1:

Abstract: Andes mountains (missing an "s" on mountains)

Response:

Corrected.

R1C2:

Abstract: the state encompassing Manaus is Amazonas, which might be a better fit than Amazônia in this context. Elsewhere you use anglicized words to refer to the greater Amazon area (e.g., Amazonian) so it's a bit of a divergence to see the Portuguese usage here. This is admittedly a minor point, and one of the authors is Brazilian, so perhaps it's fine.

Response:

Corrected to "Amazonas" as suggested.

R1C3:

L6: I brought up the 15% vs the 6.1% you reference in the abstract in my last review. It would be useful here to hedge this number at this location by pointing out the difference in domain for these two numbers, especially since you use the exact same phrasing (Amazonian floodplains) in the abstract and on this line to refer to two different areal domains. (I do appreciate the expanded discussion later in the paper about the potential limitations/uncertainties regarding your new estimate.)

Response:

We appreciate the reviewer pointing out this potential confusion. We have revised the text in both the abstract and introduction to clarify these differences in domain and ensure consistent terminology.

Revised Abstract: "By upscaling the sediment deposition rate ($1.33 \pm 0.24 \text{ kg m}^{-2} \text{ yr}^{-1}$), we estimate that $77.3 \pm 13.9 \text{ Mt}$ of sediment could be deposited in **the floodplains along the Amazon/Solimões River mainstem** every year, representing $6.1\% \pm 1\%$ of sediment discharge from the river."

The 15% value provides an example of the ratio of sediment trapped along the passing 390 km-long between Itacoatiara and Óbidos. While this figure is frequently cited in the literature, we

consider it region-specific and not necessarily appropriate as a reference in our study. To avoid potential confusion, we have removed this value from the revised introduction.

Revised Introduction: "A substantial portion of the sediment does not make its way to the ocean but is instead deposited over Amazonian floodplains (Mangiarotti et al., 2013). "

R1C4:

Figure 1: I appreciate the addition of (e) but remain suspect of the goodness of fit between modeled and observed SSCs since this is on a log-log scale. Please indicate if the black line is a best fit or 1:1 line in both (c) and (e).

Response:

Thanks for the comment. The use of log-log scale to compare the modeled and observed SSCs was because SSC values are significantly larger at the Downstream station but much smaller at the other two stations. In the revision, the scale has been reverted from log scale to a normal scale. The black dashed line in both (c) and (e) indicate 1:1 line. We added this information to the caption of Figure 1.

Figure R1: Updated Figure 1.

R1C5:

L133: I suggest "reached a maximum" instead of "topped" here

Response:

Corrected as suggested.

R1C6:

L152: I'm confused by the numbers here. Is this the same 15% referenced from [6] in the intro? As written, they appear to refer to two different quantities, that is, fractional deposition rate of mainstem sediment on floodplains vs retention of sediment influx to the floodplain itself. Is the 15% just a coincidence on both of these numbers? I don't see 11.7% in Figure 2B anywhere.

Response:

We apologize for the confusion. The number is indeed the same 15% referenced in the introduction (Mangiarotti et al., 2013). As noted in our response to R1C3, this value is the sediment trapping ratio of the floodplains along a 390 km Amazon River reach between Itacoatiara and Óbidos. Notably, the sediment trapping ratio is defined in the Methodology section as: “The sediment trapping ratio is defined as the ratio of the Amazon River sediment trapped by the floodplain, which is calculated by dividing the deposited sediment in the floodplain by the river sediment load measured at Manacapuru.” As emphasized in the revision, 11.7% is the percentage of the 0.34 Mt mean annual sediment influx deposited in the floodplain. Because the sediment influx through the river-floodplain boundary is only a fraction of the sediment discharge of the Amazon River, this value is not the sediment trapping ratio and the comparison of 15% vs. 11.7% is problematic. Since the calculation of sediment trapping ratio depends on the location and length of river reaches, to ensure a more cautious comparison, we have removed this specific comparison from the revised manuscript. However, we retain the comparison based on the deposition rate reported in Mangiarotti et al. (2013).

“Of the 0.34 Mt mean annual sediment influx through the river-floodplain boundary (Figure 2a), ~11.7% was deposited in the floodplain (Supplementary Figure S10) and the rest was returned to the Amazon River during FW (Figure 2b).”

Moreover, we have also added a new figure showing the deposition ratio over the simulation years as Supplementary Figure S10, where in the caption we defined the deposition ratio as the proportion of sediment deposited in the floodplain relative to the sediment influx.

Figure S10: The deposition ratio over the simulation years. The deposition ratio is the proportion of sediment deposited in the floodplain relative to the sediment influx.

R1C7:

Figure 3: Please double check the y-axis label of subplot c. Do the units work?

Response:

Our apologies. There was a typo in the y-axis label, which should be $D_{exp}/D_{baseline} - SSC_{exp}/SSC_{baseline}$. This typo has been corrected in the revision.

R1C8:

L232: I remain confused by the use of 11.7 here and 6.1 in the abstract. I believe they are referring to different quantities (see comment on L152) but as written it is hard to distinguish and keep track of the ratio of deposition to the mainstem flux and ratio of retention of the inflow to the floodplain. I suggest clarifying to indicate which is which when numbers are used.

Response:

Thanks for the comment. The two numbers indeed represent the quantities noted by the reviewer. To improve clarity, we have added the definition of the deposition ratio (i.e., 11.7%) in the caption of Supplementary Figure S10. Please see our response to R1C6 for more explanations. In response to this comment, the sentence has been rephrased to:

“This result highlights that floodplain lakes are a major sink of water kinetic energy in the Amazon Basin and thus largely responsible for the trapping of ~11.7% of sediment influx from the Amazon River.”

R1C9:

Figure S7: What is the meaning of the dashed lines in the figures? Are they best fit lines? If so I am surprised by their orientations, especially in (a) and (c), since the r values are negative and the slopes should be, as well.

Response:

We apologize for the confusion and agree that the 1:1 lines in Figure S7 are not appropriate. We have updated this figure with best-fit lines.

Figure S7: Scatter plots of annual mean deposition rate, water level and SSC. The black lines are the best-fit lines.

R1C10:

Finally, the figshare link includes Python pickle (.pkl) files. As described at <https://docs.python.org/3/library/pickle.html> the pickle module is not secure, and users should only unpickle data they trust, since pickle files can execute arbitrary code on a user's machine. This makes it an inappropriate format for supplemental files for the manuscript, and I strongly suggest providing data in a more standard format like netCDF which does not have the security implications of pickle files. For this reason, I have not opened the files, so it's possible something even simpler than netCDF like CSV or JSON would be appropriate.

Response:

Thank you for the suggestion. We have uploaded the files in CSV format and provided a detailed description in a README file. The original simulation output, which exceeds 450 GB in size, will be made available upon request.

Reviewer 2

The authors made numerous changes to address earlier comments from myself (Reviewer 3) and the other reviewers, resulting in a much-improved manuscript. The re-submitted product presents a much more detailed and nuanced picture of floodplain sedimentation and POC dynamics in the Amazon.

Response:

We sincerely thank the reviewer for the positive feedback and comments. We have carefully addressed the reviewer's recommendations as follows:

R2C1:

The authors argue that POC deposition does not matter when compared to NPP, except where it is deposited under anoxic conditions, where it might be oxidized to CH₄. The degree that methane production in Amazonian floodplain lakes occurs and is significant in the regional and global climate is still not clear to me. I would be fine with the authors just reporting a negative result – that floodplain POC is not significant compared to organic input from primary productivity in the Amazon – to ensure that the OC interpretations are well-supported and described consistently. Right now the paper has two main takeaways (drivers of floodplain sediment deposition and potential OC emissions as methane) and I think the paper works well as a sedimentation study without flashy results about methane emissions.

Response:

Thank you for raising this point regarding the potential significance of POC deposition in influencing CH₄ emissions and its role in the broader context of regional and global climate. We acknowledge in the abstract that the evidence we present primarily supports the conclusion that “the deposition of sediment-associated organic carbon only plays a minor role in fueling carbon dioxide and methane emissions from Amazonian floodplains.”

In the last paragraph of the discussion, we have rephrased the text to clarify that while the role of POC deposition to CH₄ emissions in specific conditions (e.g., anoxic lake areas) cannot be ignored, once put into the context of annual CH₄ emissions over the whole floodplain, the role of this process is negligible relative to that of NPP. Additionally, we have de-emphasized the CH₄ modeling implications to ensure that the focus remains on sediment deposition drivers.

“Although sediment deposition in Amazonian floodplains is not an important driver of regional CH₄ emissions, due to its strong spatial heterogeneity and seasonal variations, this process can influence CH₄ emissions under specific conditions (i.e., deep lake areas during HW) within each floodplain. For instance, we find that the POC deposition rate in the lake area is approximately five times higher than that in the wetland area. For the deepest lake area, the POC deposition rate during HW can reach 0.57 g C m⁻² day⁻¹, which becomes comparable to the measured in-lake NPP. With limited oxygen exposure, a substantial fraction of these biologically active POC could be oxidized to CH₄ during the HW period through methanogenesis (Segers, 1998; Sobek et al., 2009). ~~As lake CH₄ models rarely include this carbon source, they would likely fail to capture the related CH₄ emission pulses from floodplain lakes in the Amazon (Tan et al., 2024). Therefore, our results underscore the importance to couple floodplain hydrodynamic, POC deposition, and CH₄ dynamics for more realistic CH₄ modeling in floodplain lakes.~~ However, since deep lake areas only comprise a small fraction of the entire floodplains and the HW period only lasts 2~3 months, the overall contribution of POC deposition to regional annual CH₄ emissions is minor.”

R2C2:

Line 20: Unclear what “sediment attrition” means. Maybe “sediment abrasion?”

Response:

Thank you for the comment. “sediment attrition” refers to the the reduction in size and quantity of sediment particles due to processes like grinding or scraping, while “abrasion” specifically refers to the wearing away of sediment particles against fixed surfaces. As our intent here is to emphasize broadly on the reduction of sediment size as they travel downstream, we prefer the use of “attrition”.

R2C3:

Line 32: I don't fully understand how dams would cause a 100% decrease in sediment transport. Does that mean no sediment would come from the Amazon to the ocean? This could be rephrased to be more clear.

Response:

Flecker et al. (2022) shows that many dams proposed in sediment-rich river reaches in the western Amazon have high sediment trapping efficiencies of up to 100%. To avoid confusion, we rephrased this sentence to “It is estimated that many proposed hydropower plants in the Amazon, without prioritizing hydrological connectivity, can potentially reduce sediment transport to downstream river reaches by up to 100% (Flecker et al., 2022).”

R2C4:

Line 65: “Local floodplains” – local to the Amazon or somewhere else? I found this wording confusing.

Response:

Sorry for the confusion. “Local floodplains” refer to the floodplain local to the central Amazon Basin. In the revision, this sentence is rephrased to “Previous applications of 2-D hydrodynamic models in the central Amazon Basin as well as **its** floodplains ...”

R2C5:

Figure 1: The context imagery in the Panel A insets are really hard to see. I would use outlines of the continent and basin rather than imagery. In Panel B, it seems like there is an error in your boundary conditions to cause these flat sections of model results. Please explain why this is and if it matters in the main text. Panel E – please add error bars to the points. This will make the green points look like they have a much better fit because the SSC error bars are quite large. Caption – add a first sentence describing the figure overall before going into each panel.

Response:

Thanks for the comment. In the revised Figure 1, we have updated the insets in Panel A to enhance readability. In Panel B, the bias in the simulated water levels arises from the complex bathymetry of the Amazon floodplains, which cannot be fully resolved even with a 30-m resolution. This bias is generally less than 1 m during four out of five hydrological years, reaching approximately 3 m only in 2010. It does not affect the analysis, as the hydrodynamic simulation has been rigorously validated across various locations (Pinel et al., 2020), and the bias is localized to this specific area close to the river channel bank. In the revised manuscript, we have clarified the cause of this bias: “**In (b), the bias in the simulated water level at WL_A during LW is due to the unresolved floodplain bathymetry at this specific location.**” In Panel E, error bars have been added to the green dots for improved clarity. Additionally, the scale has been reverted from log scale to a normal scale. An introductory sentence has been added to the figure caption.

Figure 1: **The setup and validation of the hydrodynamic-sediment coupled model.** (a) The model’s computational domain (limited to the low-lying floodable regions with bottom elevation < 29 m). White labels show 8 inflow boundaries (BCF) and 3 water level open boundaries (WLBC) connecting the main channel of the Amazon River. Magenta transects at WLBCs are used to calculate the flow and sediment exchange flux between the floodplain and the Amazon River main channel. Red labels show two sampling sites of water level at WL_A and WL_B (red triangles) and three sampling sites for SSC at Upstream, Open lake and Downstream (black squares). The red dashed line outlines the lake area, which is used to estimate the lake-specific POC deposition and the corresponding oxygen exposure time (OET). The arrows in the upstream region indicate the river flow direction. The inset shows the minimized view of the domain boundary and the location of the Manacapuru gauge at the mainstem of the Amazon River. Comparison of the modeled water level (b and c) and SSC (d and e) (solid and dashed lines) against the measurements (circles and triangles) in the Janauacá floodplain. The inset in (d) shows the model-data comparison at Open lake, where model validation is based on the comparison with published mean and standard deviation values (error bar) (Amaral et al., 2022). **The black dashed line in both (c) and (e) represents the 1:1 line.** The normalized root-mean-square error (NRMSE) is 0.04, 0.03, 0.29 and 0.21 for water level at WL_A and WL_B and for SSC at Upstream and Downstream sites, respectively. **In (b), the bias in the simulated water level at WL_A during LW is due to the unresolved floodplain bathymetry at this specific location.**

R2C6:

Line 131: It might be helpful to set the reader’s expectations for what deposition rates should scale to. Each of the reviewers had slightly different ideas about this, so reminding the reader what you expect here might help.

Response:

Agree. To address this, we have added a brief discussion of expected deposition rates in similar Amazonian floodplains, based on previous studies, and used these as a reference to contextualize our modeled results. Please note that a more detailed comparison of the deposition rates is provided in the Discussion section.

“Due to the effect of floodplain hydrodynamics, the simulated sediment deposition shows large and distinct spatial variability during the four hydrological periods, **with rates ranging from less than 1 kg m⁻² yr⁻¹ in the areas with persistently fast flows to over 5 kg m⁻² yr⁻¹ in the deeper lake areas (Figures 2d-2g).** These results fall within the reported range of 0.5 to 8.8 kg m⁻² yr⁻¹ for Amazonian floodplains (Bourgoin et al., 2007; Mangiarotti et al., 2013; Forsberg et al., 1989; Smith et al., 2003), as detailed in the Discussion section.”

R2C7:

Figure 2: Please make the color bars go to negative values to capture the full range of deposition rates from your model outputs.

Response:

In the revised Figure 2, the colormap scale has been updated to $-2.5 \sim 2.5 \text{ kg m}^{-2} \text{ yr}^{-1}$.

R2C8:

Lines 272-294: This is a long paragraph with two different estimations of sediment trapping in floodplains. Integrating the two estimates, with one as an upper and the other as a lower bound, would help condense this paragraph and improve the flow of ideas.

Response:

Thank you for your suggestion. We understand the importance of integrating the two estimates for clarity. However, we believe the 6.1% estimate based on the floodplain area is more robust. The river-length-based estimate of 211 Mt per year, while also informative, may be less accurate due to its inherent assumption. To address your comment, we have restructured the paragraph into two paragraphs to improve the flow and focus. We emphasized the 6.1% estimate as the primary result while briefly mentioning the alternative estimate for context.

“By upscaling the sediment deposition rate to the floodplain area within the drainage basin of the Amazon/Solimões River mainstem (i.e., $58,103 \text{ km}^2$), we estimate that the Amazon/Solimões floodplains could trap a total amount of $77.3 \pm 13.9 \text{ Mt}$ of sediment per year and $0.98 \pm 0.18 \text{ Mt}$ of POC per year using the POC/sediment mass ratio of 1.27% (Bouchez et al., 2014). Given 1200 Mt sediment drained to the Atlantic Ocean by the Amazon River per year (Martinelli et al., 1989), sediment deposition in the floodplains could account for a $6.1 \pm 1\%$ of sediment discharge by the Amazon River. Despite a small percentage of the total sediment load retained in the Amazon floodplains annually, this sediment can be subject to extensive biogeochemical processing due to prolonged floodplain storage, as indicated by the observed long transit times (Repasch et al., 2020).

The estimated deposition is influenced by several uncertainties, such as the variability in floodplain trapping efficiency and the potential overestimation of the floodplain area (Fleischmann et al., 2022). Moreover, Janaucá may not represent all floodplains along the Solimões River, as there are sites with varying ratios of local drainage extent to lake area (Forsberg et al., 1988; Sobrinho et al., 2016). Our upscaling estimation focuses on the mainstem of the Solimões/Amazon River but excludes all tributaries, assuming that Janaucá is only representative of floodplains adjacent to the mainstem. This figure likely underestimates the total sediment deposition in the basin, because the floodplains of the tributaries could also trap a substantial amount of sediment (Trigg et al., 2012). For comparison, using the simulated river-length-specific deposition rate of $0.033 \text{ Mt km}^{-1} \text{ yr}^{-1}$ and the total mainstem length of 3349 km for the Solimões/Amazon River from HydroRIVERS (Lehner and Grill, 2013), an alternative river-length-based upscaling approach yields a much higher total deposition estimate of 211 Mt per year. However, this figure is likely an overestimate because floodplains are not always present along the river and on both sides.”

R2C9:

Line 355: Why did you pick 29 m as the elevation bound of the floodplain? Please say why and/or reference prior work.

Response:

The computational domain delineation was initially carried out in Pinel et al. (2020), where a value of 29 m, which is 4.6 m above the observed maximum water level, was chosen. This value is considered sufficient to ensure that no water from the Amazon River will exceed this level. This information is clarified in the revision: “The computational domain of Janauacá is delineated as the floodable region with an elevation lower than 29 m (4.6 m above the observed maximum water level) (Pinel et al., 2020).”

R2C10:

Line 481: Why did you pick these values for the oxycline? Please briefly describe why.

Response:

We use a dissolved oxygen (DO) value of 2 mg L⁻¹, a standard value commonly referenced in the literature (e.g., (Winton et al., 2023)), to determine the oxycline from the DO data collected during the four hydrological periods by Amaral et al. (2018). In the revision, the sentence has been rephrased to improve clarity:

“First, we identify the oxycline—the depth below which DO concentrations fall below 2 mg L⁻¹—using DO data collected during the four hydrological periods by Amaral et al. (2018) at the wind-exposed (WE) site of Janauacá. The estimated oxycline depths are 0.5 m, 2 m, 3.5 m, 1.2 m for LW, RW, HW and FW, respectively.”

R2C11:

Figure S2: I appreciate that the values for these thresholds were added to the main text, but I still don't understand why these values were selected. Is there a quantitative justification for the thresholds, for instance splitting up the annual hydrograph into 4 equal time periods?

Response:

Thanks for the comment. The four hydrological periods are typically defined from water level time series in Amazon floodplains (Pinel et al., 2020; Rudorff et al., 2018). Here, the thresholds are chosen because (a) they ensure comparable periods within a hydrological year, and (b) the high water threshold is near the top of the riverbank. This information has been added to the caption of Figure S2.

References

- Amaral, J. H. F., Borges, A. V., Melack, J. M., Sarmiento, H., Barbosa, P. M., Kasper, D., de Melo, M. L., De Fex-Wolf, D., da Silva, J. S., and Forsberg, B. R. (2018). Influence of plankton metabolism and mixing depth on CO₂ dynamics in an Amazon floodplain lake. *Science of the Total Environment*, 630:1381–1393.
- Amaral, J. H. F., Melack, J. M., Barbosa, P. M., Borges, A. V., Kasper, D., Cortés, A. C., Zhou, W., MacIntyre, S., and Forsberg, B. R. (2022). Inundation, hydrodynamics and vegetation influence carbon dioxide concentrations in Amazon floodplain lakes. *Ecosystems*, 25(4):911–930.

- Bouchez, J., Galy, V., Hilton, R. G., Gaillardet, J., Moreira-Turcq, P., Pérez, M. A., France-Lanord, C., and Maurice, L. (2014). Source, transport and fluxes of Amazon River particulate organic carbon: Insights from river sediment depth-profiles. *Geochimica et Cosmochimica Acta*, 133:280–298.
- Bourgoin, L. M., Bonnet, M.-P., Martinez, J.-M., Kosuth, P., Cochonneau, G., Moreira-Turcq, P., Guyot, J.-L., Vauchel, P., Filizola, N., and Seyler, P. (2007). Temporal dynamics of water and sediment exchanges between the Curuaí floodplain and the Amazon River, Brazil. *Journal of Hydrology*, 335(1-2):140–156.
- Flecker, A. S., Shi, Q., Almeida, R. M., Angarita, H., Gomes-Selman, J. M., García-Villacorta, R., Sethi, S. A., Thomas, S. A., Poff, N. L., Forsberg, B. R., et al. (2022). Reducing adverse impacts of Amazon hydropower expansion. *Science*, 375(6582):753–760.
- Fleischmann, A. S., Papa, F., Fassoni-Andrade, A., Melack, J. M., Wongchuig, S., Paiva, R. C. D., Hamilton, S. K., Fluet-Chouinard, E., Barbedo, R., Aires, F., et al. (2022). How much inundation occurs in the Amazon River basin? *Remote Sensing of Environment*, 278:113099.
- Forsberg, B., Godoy, J. M., Victoria, R., and Martinelli, L. A. (1989). Development and erosion in the Brazilian Amazon: A geochronological case study. *GeoJournal*, 19:399–405.
- Forsberg, B. R., Devol, A. H., Richey, J. E., Martinelli, L. A., and Dos Santos, H. (1988). Factors controlling nutrient concentrations in amazon floodplain lakes 1. *Limnology and Oceanography*, 33(1):41–56.
- Lehner, B. and Grill, G. (2013). Global river hydrography and network routing: baseline data and new approaches to study the world’s large river systems. *Hydrological Processes*, 27(15):2171–2186.
- Mangiarotti, S., Martinez, J.-M., Bonnet, M.-P., Buarque, D. C., Filizola, N., and Mazzega, P. (2013). Discharge and suspended sediment flux estimated along the mainstream of the Amazon and the Madeira Rivers (from in situ and MODIS Satellite Data). *International Journal of Applied Earth Observation and Geoinformation*, 21:341–355.
- Martinelli, L. A., Victoria, R. L., Devol, A. H., Richey, J. E., and Forsberg, B. R. (1989). Suspended sediment load in the Amazon Basin: an overview. *GeoJournal*, 19:381–389.
- Pinel, S., Bonnet, M.-P., S. Da Silva, J., Sampaio, T. C., Garnier, J., Catry, T., Calmant, S., Fragoso Jr, C. R., Moreira, D., Motta Marques, D., et al. (2020). Flooding dynamics within an Amazonian floodplain: Water circulation patterns and inundation duration. *Water Resources Research*, 56(1):e2019WR026081.
- Repasch, M., Wittmann, H., Scheingross, J. S., Sachse, D., Szupiany, R., Orfeo, O., Fuchs, M., and Hovius, N. (2020). Sediment transit time and floodplain storage dynamics in alluvial rivers revealed by meteoric ^{10}Be . *Journal of Geophysical Research: Earth Surface*, 125(7):e2019JF005419.

- Rudorff, C. M., Dunne, T., and Melack, J. M. (2018). Recent increase of river–floodplain suspended sediment exchange in a reach of the lower Amazon River. *Earth Surface Processes and Landforms*, 43(1):322–332.
- Segers, R. (1998). Methane production and methane consumption: a review of processes underlying wetland methane fluxes. *Biogeochemistry*, 41:23–51.
- Smith, L., Melack, J., Hammond, D., Smith, L. K., and Hammond, D. E. (2003). Carbon, nitrogen, and phosphorus content and Pb-derived burial rates in sediments of an Amazon floodplain lake. *Amazoniana*, 17:413–436.
- Sobek, S., Durisch-Kaiser, E., Zurbrügg, R., Wongfun, N., Wessels, M., Pasche, N., and Wehrli, B. (2009). Organic carbon burial efficiency in lake sediments controlled by oxygen exposure time and sediment source. *Limnology and Oceanography*, 54(6):2243–2254.
- Sobrinho, R. L., Bernardes, M. C., Abril, G., Kim, J.-H., Zell, C., Mortillaro, J.-M., Meziane, T., Moreira-Turcq, P., and Sinninghe Damsté, J. S. (2016). Spatial and seasonal contrasts of sedimentary organic matter in floodplain lakes of the central amazon basin. *Biogeochemistry*, 13(2):467–482.
- Tan, Z., Yao, H., Melack, J., Grossart, H.-P., Jansen, J., Balathandayuthabani, S., Sargsyan, K., and Leung, L. R. (2024). A lake biogeochemistry model for global methane emissions: Model development, site-level validation, and global applicability. *Journal of Advances in Modeling Earth Systems*, 16(10):e2024MS004275.
- Trigg, M. A., Bates, P. D., Wilson, M. D., Schumann, G., and Baugh, C. (2012). Floodplain channel morphology and networks of the middle amazon river. *Water Resources Research*, 48(10).
- Winton, R. S., López-Casas, S., Valencia-Rodríguez, D., Bernal-Forero, C., Delgado, J., Wehrli, B., and Jiménez-Segura, L. (2023). Patterns and drivers of water quality changes associated with dams in the tropical andes. *Hydrology and Earth System Sciences*, 27(7):1493–1505.